# Effects of super-powerful tropospheric Western Pacific phenomenon of September–October 2018 on ionosphere over China: Results from oblique sounding

Leonid F. Chernogor[1, 2, 3], Kostiantyn P. Garmash[1], Qiang Guo[2], Victor T. Rozumenko[1], Yu Zheng[3]

[1]V. N. Karazin Kharkiv National University, Kharkiv, 61022, Ukraine
e-mail: *Leonid.F.Chernogor@gmail.com*
e-mail: *Garmash@karazin.ua*
e-mail: *vtrozumenko@gmail.com*
[2]Harbin Engineering University, Harbin, 150001, China
e-mail: *guoqiang@hrbeu.edu.cn*
[3]Qingdao University, Qingdao, 266071, China
e-mail: *zhengyu@qdu.edu.cn*

*Correspondence to*: Yu Zheng (zhengyu@qdu.edu.cn)

**Abstract.** Doppler measurements at oblique propagation paths from the City of Harbin, People's Republic of China (PRC), to ten HF radio broadcast stations in the PRC, Japan, Mongolia, and the Republic of Korea captured the response in the ionosphere to the super typhoon Kong-Rey action from 30 September 2018 to 6 October 2018. The Harbin Engineering University coherent software defined radio system accumulates the database containing the complex amplitudes of the radio signals acquired along 14 propagation paths since 2018. The complex amplitudes are used for calculating the temporal dependences of the Doppler spectra and signal amplitudes, and the Doppler spectra are used to plot the Doppler shift as a function of time, $f_D(t)$, for all rays. The scientific objectives of this study are to reveal the possible perturbations caused by the action of typhoon Kong-Rey, and to estimate the magnitudes of wave parameters of the ionospheric plasma and radio signals. The amplitudes, $f_{Da}$, of the Doppler shift variations were observed to noticeably increase (factor of ~2–3) on 1–2 and 5–6 October 2018, while the 20–120 min periods, $T$, of the Doppler shift variations suggest that the wavelike disturbances in the ionosphere are caused by atmospheric gravity waves. The periods and amplitudes of quasi-sinusoidal variations in the Doppler shift, which have been determined for all propagation paths, may be used to estimate the amplitudes, $\delta_{Na}$, of quasi-sinusoidal variations in the electron density. Thus, $T \approx 20$ min and $f_{Da} \approx 0.1$ Hz yield $\delta_{Na} \approx 0.4\%$, whereas $T \approx 30$ min and $f_{Da} \approx 0.2$ Hz give $\delta_{Na} \approx 1.2\%$. If $T \approx 60$ min and $f_{Da} \approx 0.5$ Hz, then $\delta_{Na} \approx 6\%$. The periods $T$ are found to change within the 15–120 min limits, and the Doppler shift amplitudes, $f_{Da}$, show variability within the 0.05–0.4 Hz limits.

# 1 Introduction

A violent tropical cyclone arising in the northwestern Pacific Ocean is termed the typhoon. In record-breaking typhoons, the atmospheric pressure drops down to 870 hPa, while the pressure deficit reaches 140 hPa, and the wind speed attains a maximum of 85 m s$^{-1}$, with 94 m s$^{-1}$ maximum gusts.

Prasad et al. (1975) were the first to ascertain the influence of meteorological processes, namely, tropical cyclones on the ionosphere. Hung and Kuo (1978, 1985) described observations of traveling ionospheric disturbances (TIDs) as the manifestations of the atmospheric gravity waves (AGWs) generated by hurricanes. Krishmam Raju et al. (1981) have studied the influence of infrasound generated by thunderstorms. Observations of AGWs from meteorological origin have been reported elsewhere (Boška and Šauli, 2001; Šindelarova et al., 2009; Chernigovskaya et al., 2015).

The coupling between typhoons and the ionosphere and overlying magnetosphere occurs via a range of mechanisms. Observational studies conducted in recent years have shown that typhoons significantly influence the upper atmosphere, including the ionosphere. Recently, theoretical studies on the coupling between the lower and upper atmosphere, which occur through AGWs, have been published as well (Hickey et al., 2001, 2011; Kuester et al., 2008, Gavrilov and Kshevetskii. 2015, Karpov and Kshevetskii, 2017). Such a mechanism for coupling is naturally called the acoustic–gravity mechanism (Chernogor, 2006, 2012).

Typhoons are accompanied by water vapor condensation, the development of powerful convective lift, and the appearance of severe thunderstorms (Mikhailova et al., 2000, 2002). Lightning discharges act to generate electromagnetic emissions that may be capable of heating electrons and perturbing the electron density in the ionospheric *D* region (Nickolaenko and Hayakawa, 1995; Chernogor, 2006, 2012). The large enough fluxes of electromagnetic emissions lead to pitch angle scattering of energetic electrons in the radiation belts via wave-particle interaction, and consequently, part of the electrons precipitates into the lower ionosphere (Inan et al., 2007; Voss et al., 1984, 1998; Bortnik et al., 2006). As a result, secondary perturbations in the plasma conductivity (~100–150 km altitude) and in the geomagnetic and electric fields capable of affecting processes in the magnetosphere can arise. Such a mechanism should be considered as an electromagnetic mechanism (Chernogor, 2006, 2012).

An increase in the quasi-stationary electric field may be of different origin (Mikhailova et al., 2000; Isaev et al., 2002, 2010; Sorokin et al., 2005; Pulinets et al., 2014). Localized ~$10^{-9}$–$10^{-8}$ A m$^{-2}$ electric currents arise within thunderstorm clouds at 10–15 km altitude, which disturb the global electric circuit and increase by 1–2 orders of magnitude quasi-sinusoidal electric fields that are mapped to the ionosphere and magnetosphere and affect the motion of high-energy electrons trapped in the radiation belts. Under certain conditions, the precipitation of these electrons may occur into the ionosphere, and a repeated coupling between the subsystems in the ocean–atmosphere–ionosphere–magnetosphere (OAIM) system happen (Chernogor, 2006, 2012). This mechanism for coupling may be termed the electric mechanism (Chernogor, 2006, 2012). Thus, powerful typhoons are capable of governing the coupling between the subsystems in the OAIM system.

A lot of studies deal with the acoustic–gravity mechanism, and therefore this mechanism has been studied better than the others. The major role AGWs play in coupling different atmospheric regions under the influence of typhoons and hurricanes on the upper atmosphere is discussed by Okuzawa et al. (1986), Xiao et al. (2007), Vanina–Dart et al. (2007), Afraimovich et al. (2008), Polyakova and Perevalova (2011, 2013), Zakharov and Kunitsyn (2012), Suzuki et al. (2013), Chou et al. (2017), Li et al. (2017, 2018), Chum et al. (2018), Zakharov et al. (2019, 2022). These researchers invoked various measurement techniques for probing the ionosphere: GPS technology, ionosondes, rocket techniques, and HF Doppler technique.

The manifestations of the ionospheric response to the super typhoons Hagibis, Ling-Ling, Faxai, and Lekima in radio wave characteristics in the 5–10 MHz band have been studied by Chernogor et al. (2021, 2022) and Zheng et al., (2022). The variations in the main features of radio waves have been determined, and aperiodic and quasi-sinusoidal perturbations in the electron density have been ascertained.

The effect of sudden stratospheric warming events, variations in space weather, solar activity, and of AGWs on the coupling between the subsystems in the atmosphere–ionosphere system has been analyzed in the review by Yiğit et al. (2016), whereas twenty years earlier, the review by Hocke and Schlegel (1996) could only point to the AGW/TID relationship. Since then, data have been compiled for some parameters of medium-scale traveling ionospheric disturbances (MSTIDs), one of the mechanisms for affecting the ionosphere by typhoons. The parameters of interest to typhoon/ionosphere coupling studies include the propagation direction. Of particular interest to the current study, which is conducted in the area roughly to the west of Japan, are data collected in Japan. Using airglow images, a clear preference for southwestward propagation has been shown by Kubota et al. (2000) and Shiokawa et al. (2003), while Fukushima et al. (2012) observations made over a seven-year period in Indonesia estimated the propagation direction to be within ±30 degrees from the source directions of MSTIDs in 81% of the MSTID events. Otsuka et al. (2008) investigated a relationship between nighttime MSTIDs and sporadic $E$ layer, another phenomenon of interest to typhoon/ionosphere coupling. Observations made in the western hemisphere are in agreement with those made over the Pacific Ocean (Paulino et al., 2016; Frissell et al. 2014; Paulino et al., 2018). The latter study by Paulino et al. is noteworthy because it showed that the observed anisotropy in the propagation direction can fully be explained by the filtering process of the wind.

The results of the latest observations are presented in papers by Kong et al. (2017), Li et al. (2018), Zhao et al. (2018), Song et al. (2019), Wen and Jin (2020), Chen et al. (2020), Ke et al. (2020), Zhao et al. (2020), Das et al. (2021), Freeshah et al. (2021), Chernogor et al. (2021, 2022), Zakharov et al. (2019, 2022). They show that the influence of typhoons on the ionosphere might be expected to significantly depend on typhoon parameters, local time, season, solar cycle changes, and on the state of atmospheric and space weather. To date, there remains insufficient knowledge about this influence and therefore the study of the ionospheric response to any new typhoon is of interest. In this paper, super typhoon Kong-Rey, the most powerful worldwide typhoon in 2018, has been chosen to analyze the ionospheric response to the typhoon action.

The scientific objectives of this study is to determine the response of the ionosphere to approaching super typhoon Kong-Rey by making use of variations in Doppler spectra, Doppler shift, and HF signal amplitudes recorded at oblique

propagation paths, as well as to estimate the parameters of the ionospheric perturbations. An estimate of the joint influence of the typhoon and the dusk terminator is also a phenomenon of interest. The observations were made using the Harbin Engineering University, the People's Republic of China (PRC), multifrequency multiple path coherent software defined radio system for probing the ionosphere at oblique incidence. The data sets discussed in this paper may be obtained from the website at https://dataverse.harvard.edu/dataset.xhtml?persistentId=doi:10.7910/DVN/VHY0L2 (Garmash, 2022), and the software for Passive 14-Channel Doppler Radar may be obtained from https://dataverse.harvard.edu/dataset.xhtml?persistentId=doi:10.7910/DVN/MTGAVH (Garmash, 2021).

## 2 General information on super typhoon Kong-Rey

Table 1 presents basic information on typhon Kong-Rey, a Category 5 tropical storm; part of the information was retrieved from http://agora.ex.nii.ac.jp/digital-typhoon/summary/wnp/s/201825.html.en. It shows that the typhoon originated on 29 September 2018 and ceased to exist on 6 October 2018. A noticeable decrease in pressure took place on 30 September 2018, when the pressure reached a minimum value of 900 hPa, while the pressure deficit attained a maximum of 105 hPa (see also Figure 1). The wind speed attained a maximum of 215 km h$^{-1}$ or 60 m s$^{-1}$, with 77 m s$^{-1}$ maximum gusts. The largest radius of the storm wind was 260 km, and the largest radius of the gale wind was 750 km. The length of the typhoon path is estimated to be 4,107 km, with an average speed of 23.6 km h$^{-1}$ or 6.6 m s$^{-1}$. The dynamic wind pressure is estimated to attain 2.25 kPa, with a maximum gust pressure of 3.8 kPa. The kinetic energy of the rotating air was estimated to be close to $1.65 \times 10^{18}$ J, while the mean power was estimated to attain $1.7 \times 10^{13}$ W. On 1 and 2 October 2018, the super typhoon energy was a maximum, and on 2 October 2018 the typhoon moved to the system probing the ionosphere closer by ~600 km (Figure 1). On 5 October 2018, the typhoon was 250 km off the shores of the PRC, when the pressure deficit was observed to be ~30 hPa.

Table 1. Basic parameters of super typhoon Kong-Rey (Courtesy of Asanobu KITAMOTO, Digital typhoon, National institute of informatics, Japan).

| | |
|---|---|
| Birth | 2018-09-29 06:00:00 UTC |
| Death | 2018-10-06 12:00:00 UTC |
| Lifetime | 174 h/7.250 days |
| Minimum Pressure | 900 hPa |
| Pressure Maximum Deficit | 105 hPa |
| Maximum Wind Speed | 215 km h$^{-1}$ (60 m s$^{-1}$) |
| Largest Radius of Storm Wind | 260 km |
| Largest Radius of Gale Wind | 750 km |
| Length of Movement | 4,107 km |
| Average Speed | 23.6 km h$^{-1}$ (6.56 m s$^{-1}$) |

| | |
|---|---|
| Range of Movement | Latitude 25.3°: Longitude 16.7° |
| Typhoon Kinetic Energy | $1.65 \times 10^{18}$ J |
| Typhoon Power | $1.7 \times 10^{13}$ W |
| Rainfall | 250–300 mm h$^{-1}$ |
| Maximum Pressure Drop | –25 hPa / 6 h; –40 hPa / 12 h<br>–65 hPa / 24 h; –96 hPa / 48 h |
| Data Start | 2018-09-28 00:00:00 UTC |
| Data End | 2018-10-07 12:00:00 UTC |
| Data Duration | 228 h / 9.500 days |

## 3 Analysis of the state of space weather

A comprehensive analysis of space weather is required in order to ascertain the ionospheric response to the super typhoon action.

Figure 2 accumulates knowledge regarding the state of space weather during the super typhoon Kong-Rey event. First, consider the parameters of the solar wind (retrieved from https://omniweb.gsfc.nasa.gov/form/dx1.html). Under quiet conditions, the proton number density is observed to be close to $5 \times 10^6$ m$^{-3}$, whereas on 29 September 2018 and 1, 3, and 5 October 2018, it shows increases up to $(15–20) \times 10^6$ m$^{-3}$. On 26 and 30 September 2018, as well as on 3–4 October 2018, the plasma flow speed increases from ~400 km s$^{-1}$ to 500–520 km s$^{-1}$. During the same UT period, plasma temperature increases from ~$(2–3) \times 10^4$ K to ~$(1.2–1.5) \times 10^5$ K, while the dynamic solar wind pressure increases from ~1 nPa to 4–5 nPa. The $B_y$ component of the interplanetary magnetic field exhibits temporal variability within the –5.9 nT to 11.6 nT limits, while the $B_z$ component changes from –4.7 nT to 4.0 nT.

a                                    b

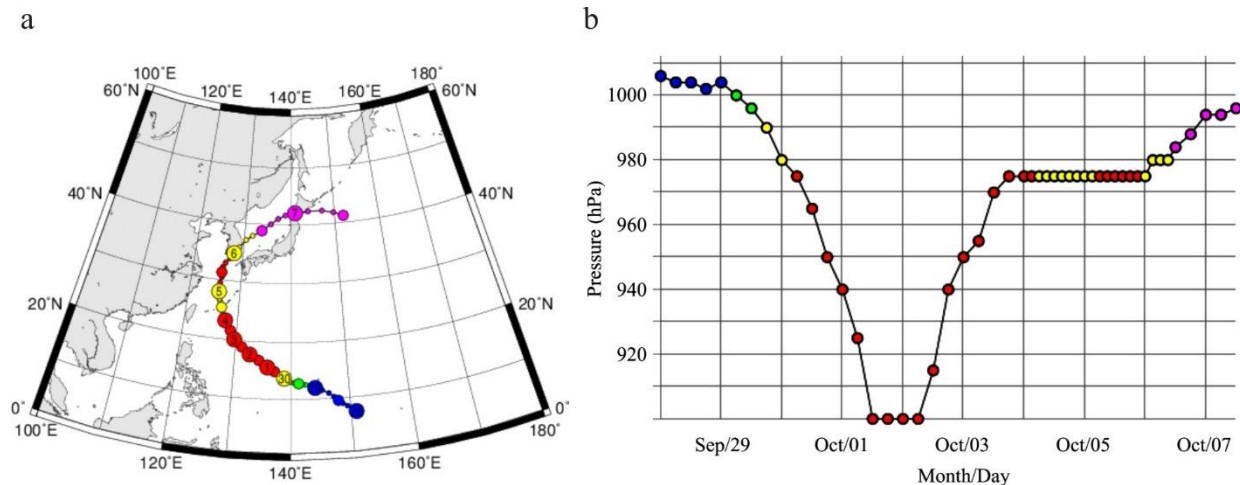

**Figure 1: Super typhoon Kong-Rey (a) trajectory and (b) pressure (Courtesy of Asanobu KITAMOTO, Digital Typhoon, National Institute of Informatics, Japan).**

131    On 26 and 29 September 2018, as well as on 1 and 3–4 October 2018, the calculated energy input, $\varepsilon_A$, into the
132 Earth's magnetosphere from the solar wind shows increases up to ~5 GJ s$^{-1}$.

133    The $K_p$ index exhibits sporadical increases to 3.0–3.7, while the $D_{st}$ index shows fluctuations from –16 nT to
134 16 nT.

135    Table 2 presents temporal variations in the radio flux at 10.7 cm ($F_{10.7}$) index for the 26 September to 09 October
136 2018 period.

137    Thus, solar activity and the state of space weather were conducive to observing the ionospheric effects from typhoon
138 Kong-Rey. Only on 7 October 2018, a moderate magnetic storm started, with $K_{p\max} = 5.3$, and $D_{st\min} \approx -53$ nT. Thus, the days
139 of 26 and 27 September 2018, the first half of 28 and entire 29 September 2018, and partially the days of 1 and 2 October
140 2018 were weakly disturbed. The magnetic storm occurred after the typhoon ceased to exist, from October 7 through 9,
141 2018, when the Doppler shifts exhibited variations greater than those observed under the action of the typhoon, which
142 justifies the need for a thorough analysis of space weather. Consequently, 28 September 2018, and 4 October 2018, have
143 been chosen to be quiet time references.

144 **4 Analysis of the state of the ionosphere**

145 The state of the ionosphere was monitored by the ionosonde nearest to Harbin, i.e., the WK546 URSI code ionosonde
146 located in the city of Wakkanai (45.16°N, 141.75°E) in Japan (Guo et al., 2019a, 2019b, 2020; Chernogor et al., 2020; Luo
147 et al., 2020). Figure 3 shows UT variations in the main ionogram parameters. The minimum frequency, $f_{\min}$, observed on
148 ionograms exhibited fluctuations around 1.5 MHz. The critical frequencies of the $E$ layer, $f_{oE}$, were close to 3 MHz during
149 the day, and gradually decreased to 1.8–2.0 MHz in the evening hours. At night, measurements of $f_{oE}$ were impossible. The
150 blanketing frequency of the sporadic $E$ layer, most often, showed fluctuations within the 3–8 MHz limits, however
151 sometimes it could attain 13–15 MHz. The ordinary-wave critical frequency $f_oF_2$ was observed to be 4–6 MHz during the
152 day and to decrease to 3.0–3.5 MHz at night.

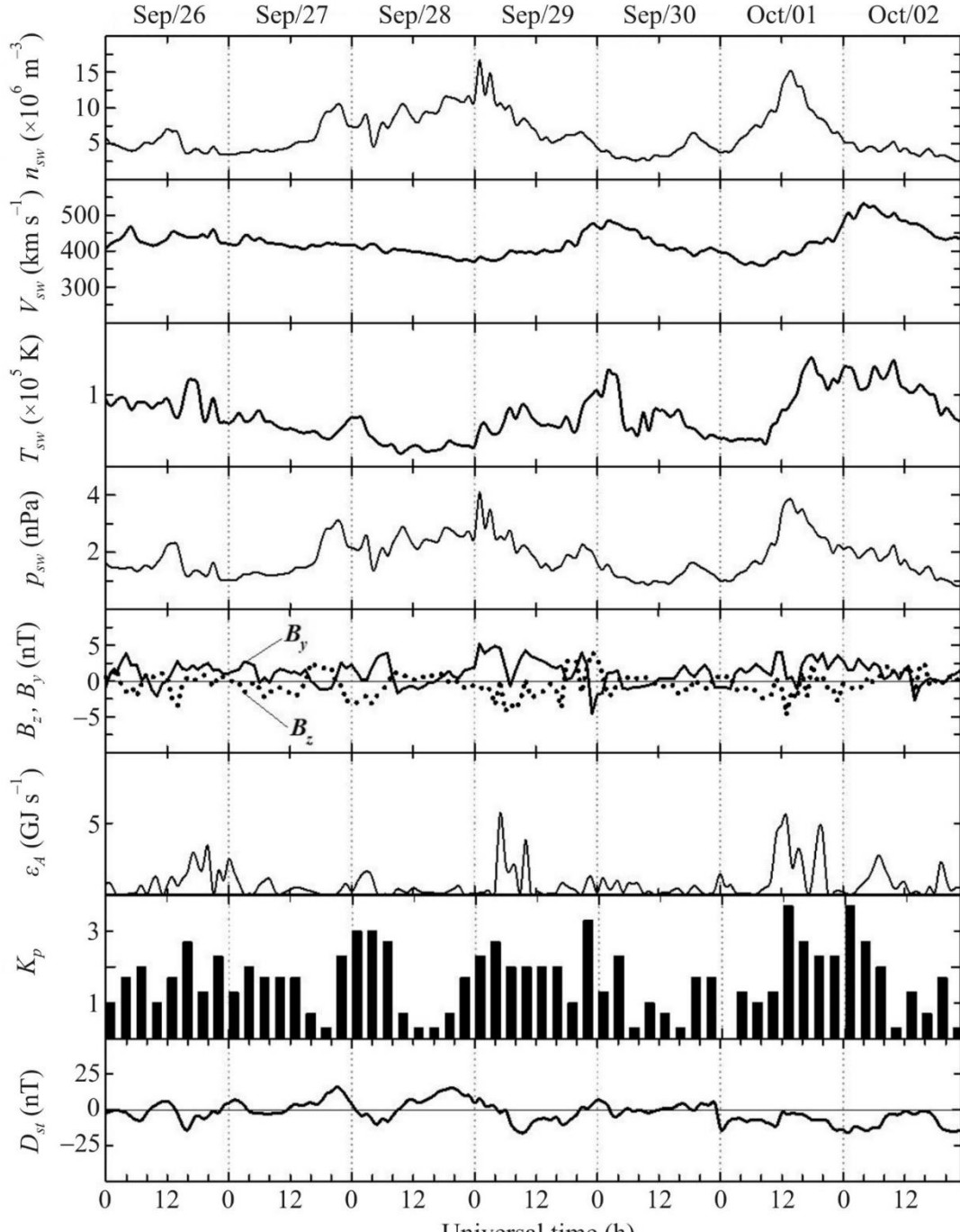

153

Figure 2: Universal time dependences of the solar wind parameters: proton number density $n_{sw}$, plasma flow speed $V_{sw}$, plasma temperature $T_{sw}$, dynamic solar wind pressure $p_{sw}$, $B_z$ and $B_y$ components of the interplanetary magnetic field, calculated energy input, $\varepsilon_A$, into the Earth's magnetosphere from the solar wind; and $K_p$- and $D_{st}$-indices for the 26 September–2 October, 2018 period.

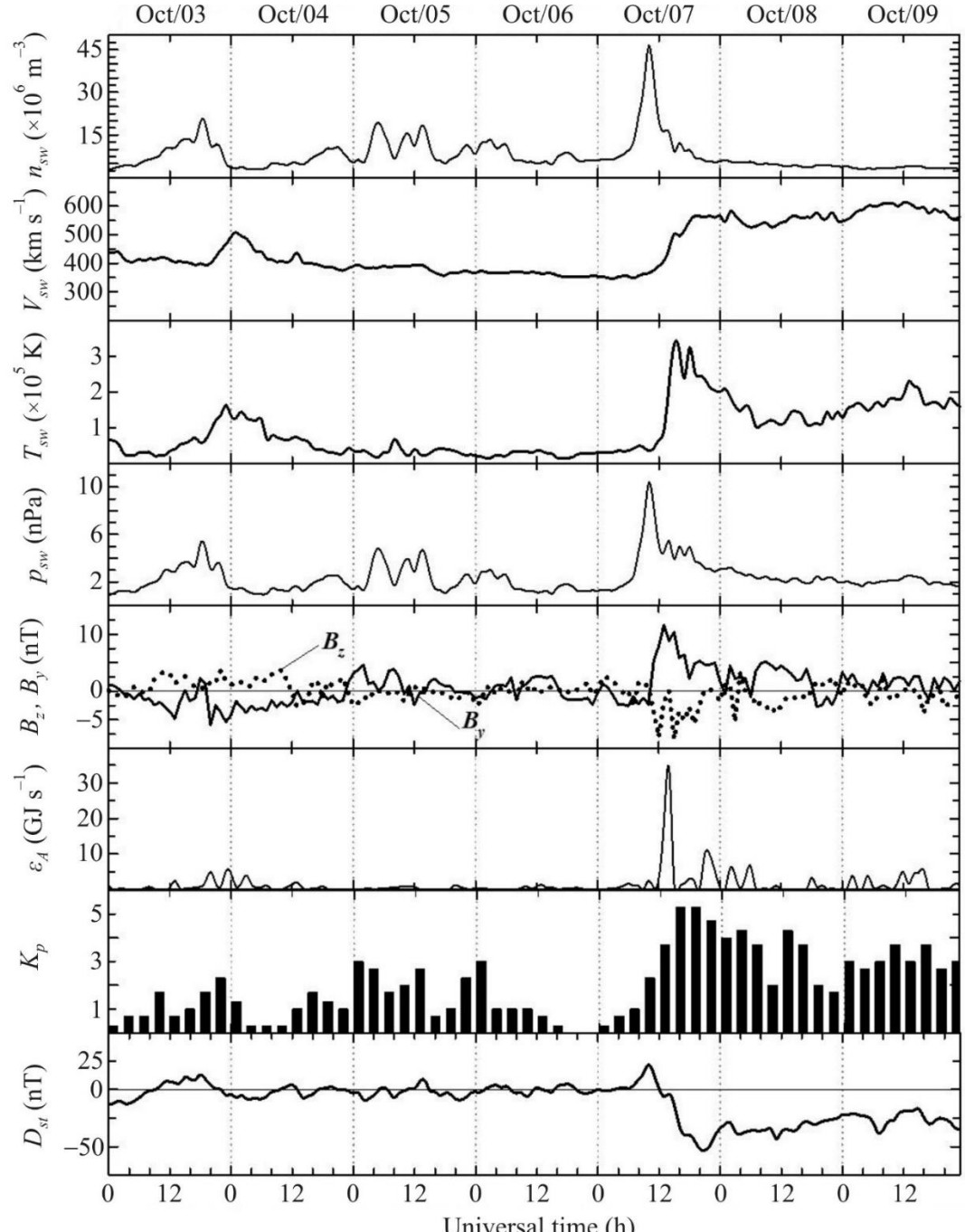

**Figure 2: Continued for the 03–09 October 2018 period.**

Table 2. Daily $F_{10.7}$ index for the 26 September 2018–9 October 2018 period

| Date (2018) | 26 Sep. | 27 Sep. | 28 Sep. | 29 Sep. | 30 Sep. | 1 Oct. | 2 Oct. | 3 Oct. | 4 Oct. | 5 Oct. | 6 Oct. | 7 Oct. | 8 Oct. | 9 Oct. |
|---|---|---|---|---|---|---|---|---|---|---|---|---|---|---|
| $F_{10.7}$ | 69.3 | 67.4 | 69.4 | 68.9 | 68.5 | 70.3 | 67.1 | 68.4 | 67.2 | 68.7 | 68.6 | 69.4 | 68.7 | 69.3 |

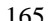

**Figure 3: Universal time variations in ionogram parameters determined with an update rate of one measurement per 1 h: critical frequencies $f_{min}$, $f_0E$, $f_0E_s$, and $f_0F_2$ for the 29 September–2 October 2018 period.**

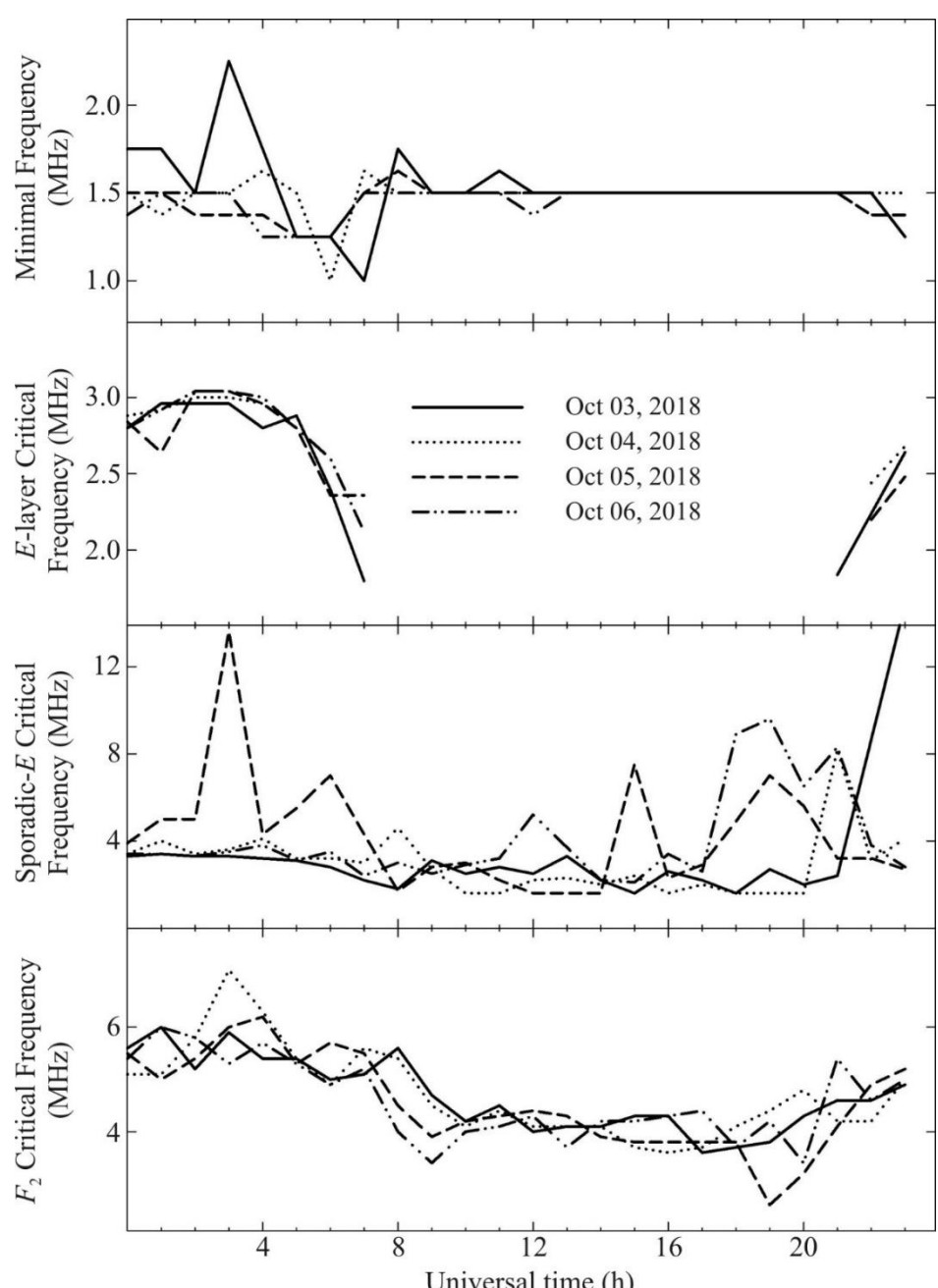

**Figure 3: Continued for the 3–6 October 2018 period.**

Figure 4 shows temporal variations in the virtual heights. The virtual heights $h'_E$ are observed to vary mainly within
the 95–105 km, whereas the virtual heights $h'_{Es}$ most frequently show variations within the 90–110 km limits, which could
sometimes exhibit an increase to 140–160 km.

## 5 Instrumentation and techniques

The study of the effects from the super typhoon was conducted using the Harbin Engineering University multifrequency multiple path coherent software defined radio system for probing the ionosphere at oblique incidence (Guo et al., 2019a, 2019b, 2020; Chernogor et al., 2020; Luo et al., 2020). The system utilizes radio transmissions of broadcast stations located in the PRC, the Republic of Korea, Japan, the Russian Federation, and Mongolia, the signals of which are received and processed at the Harbin Engineering University campus (45.78°N, 126.68°E).

Continuous monitoring of the dynamic processes operating in the ionosphere is done along 14 propagation paths in the ~5 – 10 MHz band (Table 3, Figure 5) as described by Guo et al. (2019*a*, 2019*b*, 2020), Chernogor et al. (2020), and Luo et al. (2020). In the event under study, post-analysis of the data acquired along six propagation paths has shown that the data are not suitable for processing.

Monitoring the dynamic processes in the ionosphere is done via calculating the temporal dependences of the Doppler spectra and signal amplitudes. The Doppler spectra are used to plot the Doppler shift as a function of time, $f_D(t)$, for all rays under analysis.

Spectrum analysis is performed applying the autoregressive technique of Marple (1987), which provides a frequency resolution of 0.01 Hz over ~20 s intervals with 7.5 s time resolution.

The $f_D(t)$ dependences can be used to calculate the trend $\overline{f}_D(t)$, the fluctuations $\delta f_D(t) = f_D(t) - \overline{f}_D(t)$, and the systems spectral analysis can be undertaken over 60 – 280 min intervals to select harmonics in the $T \leq 5$ min and $T = 10 - 140$ min period ranges (Chernogor, 2008).

For over about 50 years, one of the co-authors, Chernogor, L. F., has developed a general methodology for revealing perturbations launched in the ionosphere by various significant inputs of energy into the lithosphere–atmosphere–ionosphere–magnetosphere system. To put the development of this methodology into perspective, one should remember that the development of this methodology has been accompanied by tremendous, unparalleled technological advances, from analogue instruments and film-based recordings to new software-defined radio sensors.

Used in this study, the radio system probes the ionosphere at 14 radio propagation path midpoints of the order of 1,000 km distance apart, which are randomly distributed in the ~100–300 km altitude range. Generally, the perturbations under study may be produced either by an impulsive release of energy at a fixed location, as in the case of an earthquake, or by significant releases of energy, which change their location and power as well as persist for a few days, as in the case of a typhoon. On the way from their origin to the radio propagation path midpoints in the upper atmosphere, the perturbations may undergo various nonlinear transformations. In the case of a typhoon event, atmospheric gravity waves, generated via a nonlinear prosses (Drobyazko and Krasil'nikov, 1975), travel up to the ionosphere (partially dissipating their energy for heating the neutral air) and launch secondary gravity waves in the wave breaking regions (see, e.g., Vadas et al., 2003; Vadas and Crowley, 2010). The latter waves in the atmosphere modulate the electron density, which can result in the level of reflection variability, the appearance of a few rays, or in some cases, in diffuseness in the Doppler measurements or spread *F*

in ionograms, which is an indicator of the occurrence of plasma irregularities in the ionosphere (see, e.g., Perkins (1973)). As

a consequence, the measurements taken at each midpoint produce a single realization of a random process, which means that

the Doppler or amplitude signatures of the sources of perturbations are unrepeatable neither in time nor in space. The

observational methodology that enables identification and investigation of such perturbations arising from any deposition of

large amounts of energy include the following basic principles invoked consecutively. (i) During the initial stage of

employing this methodology, the perturbations originating from a particular powerful source are in principle not

distinguishable qualitatively from the perturbations caused by energy released from any other powerful source. (ii) A

particular powerful source releasing energy can be associated with any changes in the character of the signal (Doppler shift,

Doppler spectrum, the number of rays, discrete spectrum broadening, changes in the signal amplitude, etc.), in accordance

with (i) above). This condition is necessary but insufficient. (iii) Intercomparisons between the behavior of radio wave

characteristics observed prior to and after an impulsive release of energy must be made. (iv) An intercomparison of the

behavior of the radio wave characteristics observed on the day when a particular massive release of energy occurred and

during quiet time reference days must be made. Any differences may be due to this particular source. Points (iii) and (iv)

serve as control stages. During these stages, the effects that are not associated with the massive release of energy are

discarded. (v) The magnitudes of the speeds of propagation of the disturbances must have a physical significance and

correspond to known types of waves (seismic, atmospheric gravity waves, infrasound, magnetohydrodynamic). This stage

proves sufficiency. (vi) The data acquired over a large (10–14, in the case of the Harbin Engineering University system)

number of propagation paths must be consistent with each other to prove sufficiency additionally. (vii) The main signs of a

particular powerful source should be observed during other analogous events. First of all, this principle refers to the observed

velocities and types of waves. The speeds of perturbations traveling to the radio propagation path midpoints should be

contained within the speed limits characteristic of each particular wave type.

**6 Ionospheric results from oblique incidence sounding**

The post analysis of the data collected during the super typhoon Kong-Rey event has shown that the transmissions from only

eight of the fourteen transmitters in the ~6–10 MHz band are suitable for studying the super typhoon Kong-Rey event

(Figure 5). The specifications of the transmitters and radio-wave propagation paths are presented in Table 3. Since the

lengths of the propagation paths are found to be ~1000–2000 km and the frequencies of the sounding radio waves are

relatively small, the sounding waves were reflected either from the $E$ layer or from the sporadic $E$ during the daytime when

the Doppler shift, $f_D$, was observed to be ~0 Hz. Consequently, these measurements were ineffective in observing

ionospheric dynamics. At night, the radio waves were reflected from the ionospheric $F$ region and only sometimes from the

sporadic $E$. The Doppler shift of the radio waves reflected from the $F$ region exhibited variations from ~0.1 Hz to ~0.5 Hz

and greater. Therefore, the measurements made during nights, evenings, and mornings could be used for studying

ionospheric dynamics. The observations suffer another drawback: the transmitters of the broadcast radio stations did not
transmit continuously.

The Doppler spectra, Doppler shift, and the signal amplitudes in the main rays along all propagation paths exhibited
relatively small variability. The smallest variations were observed to occur on 28 September 2018, which was chosen to be
as a quiet time reference.

The Doppler spectra and Doppler shift in the main rays, and the signal amplitudes showed the greatest variability on 7
and 8 October 2018 because it occurred due to the magnetic storm, which is not dealt with in detail in this study.

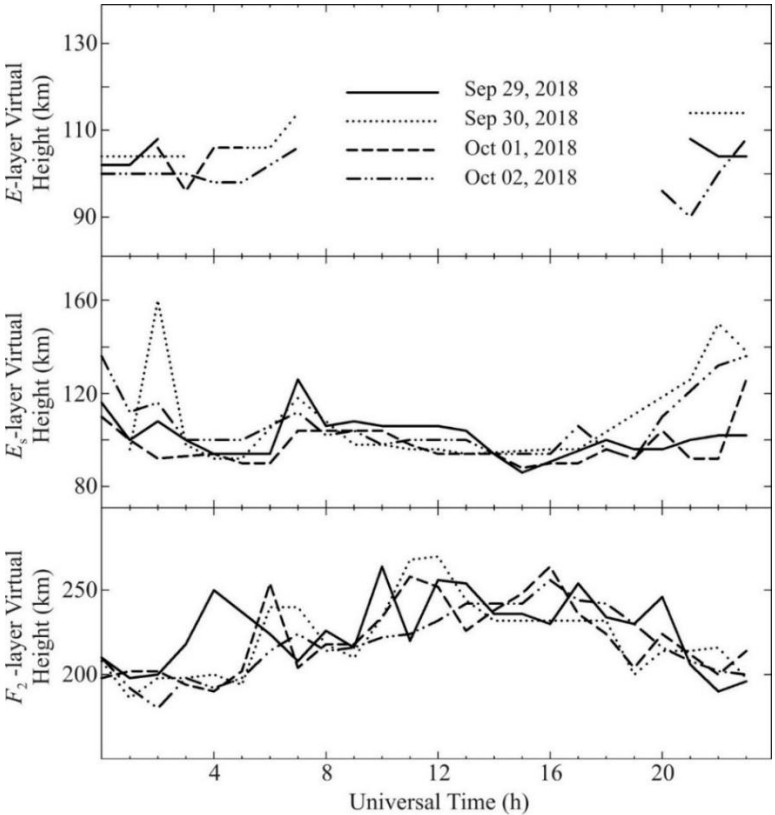

**Figure 4: Universal time variations in ionogram parameters determined with an update rate of one measurement per 1 h: virtual**
**heights $h'_E$, $h'_{Es}$, and $h'_{F2}$ for 29 September–2 October 2018 period.**

**6.1 Hwaseong to Harbin radio-wave propagation path**

This transmitter operating at 6,015 kHz is located in the Republic of Korea at a great-circle range, $R$, of ~950 km from the
receiver; it was switched off from 00:00 UT to 03:30 UT.

On 29 and 30 September 2018, the Doppler shift was observed to be less than ±(0.2–0.3) Hz (Figure 6). From 09:00
UT to 15:00 UT, the Doppler spectra showed diffuseness, and the Doppler shift exhibited quasi-sinusoidal variations with a
~20–30 min period, $T$, and a ~0.1–0.2 Hz amplitude, $f_{Da}$, respectively.

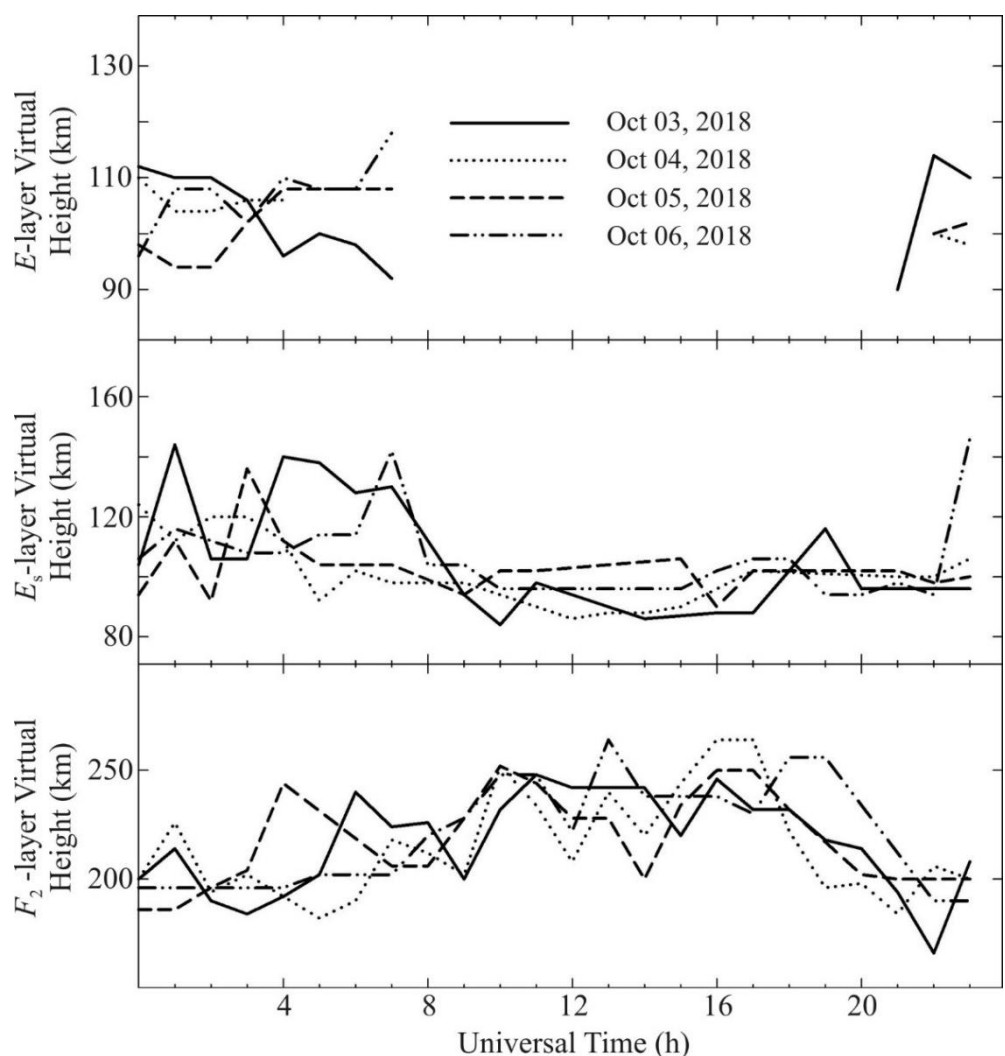

Figure 4: Continued for the 2–6 October 2018 period.

Doppler spectra broadening was absent during the interval from 1 to 6 October 2018. On 1 October 2018, $f_{Da} \approx 0.3$–0.5 Hz, and $T \approx 20$–120 min.

On 2 October 2018, the 20–80 min period amplitude of Doppler shift did not exceed 0.1–0.3 Hz. A single 140 min period wavelet in $f_{Da}$ up to 0.5 Hz originated on 3 October 2018. On 4 October 2018, the amplitude of Doppler shift did not exceed 0.2 Hz, while observable quasi-sinusoidal processes were practically absent. On 5 and 6 October 2018, the Doppler shift exhibited sporadic increases up to 0.5–0.6 Hz and decreases down to –(0.5–0.7) Hz; over the rest of the time, $f_{Da} \approx 0.1$ Hz, while $T \approx 20$–30 min.

The amplitudes of individual variations in the signal strengths did not exceed 10–15 dBV.

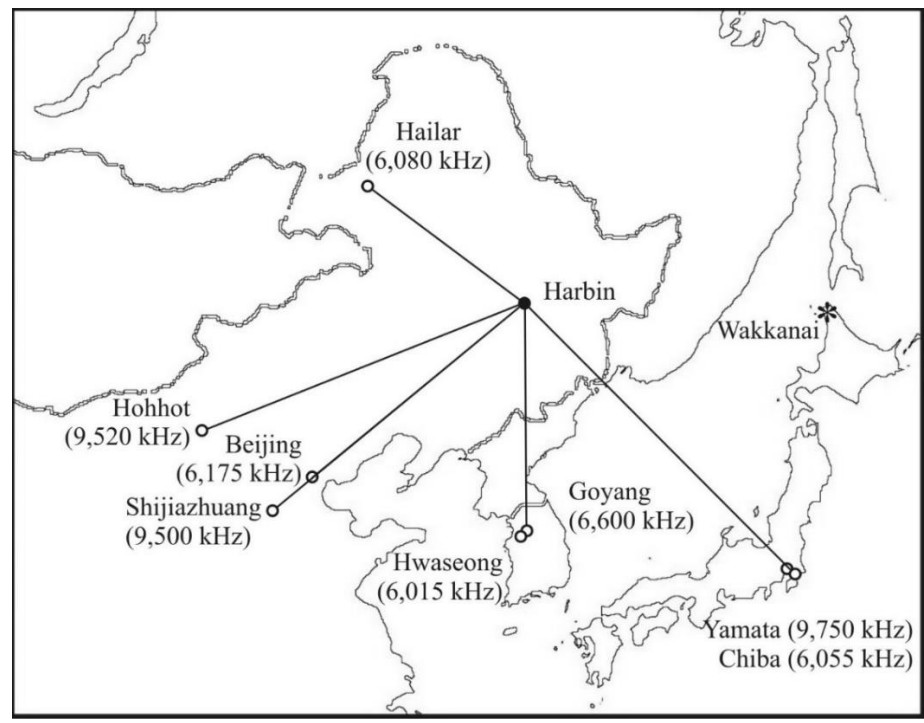

**Figure 5: Layout of the propagation paths used for monitoring dynamic processes operating in the ionosphere.**

Table 3. Basic information on radio-wave paths. The data are retrieved from https://fmscan.org/index.php.

| Frequency (kHz) | Transmitter coordinates | Location (Country) | Distance to Harbin (km) | Path midpoint coordinates |
|---|---|---|---|---|
| 6,015 | 37.21°N, 126.78°E | Hwaseong (Korea) | 950 | 41.50°N, 126.73°E |
| 6,055 | 35.47°N, 140.21°E | Chiba/Nagara (Japan) | 1,610 | 40.63°N, 133.45°E |
| 6,080 | 49.18°N, 119.72°E | Hailar/Nanmen (PRC) | 645 | 47.48°N, 123.2°E |
| 6,175 | 39.75°N, 116.81°E | Beijing (PRC) | 1,050 | 42.77°N, 121.75°E |
| 6,600 | 37.60°N, 126.85°E | Goyang (Korea) | 910 | 41.69°N, 126.77°E |
| 9,500 | 38.47°N, 114.13°E | Shijiazhuang (PRC) | 1,310 | 42.13°N, 120.41°E |
| 9,520 | 40.72°N, 111.55°E | Hohhot (PRC) | 1,340 | 43.25°N, 119.12°E |
| 9,750 | 36.17°N, 139.82°E | Yamata (Japan) | 1,570 | 40.98°N, 133.25°E |

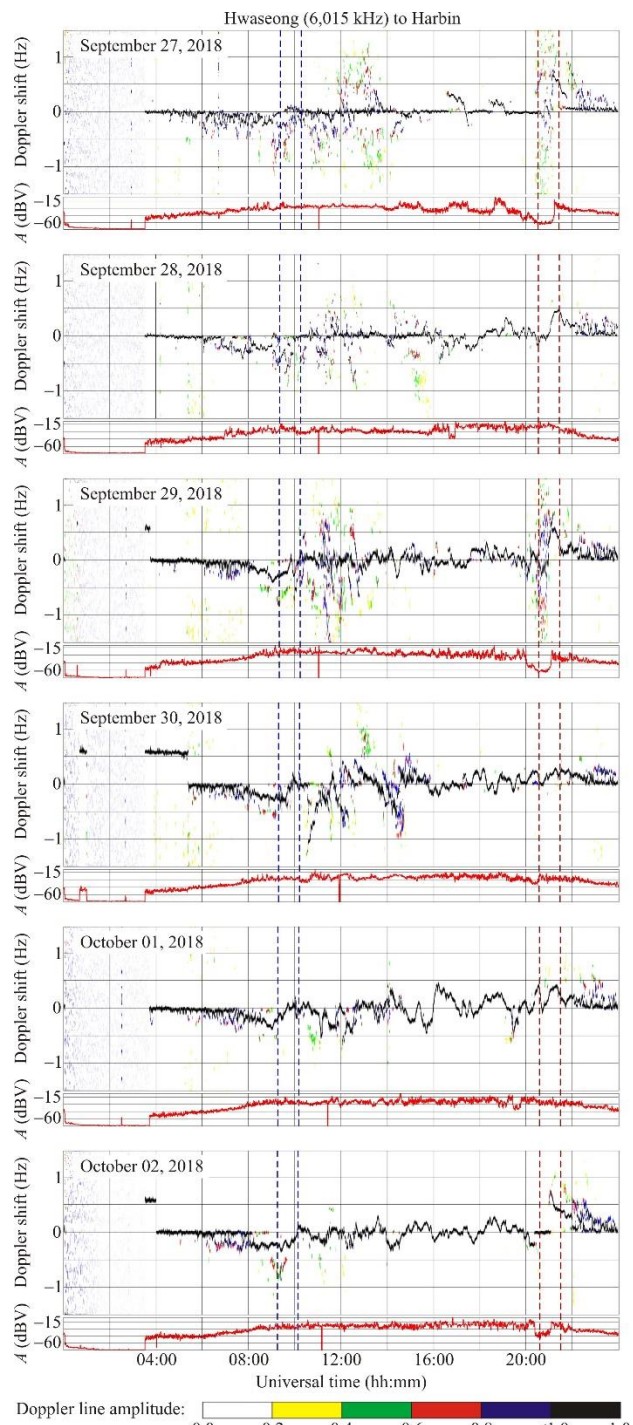

**Figure 6: Universal time variations in the Doppler spectra and relative signal amplitude, *A*, along the Hwaseong to Harbin propagation path for the 27 September – 2 October 2018 period. Vertical dashed lines designate instances of sunrise (two right red lines) and sunset (two left blue lines) at the ground and at 100 km altitude. The signal amplitude, *A*, at the receiver output in decibels, dBV (relative to 1 V), is shown below the Doppler spectra in each panel.**

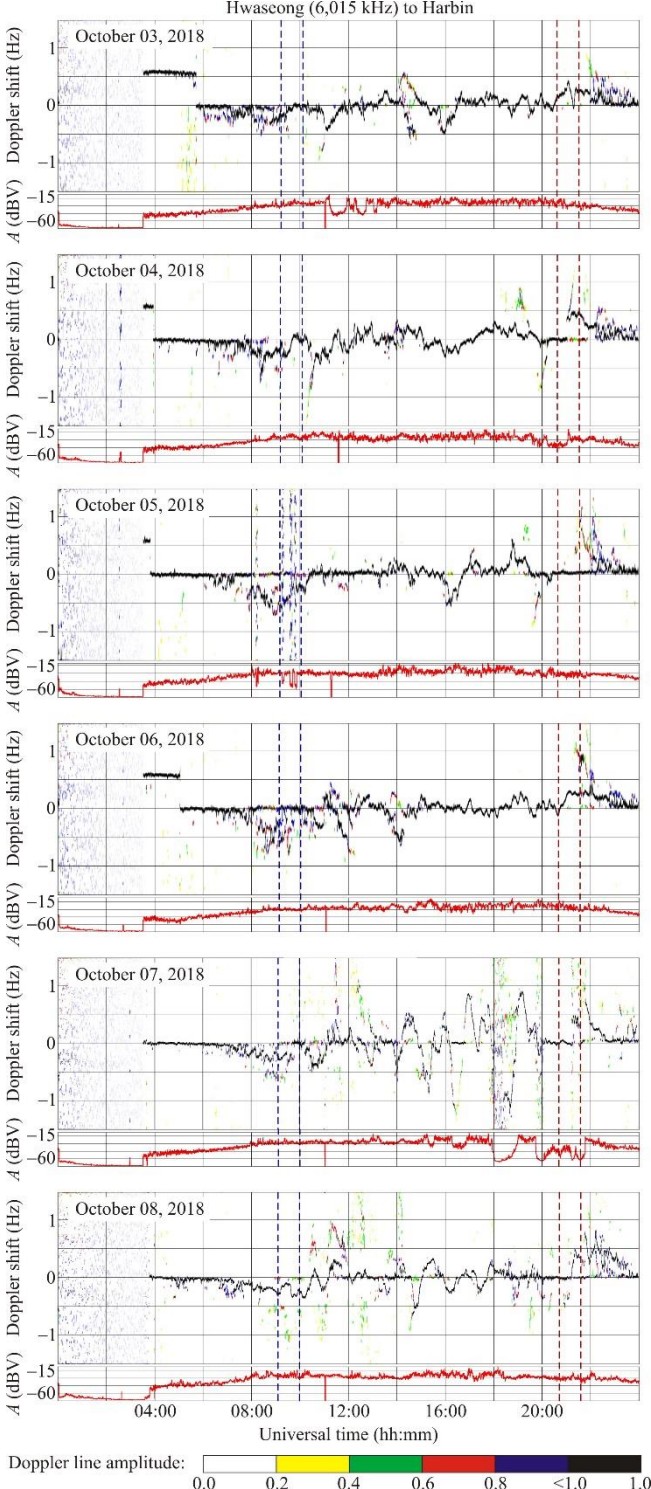

**Figure 6: Continued for the 3–8 October 2018 period.**

### 6.2 Chiba/Nagara to Harbin radio-wave propagation path

The radio station operating at 6,055 kHz is located in Japan at a great-circle range, $R$, of ~1,610 km; it was switched off from 15:00 UT to 22:00 UT.

Figure 7 shows that the Doppler spectra exhibit significant, up to ±1.5 Hz, broadening and such a diffuseness that the main ray is practically not distinguishable during the 27–30 September 2018 period. On 1 October 2018, the Doppler shift shows ~60 min period, $T$, quasi-sinusoidal variations with an ~0.3–0.4 Hz amplitude, whereas the signal amplitude, $A(t)$, exhibits ~30 min period, $T$, variations with a 5 dBV amplitude. On 5 and 6 October 2018, quasi–sinusoidal variations are also noted in the Doppler spectra, with ~0.4–0.6 Hz amplitudes, $f_{Da}$, and with ~60 min and 120 min periods, $T$. On 6 October 2018, the signal amplitude exhibits ~30 min and ~60 min period, $T$, quasi-sinusoidal variations with a ~5 dBV amplitude.

### 6.3 Hailar to Harbin radio-wave propagation path

This transmitter operating at 6,080 kHz is located in the PRC at a great-circle range, $R$, of 646 km; the transmissions were absent from 02:30 UT to 09:30 UT, whereas the observations of ionospheric dynamics were made impossible during the 14:30–20:00 UT period.

The variations in the Doppler spectra and the Doppler shift during sunlit hours on the reference days and on 1–2 October 2018 and 5 October 2017 were practically the same (Figure 8).

### 6.4 Beijing to Harbin radio-wave propagation path

This radio station operated at 6,175 kHz in the PRC at a great-circle range, $R$, of ~1,050 km from the receiver. The transmitter was switched off during 00:00 UT to 09:00 UT and from 18:00 UT to 20:00 UT periods.

On 29 September 2018, as well as on the next day, the Doppler shift showed small variations ~0.1 Hz (Figure 9), which exhibited increases of up to 0.3–0.5 Hz only over separate time intervals. On 1 October 2018, the Doppler shift exhibited quite ordered variations, with ~30 min and ~110 min period, $T$, oscillations, and amplitudes, $f_{Da}$, attaining 0.5 Hz. On 2 October 2018, the amplitude $f_{Da}$ decreased to 0.3 Hz, whereas the periods were observed to vary from 20 min to 110 min.

On 3, 4, and 6 October 2018, the Doppler shift showed quasi-sinusoidal variations with periods, $T$, in the 20 min to 90 min range and with 0.1–0.2 Hz amplitudes, $f_{Da}$.

After 14:00 UT on 5 October 2018, the Doppler shift amplitude, $f_{Da}$, was observed to increase to 0.2–0.4 Hz and to exhibit periods, $T$, in the range from 20 min to 80 min.

The signal amplitude exhibited temporal variability within the 10 dBV limits.

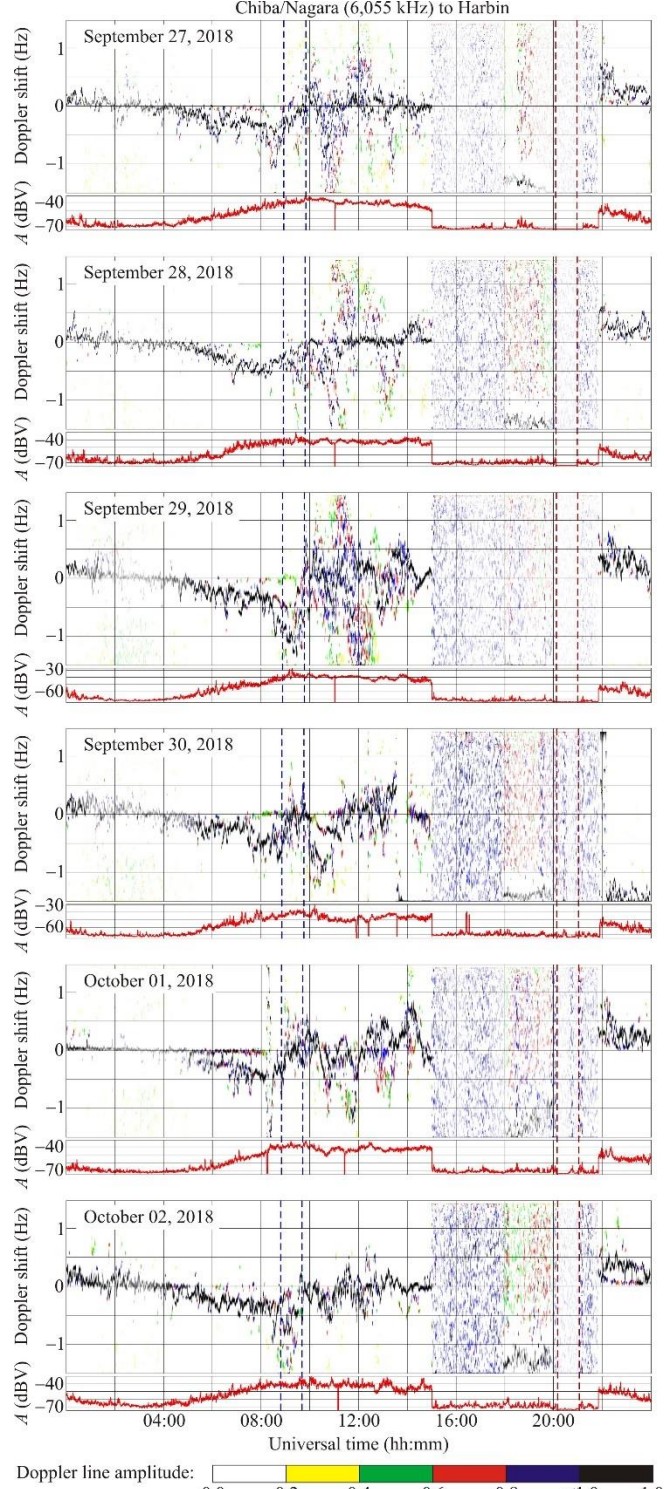

**Figure 7: The same as in Figure 6 but for the Chiba/Nagara to Harbin radio-wave propagation path at 6,055 kHz for the 27**
**September – 2 October 2018 period.**

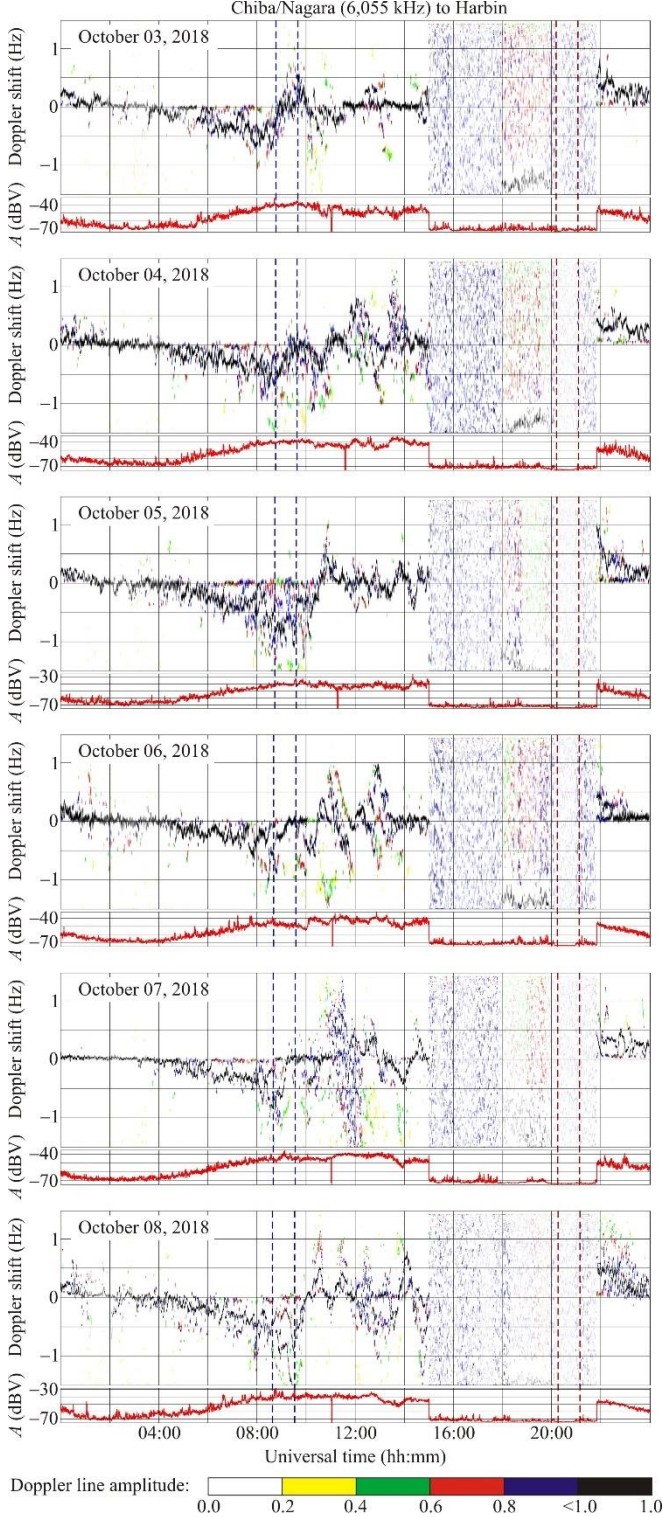

Figure 7: Continued for the 3–8 October 2018 period.

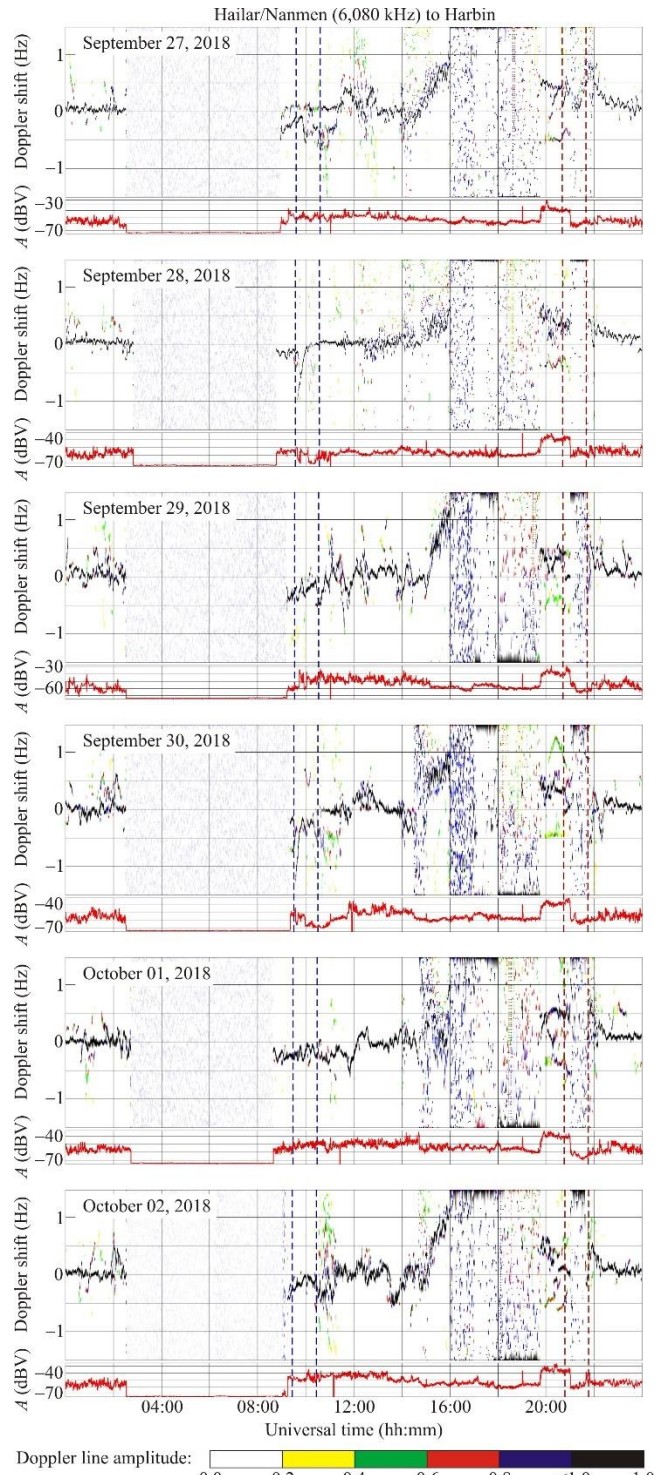

**Figure 8: The same as in Figure 6 but for the Hailar to Harbin radio-wave propagation path at 6,080 kHz for the 27 September – 2 October 2018 period.**

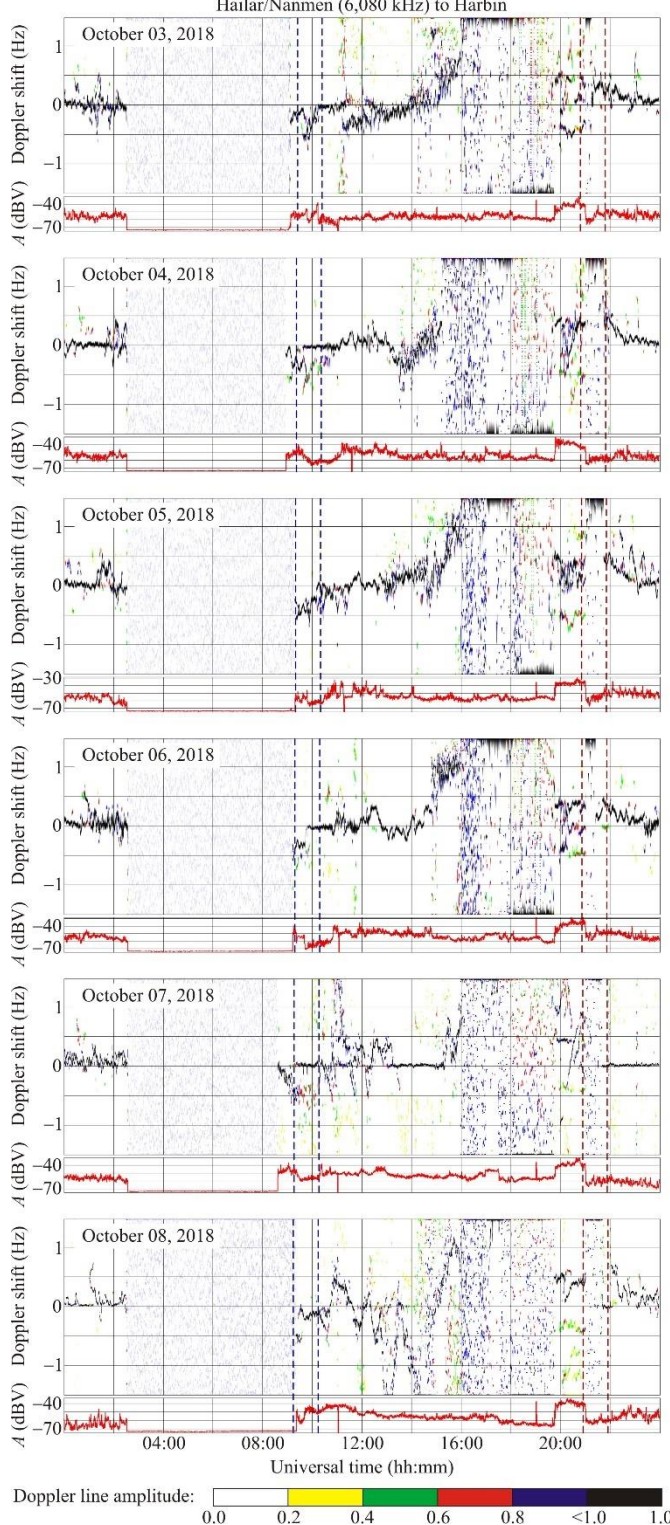

**Figure 8: Continued for the 3–8 October 2018 period.**

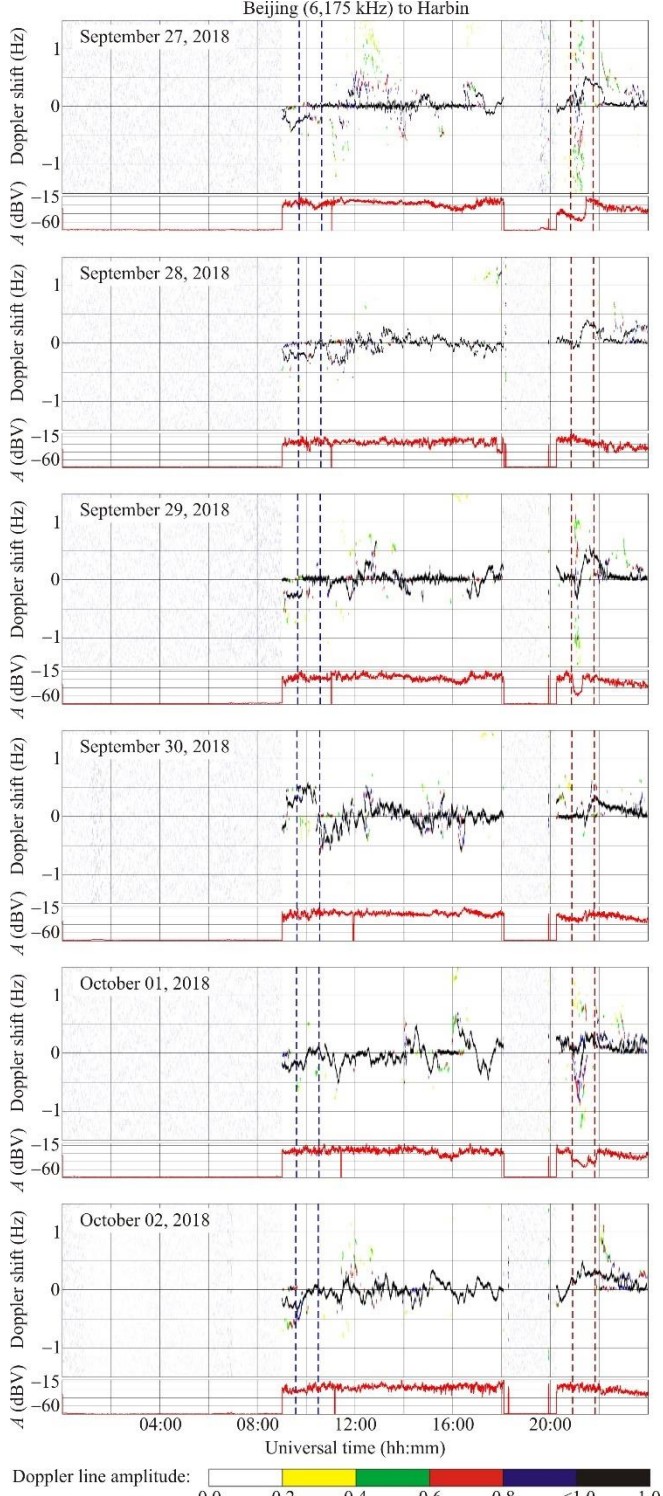

**Figure 9: The same as in Figure 6 but for the Beijing to Harbin radio-wave propagation path at 6,175 kHz**
**for the 27 September – 2 October 2018 period.**

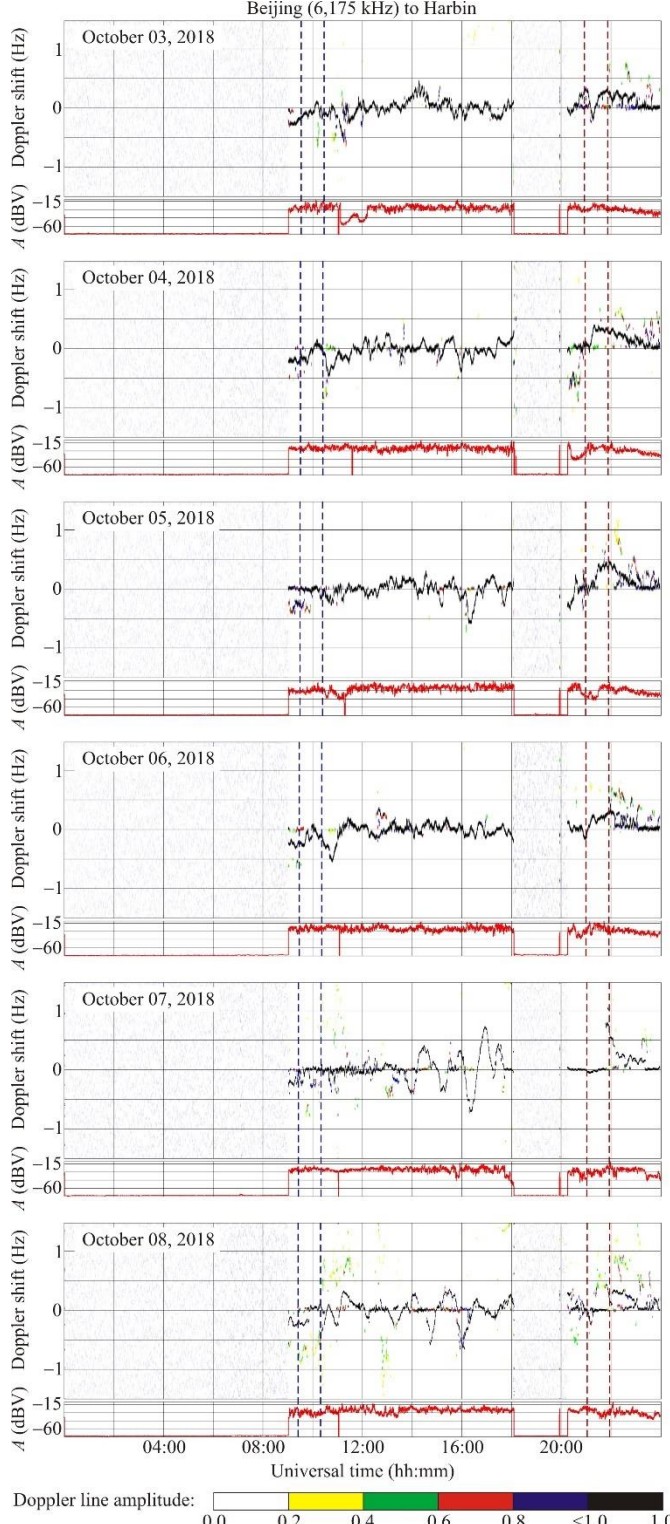

**Figure 9: Continued for the 3–8 October 2018 period.**

## 6.5 Goyang to Harbin radio-wave propagation path

This transmitter operating at 6,600 kHz is located in the Republic of Korea at a great-circle range of ~910 km from the receiver; it was switched off from 00:00 UT to 05:00 UT and from 22:20 UT to 24:00 UT.

On 29 September 2018, the Doppler shift showed fluctuations within the ±(0.2–0.3) Hz limits (Figure 10). Over the next day, the Doppler spectrum broadening was observed to occur from 12:00 UT to 16:00 UT whereas the Doppler shift exhibited quasi-sinusoidal ~20–24 min period, $T$, 0.1–0.2 Hz amplitude, $f_{Da}$, variations.

The Doppler ±0.6 Hz spectrum broadening was observed to occur on 1 October 2018, while the Doppler spectra exhibited ~20–120 min period, $T$, ~0.1–0.7 Hz amplitude, $f_{Da}$, variations; considerable, up to 20 dBV, variations were noted in the signal amplitude.

On 2 October 2018, the Doppler shift exhibited significant, ±(0.2–0.3) Hz, variations, with a quasi-period, $T$, of 24 min and amplitude, $f_{Da}$, of ~0.2 Hz. Considerable fluctuations in the Doppler spectra, the Doppler shift, and in the signal amplitude were noted on 3 October 2018. On 4 October 2018, from 14:00 UT to 20:00 UT, the Doppler shift showed changes within the –0.3 Hz to 0.3 Hz limits, the quasi-sinusoidal processes were expressed weakly, and the signal amplitude fluctuated wildly, by 30 dBV. On 5 October 2018, the variations in the Doppler shift did not exceed ±0.2 Hz, the fluctuations in the signal amplitude were also insignificant. The Doppler shift was observed to increase up to ±(0.3–0.5) Hz during the 6 October 2018 11:00–14:00 UT period, whereas from 15:00 UT to 18:00 UT, the quasi-sinusoidal oscillations in the Doppler shift were observed to occur with a ~20 min period, $T$, and ~0.1 Hz amplitude, $f_{Da}$, while quasi-sinusoidal variations in the signal amplitude, $A(t)$, were observed to occur with a ~55–80 min period, $T$, and ~7–8 dBV amplitude.

## 6.6 Shijiazhuang to Harbin radio-wave propagation path

This radio station operating at 9,500 kHz is located in the PRC at a great-circle range, $R$, of ~1,310 km from the receiver.

Figure 11 shows that the value of Doppler shift, $f_D(t)$, was close to zero on each night. The Doppler shift was observed to be negative, attaining a minimum of –(0.20–0.25) Hz, two to three hours before sunset at the ground. During the UT night of 29 September 2018, the signal amplitude was observed to fluctuate wildly within the 20 dBV limits and to be accompanied by fluctuations in the Doppler shift. A second ray shifted by –0.5 Hz with respect to the main ray was observed to appear during the 16:00–20:00 UT period. From 30 September 2018 through 6 October 2018 UT nights, the signal frequency approached the maximum usable frequency and ionospheric signal was about to penetrate the ionosphere, which resulted in a 10–20 dBV decrease in the signal amplitude, whereas the Doppler spectra became low informative. These circumstances have significantly hampered the search for the ionospheric response to the super typhoon action. Nevertheless, the Doppler shift exhibited considerable, attaining –1 Hz, variations on 1 October 2018. Significant variations in the Doppler shift were noted at the beginning of the 2 October 2018, UT night and after UT midnights on 3, 4, 5, and 6 October 2018; in particular, the ray shifted by –0.5 Hz was recorded.

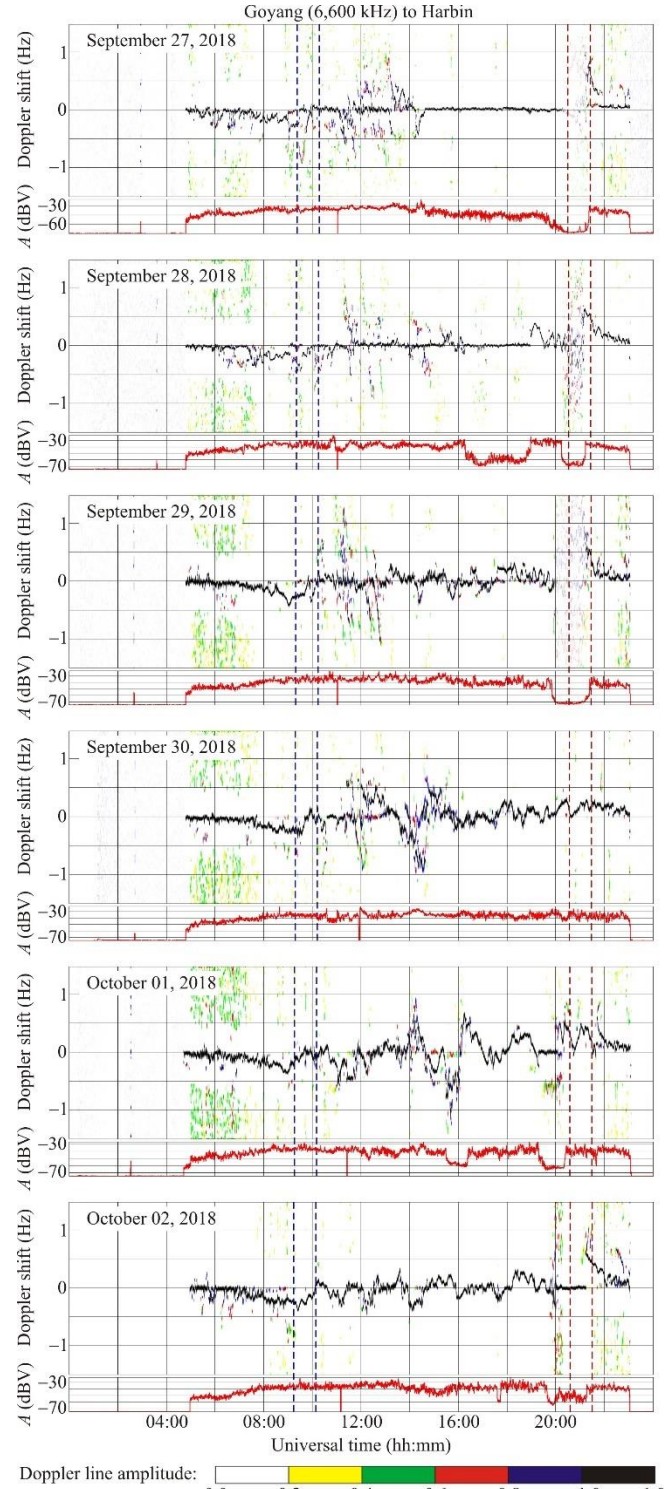

**Figure 10: The same as in Figure 6 but for the Goyang to Harbin radio-wave propagation path at 6,600 kHz for the 27 September**
**– 2 October 2018 period.**

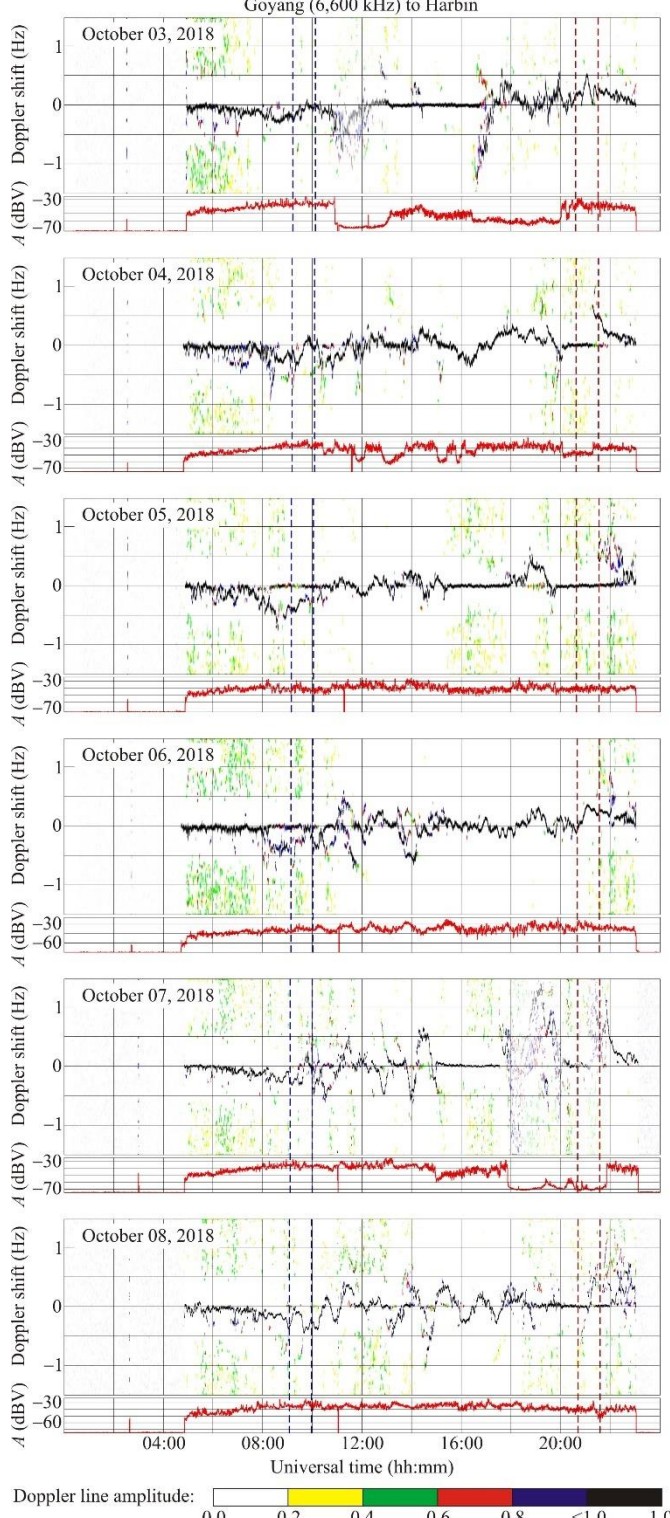

Goyang (6,600 kHz) to Harbin

Doppler line amplitude:

| 0.0 | 0.2 | 0.4 | 0.6 | 0.8 | <1.0 | 1.0 |

**Figure 10: Continued for the 3–8 October 2018 period.**

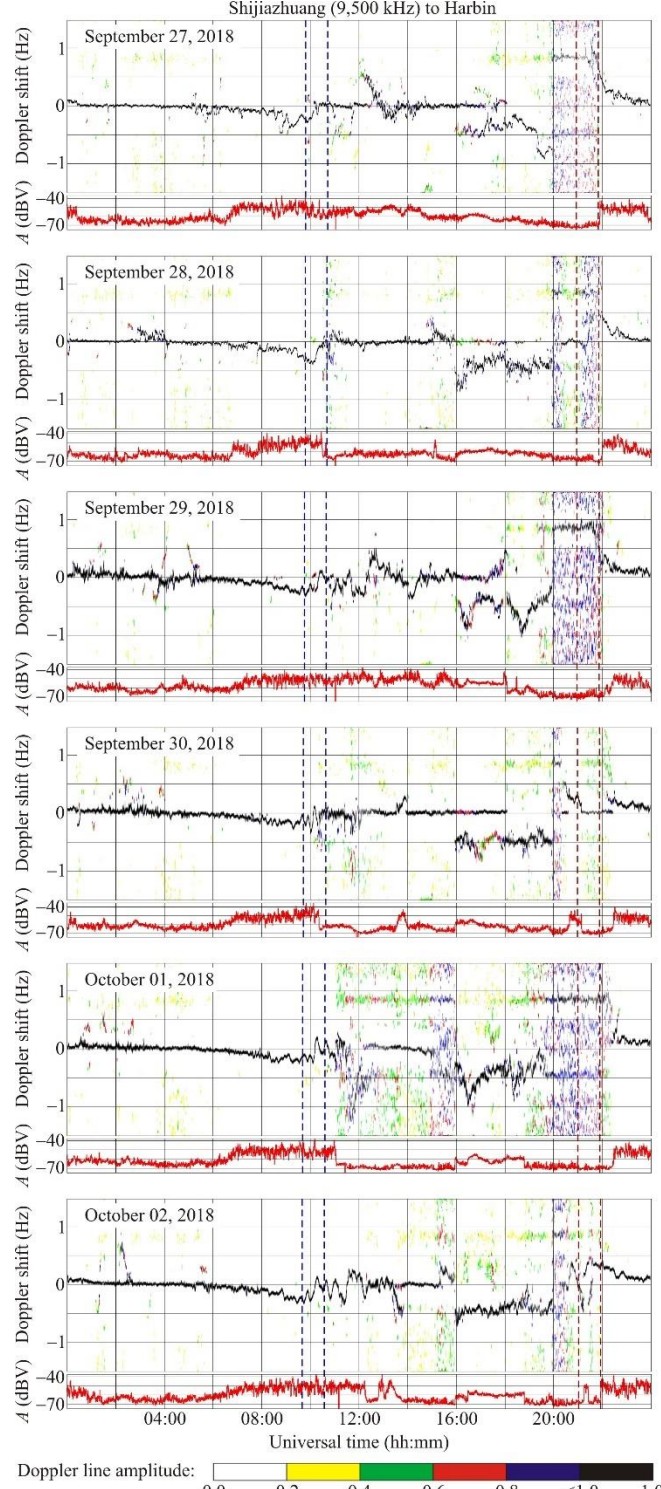

**Figure 11: The same as in Figure 6 but for the Shijiazhuang to Harbin radio-wave propagation path at 9,500 kHz for the 27 September – 2 October 2018 period.**

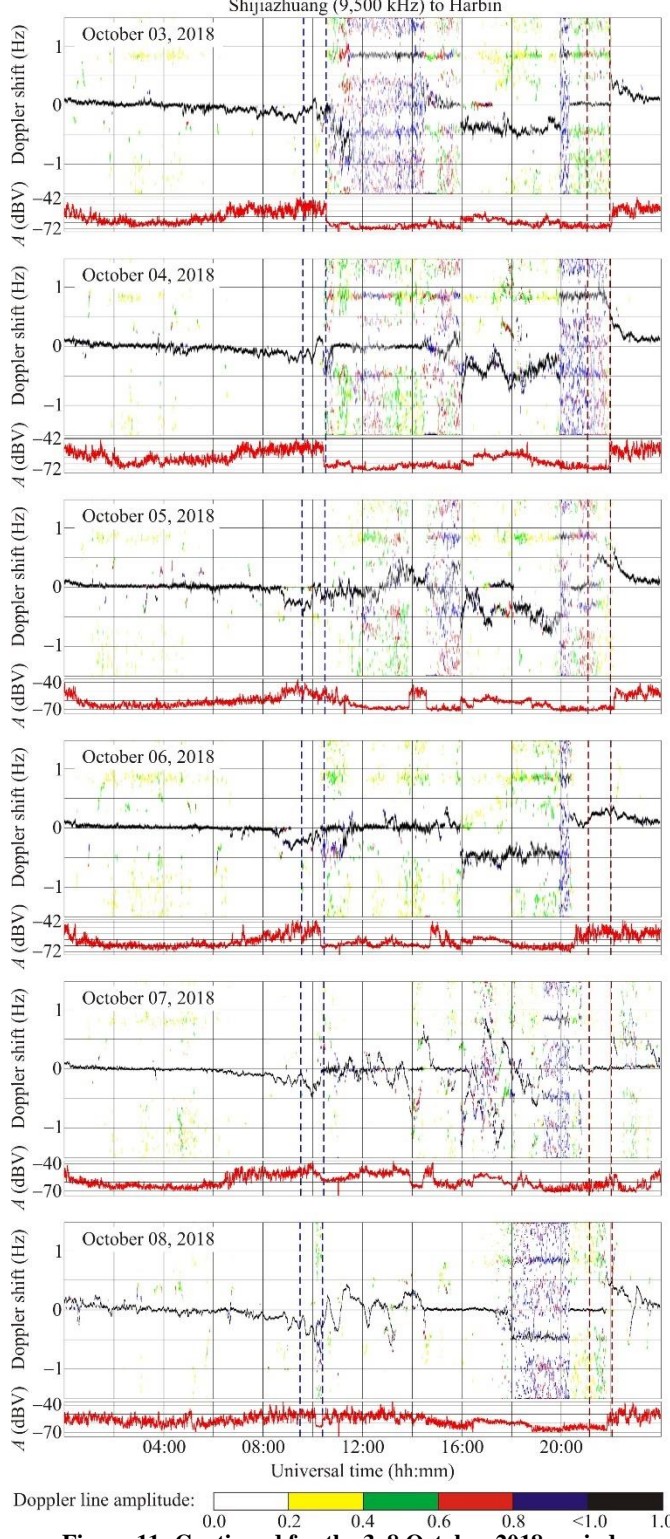

Figure 11: Continued for the 3–8 October 2018 period.

### 6.7 Hohhot to Harbin radio-wave propagation path

This transmitter operating at 9,520 kHz is located in the PRC at a great-circle range, $R$, of ~1,340 km; it was switched off from 16:00 UT to ~22:00 UT.

The frequency of this radio wave became greater than the maximum usable frequency and the radio wave penetrated the ionosphere during the second half of all nights (see Figure 12), the received signal was absent, and the observation of the ionospheric dynamics became impossible. The Doppler spectra exhibited substantial variations ($\pm0.5$ Hz) on 29 September 2018, from 12:00 UT to 16:00 UT. On the UT night of 30 September 2018, the reflection of radio waves took place from the sporadic $E$ layer, resulting in $f_D(t) \approx 0$ Hz, and during the UT night of 1 October 2018, the Doppler shift $f_D(t) \approx 0$ Hz as well. On the UT night of 2 October 2018, the Doppler shift showed changes from –0.3 Hz to 0.3 Hz, while the signal amplitude exhibited considerable variability, up to 20 dBV. During the UT nights of 3–6 October 2018, the measurements were ineffective, whereas $f_D(t) \approx 0$ Hz during sunlit hours.

### 6.8 Yamata to Harbin radio-wave propagation path

This radio station operating at 9,750 kHz is located in Japan at a great-circle range, $R$, of ~1,570 km. The transmitter was switch off from 16:00 UT to ~22:00 UT.

A characteristic feature of these observations is that two signals were received, the Doppler shift of which were shifted by 1 Hz from 29 September 2018 through 3 October 2018 and by 0.5 Hz from 4 October 2018 through 6 October 2018, as can be seen in Figure 13.

During all sunlit hours, the Doppler shift exhibited insignificant variations, whereas it became negative in the evening. The Doppler shift and the Doppler spectrum variations were observed to be significant (from –1 Hz to 1 Hz) during the UT nights of 29 and 30 September 2018. On 1 October 2018, the Doppler spectra exhibited diffuseness, while the signal amplitude $A(t)$ variability was observed to attain 30 dBV. During the UT night of 2 October 2018, the Doppler shift was observed to vary from –0.4 Hz to 0.4 Hz, while the variations in $A(t)$ also attained 30 dBV. The Doppler spectra and the Doppler shift showed insignificant temporal variability on 3 and 4 October 2018; at the same time the signal amplitude exhibited 20–30 dBV variations.

On 5 and 6 October 2018, the magnitude of the Doppler shift variations attained $\pm0.2$ Hz, while the signal amplitude exhibited considerable changes in amplitude, up to 30 dBV.

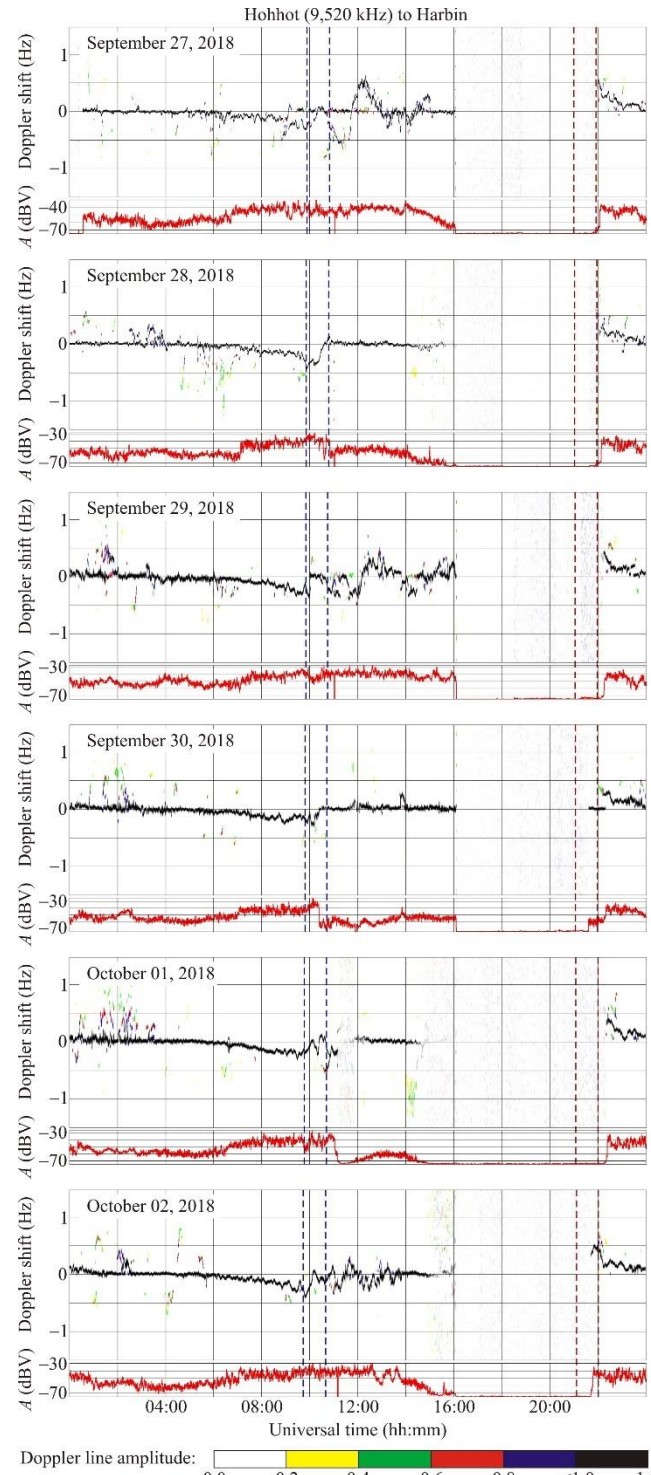

**Figure 12: The same as in Figure 6 but for the Hohhot to Harbin radio-wave propagation path at 9,520 kHz for the 27 September**
**– 2 October 2018 period.**

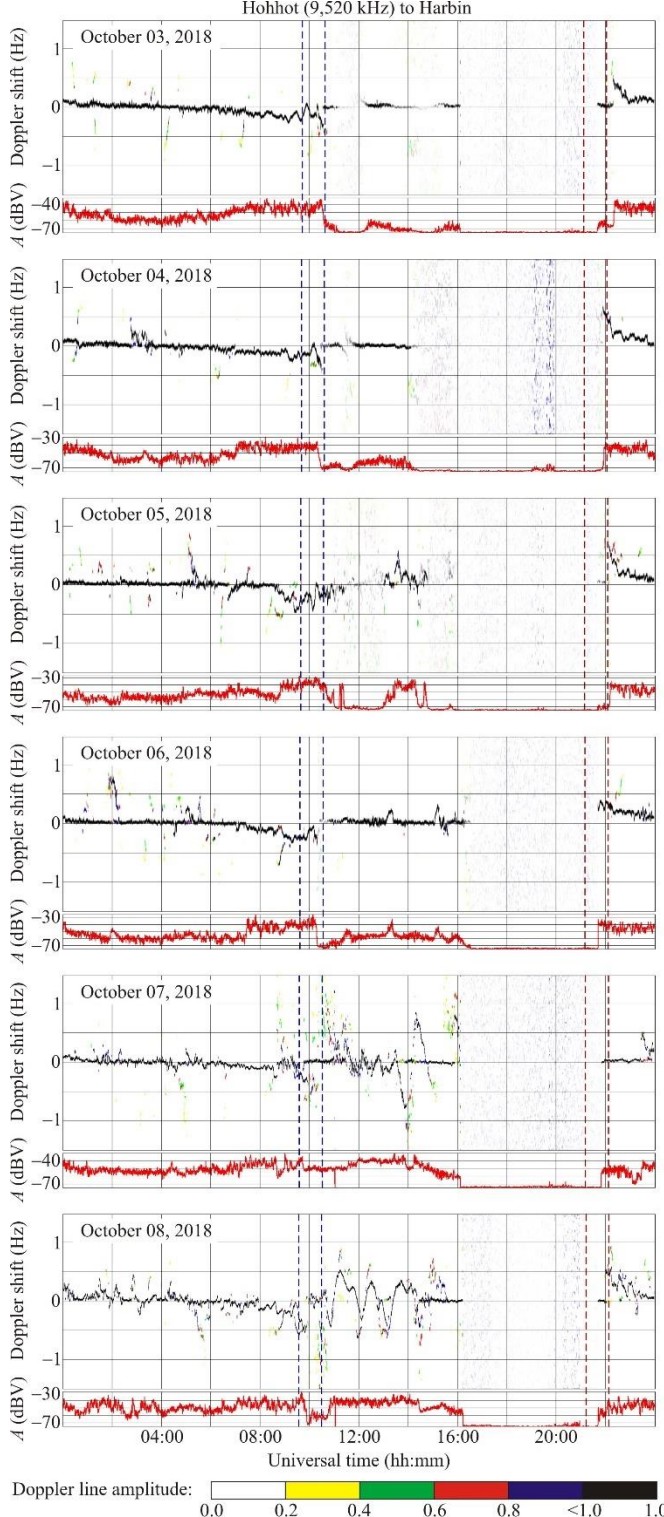

**Figure 12: Continued for the 3–8 October 2018 period.**

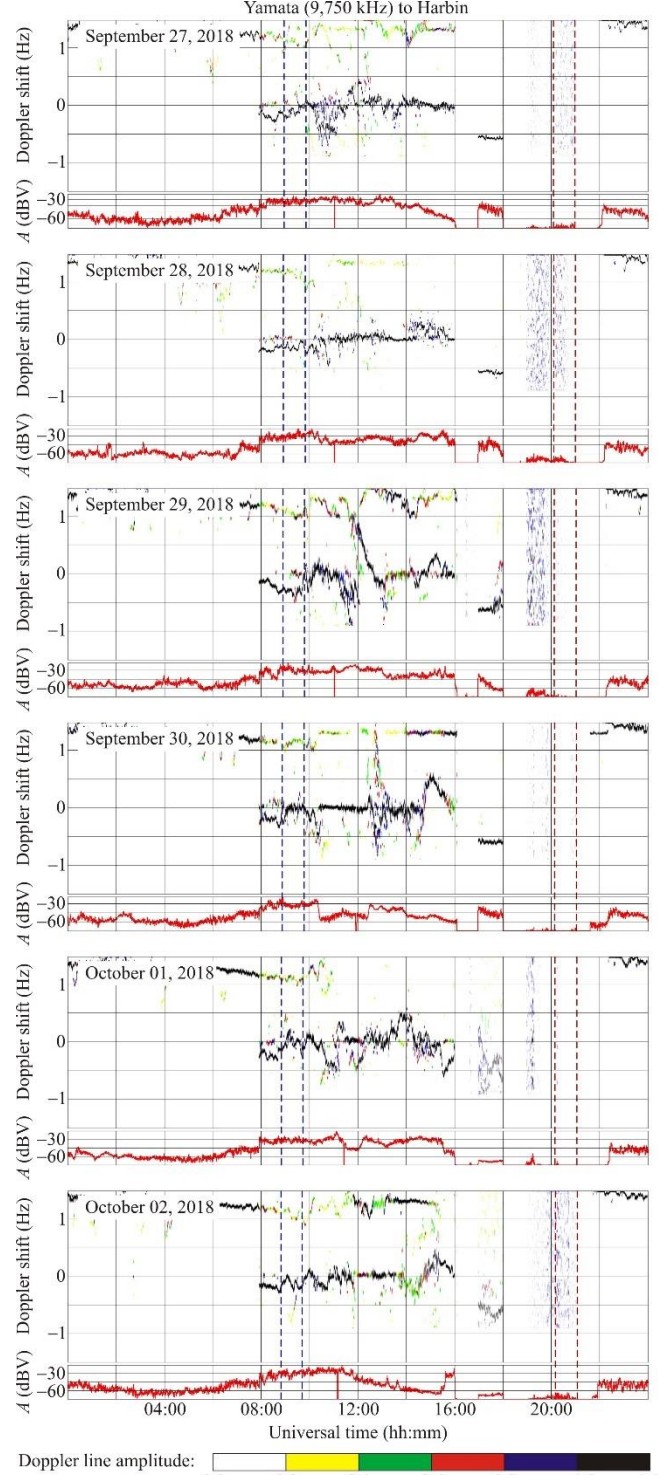

**Figure 13: The same as in Figure 6 but for the Yamata to Harbin radio-wave propagation path at 9,750 kHz for the 27 September – 2 October 2018 period.**

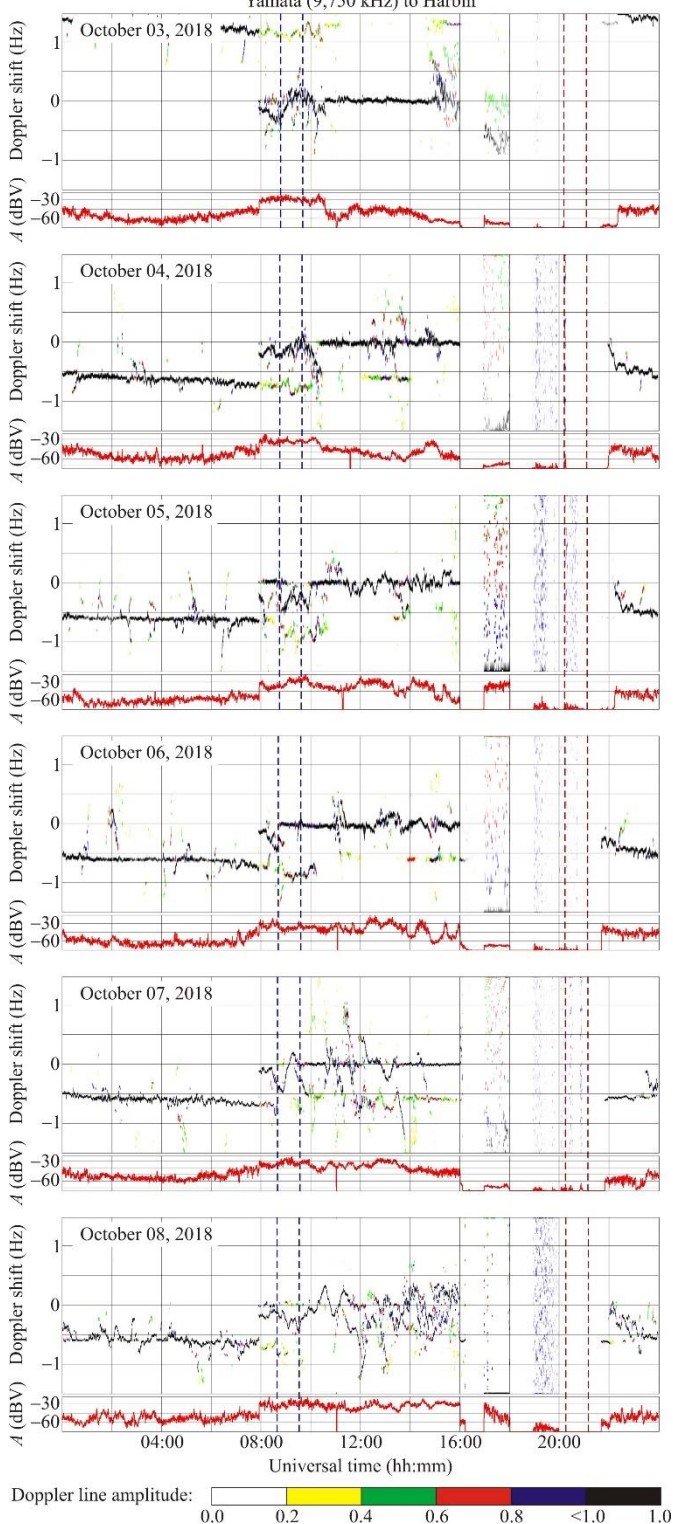

Figure 13: Continued for the 3–8 October 2018 period.

## 7 Discussion

### 7.1 Ionospheric effects from the super typhoon

The Doppler spectra and the Doppler shift observed during sunlit hours exhibited insignificant temporal variability through the course of the typhoon action, since the radio waves in the ~6–10 MHz band were reflected from the ionospheric $E$ region or from the sporadic $E$. At night, the radio waves were reflected from the ionospheric $F$ region, and the Doppler shift was observed to reach maximum values of up to 0.5–1 Hz. The temporal variations in the Doppler shift were also generated by the movement of the solar terminator.

The diurnal variations in the signal amplitude $A(t)$ were observed to attain 30 dBV, while $A(t)$ at night was three orders of magnitude greater than $A(t)$ during the daytime, which is due to the disappearance of the absorbing $D$ region.

In order to find out that the observed Doppler shift variations are associated with the typhoon, the Doppler variations were low-pass filtered, and the Doppler variations, with periods of greater than 40 min, were found to occur during the 2 October 2018 10:00–14:00 UT period, along all propagation paths. A characteristic feature, a fading, which could be traced in all temporal dependences of the identified Doppler variations, was selected for analyzing. The UT moments, $t^*$, when this feature arrived at each propagation path midpoint are presented in Table 4. At 12:00 UT on 2 October 2018, the typhoon center was located at (18.9° N, 131.2° E) at the distances $D$ from the propagation path midpoints, with the midpoint of the 9.750 MHz propagation path being closest (2,492 km) to the typhoon center, while other midpoints were found to be at $(2{,}492 + \Delta D)$ km ranges, where the characteristic feature arrived with time delays of $\Delta t$ with respect to the arrival time at the 9.750 MHz midpoint. As can be seen in Table 4, the $\Delta D$ and $\Delta t$ yield the values of the apparent speeds, $v$, quite close to each other. These estimates testify to the adequacy of the assumption that the propagation of the disturbances from the typhoon is the cause of the observed Doppler shift variations. The mean value of the speed of the strongest, 60 – 70 min, period component, estimated to be $205\pm6$ m s$^{-1}$, corresponds to a TID with wavelength equal to approximately 800 km. Taking a look at the Kong-Rey trajectory in Figure 1a, one can notice that the TIDs traveled northwestward in this case, contrary to the southwestward direction observed in this area of the world in the climatological study by Shiokawa et al. (2003).

Table 4. Distances $D$ over which TIDs traveled at apparent speeds $v$ and arrived at the propagation pass midpoints from the center of typhoon Kong-Rey with relative time delays $\Delta t$ at the UT moments $t^*$.

| $f$ (MHz) | 6.015 | 6.055 | 6.080 | 6.175 | 6.600 | 9.500 | 9.520 | 9.750 |
|---|---|---|---|---|---|---|---|---|
| $D$ (km) | 2,574 | 2,454 | 3,296 | 2,826 | 2,595 | 2,803 | 2,963 | 2,492 |
| $\Delta D$ (km) | 120 | 38 | 842 | 372 | 141 | 349 | 509 | 0 |
| $t^*$ (UT) | 11:00 | 10:53 | 12:00 | 11:20 | 11:03 | 11:15 | 11:25 | 10:50 |
| $\Delta t$ (min) | 10 | 3 | 70 | 30 | 12 | 28 | 38 | – |
| $v$ (m s$^{-1}$) | 200 | 210 | 200 | 205 | 195 | 205 | 220 | – |

The ionospheric effects from super typhoon Kong-Rey are discussed further below. Super typhoon Kong-Rey's power gained a maximum value during the second half of 1 October 2018 and, consequently, the Doppler shift and the Doppler spectra showed the greatest variations during the UT night of 1 October 2018, despite the propagation path midpoints were located ~2,800–3,300 km away from the super typhoon. As should be expected, the greatest effects were observed to occur at the propagation path midpoints located closest to the typhoon, i.e., in the signals transmitted from the radio stations at Chiba/Nagara, Goyang, Yamata (Japan) and Hwaseong (Republic of Korea). At the same time, the ionospheric effects from super typhoon Kong-Rey were absent along the Hohhot to Harbin radio-wave propagation path located at the farthest range to the typhoon on 1 October 2018.

On 2 October 2018, super typhoon Kong-Rey moved closer to the propagation path midpoints by only ~600 km; however, its power reduced by a factor of approximately 2 during the night. As a result, the ionospheric response to the typhoon action reduced noticeably, and the ionospheric effects were either weak or absent during the daytime of 2 October 2018, as well as during sunlit hours on 3 and 4 October 2018.

Figure 1a shows that a surge in the typhoon's power (marked in red) appeared during the time interval between the noon of 5 and 6 October 2018, when the typhoon and the propagation path midpoints were at ~1,000–1,500 km distances apart. Consequently, increases in the amplitude of Doppler shift and, partly, in the signal amplitude variations were observed to occur on 5 October 2018 as well as along a number of propagation paths on 6 October 2018 despite the typhoon's power was reduced by a factor of ~3, as compared to that observed on 1 October 2018.

## 7.2 Wavelike disturbances

Wavelike disturbances in the ionosphere can be seen even on 29 and 30 September 2018, whereas on 1–2 and 5–6 October 2018, a noticeably increase (by a factor of ~2–3) in the amplitude of Doppler shift and in the signal amplitude variations, in a number of cases, were observed to occur. Based on the periods (from 20 min to 120 min), the wavelike disturbances in the ionosphere are caused by atmospheric gravity waves (Gossard and Hooke, 1975).

The basic parameters of wave disturbances associated with the action of typhon Kong-Rey are presented in Table 5.

Table 5.

Basic parameters of wave disturbances in October 2018.

| Radio-wave propagation path | Date | | | |
|---|---|---|---|---|
| | 1 October | 2 October | 5 October | 6 October |
| Hwaseong to Harbin | $T = 120$; 24 min $f_{Da} = 0.4$; 0.1 Hz | $T = 120$; 24 min $f_{Da} = 0.25$; 0.1 Hz | $T = 100–110$; 15 min $f_{Da} = 0.1–0.2$; 0.05 Hz | $T = 120$; 20 min $f_{Da} = 0.3$; 0.05 Hz |
| Chiba/Nagara to Harbin | $T = 60–80$ min $f_{Da} = 0.4$ Hz | $T = 20–30$ min $f_{Da} = 0.2–0.3$ Hz | $T = 30–40$ min $f_{Da} = 0.3$ Hz | $T = 100$ min $f_{Da} = 0.3$ Hz |
| Hailar/Nanmen to Harbin | $T = 80$; 15 min $f_{Da} = 0.4$; 0.05 Hz | $T = 40–50$ min $f_{Da} = 0.2–0.3$ Hz | $T = 40$; 20 min $f_{Da} = 0.2–0.3$ Hz | $T = 40–60$ min $f_{Da} = 0.1–0.2$ Hz |
| Beijing to Harbin | $T = 30–40$; 20–24 min $f_{Da} = 0.2$; 0.1 Hz | $T = 60$; 20–25 min $f_{Da} = 0.2$; 0.1 Hz | $T = 40–60$; 20 min $f_{Da} = 0.2$; 0.05 Hz | $T = 80$; 20–30 min $f_{Da} = 0.2$ Hz |
| Goyang to Harbin | $T = 120$; 30–40 min $f_{Da} = 0.3$; 0.1 Hz | $T = 40–60$; 30 min $f_{Da} = 0.2$; 0.1 Hz | $T = 80–90$ min $f_{Da} = 0.2$ Hz | $T = 100–120$; 20 min $f_{Da} = 0.4$; 0.1 Hz |
| Shijiazhuang to Harbin | $T = 120$ min $f_{Da} = 0.3$ Hz | $T = 60$ min $f_{Da} = 0.3$ Hz | $T = 120$ min $f_{Da} = 0.3$ Hz | $T = 120$ min $f_{Da} = 0.1$ Hz |
| Hohhot to Harbin | – | $T = 120$; 40–50 min $f_{Da} = 0.2–0.3$ Hz | – | – |
| Yamata to Harbin | $T = 65–80$ min $f_{Da} = 0.3–0.4$ Hz | $T = 80$ min $f_{Da} = 0.2$ Hz | $T = 40$; 25–30 min $f_{Da} = 0.2$; 0.1 Hz | $T = 70$ min $f_{Da} = 0.2$ Hz |

Given known $f_{Da}$ and $T$, the amplitude, $\delta_{Na}$, of quasi-sinusoidal variations in the electron density can be estimated on a relative scale. To do this, one can use the following equation (Chernogor et al., 2020; Guo et al., 2020):

$$\delta_{Na} = \frac{K}{4\pi} \frac{cT}{L} \frac{f_{Da}}{f}$$ (1)

where

$$K = \frac{1 + \cos\theta}{2\left(1 + 2\xi\tan^2\theta\right)\cos^2\theta} \; ; \; \xi = \frac{z_r - z_0}{r_0} \; ; \; \tan\theta = \frac{R}{2z_r} ,$$ (2)

$c$ is the speed of light, $\theta$ is the angle of incidence with respect to the vertical at the basis of the ionosphere, $z_0$ is the altitude

of the beginning of the layer giving a contribution to the Doppler shift, $z_r$ is the altitude of reflection, $r_0$ is the mean Earth's

radius, $L$ is the thickness of the atmospheric region giving a contribution to the Doppler shift.

Substituting $T \approx 20$ min and $f_{Da} \approx 0.1$ Hz in (1) and taking into account (2) yield $\delta_{Na} \approx 0.4\%$. If $T \approx 30$ min and

$f_{Da} \approx 0.2$ Hz, then $\delta_{Na} \approx 1.2\%$. Also, $T \approx 60$ min and $f_{Da} \approx 0.5$ Hz give $\delta_{Na} \approx 6\%$. Thus, the super typhoon action in the

ionosphere leads to an increase in the amplitude of variations in the electron density, depending on the period of quasi-

sinusoidal disturbances, by a fraction to several per cent.

In addition, the amplitude of quasi-sinusoidal variations in the Doppler shift is observed to increase along many

propagation paths under the joint action of the super typhoon and dusk terminator passing by. Consequently, the synergistic

action of the dusk terminator and typhoons takes place in the ionosphere. An effect analogous to the one mentioned above

was observed earlier by Edemsky and Yasyukevich (2018) who made use of GPS technology for probing wave disturbances.

During the dawn terminator, such an effect is not reliably observed.

**7.3 Comparison of ionospheric effects from typhoons**

A multifrequency multiple path coherent software defined radio system developed at the Harbin Engineering University has

been in routine use for several years for determining variations in ionospheric parameters and in radio wave characteristics in

the 5–10 MHz band, which accompanied the movement of super typhoon Hagibis (Chernogor et al., 2021), Ling-Ling and

Faxai (Chernogor et al., 2022), Lekima (Zheng et al., 2022), Kong-Rey, etc. The response of the ionosphere to typhoon

action has been shown to be dependent not only on the parameters of typhoons, but also on the state of atmospheric and

space weather, local time, and on other geophysical parameters. Not only are common manifestations in the response found,

but also individual manifestations that are characteristic of a particular super typhoon. The common manifestations include:

(1) the aperiodic (chaotic) character of the ionospheric response; (2) the magnitude of the response shows an apparent

abatement with increasing distance between the typhoon and the propagation path midpoints; (3) the response exhibits a

maximum with distance (between the typhoon and the propagation path midpoint) approaching a minimum; (4) Doppler shift

spectrum broadening up to ±1 Hz due to an increase in the number of rays; (5) the occurrence of quasi-sinusoidal variations

in the Doppler shift with amplitudes of ~0.1–0.5 Hz and periods of 2–5 min and 10–100 min; (6) the generation or

enhancement of infrasound (periods, $T$, of ~2–5 min) and atmospheric gravity waves (periods, $T$, of ~10–100 min); (7)

disturbances of the electron density amplitudes in these wave fields attaining ~1 % and ~10%, respectively, and greater; (8)

aperiodic perturbations (for the most part, increases) in the electron density that could attain a few tens of per cent.

The instrument created by the authors of this paper permitted the confirmation of only one mechanism of affecting

the ionosphere with a typhoon, i.e., the acoustic–atmospheric gravity waves. To reveal electromagnetic and electric

mechanisms, one should employ other instruments.

## 8 Conclusions

(1) The Harbin Engineering University multifrequency multiple path coherent software defined radio system for probing the

ionosphere at oblique incidence have been used to detect the ionospheric effects over the People's Republic of China during

the 27 September 2018 to 8 October 2018 period encompassing the super typhoon Kong-Rey event. The movement of the

super typhoon was accompanied by significant variations in radio wave characteristics in the 5–10 MHz band.

(2) The ionospheric response to the super typhoon action was clearly observed to occur on 1–2 October 2018 when

the typhoon was 2,800–3,300 km away from the propagation path midpoints and super typhoon Kong-Rey's energy gained a

maximum value, and on 5–6 October 2018 when the typhoon was 1,000–1,500 km away from the midpoints and its energy

decreased by a factor of approximately 3.

(3) The ionospheric effects are more pronounced along the nearest propagation paths, whereas no effect is detected

along the propagation path at the greatest distance from the typhoon.

(4) The super typhoon action on the ionosphere was accompanied by the generation or amplification of quasi-

sinusoidal variations in the Doppler shift by a factor of 2–3, as well as by noticeable variations in the signal amplitude. The

Doppler spectra were observed to broaden in a number of cases.

(5) The period of wave perturbations exhibited variability in the ~20 min to ~120 min range. This meant that the

perturbations in the ionospheric electron density were caused by atmospheric gravity waves (AGWs) generated by the

typhoon: the greater the AGW period, the greater the Doppler shift. As the period increased from 20 min to 120 min, the

Doppler shift amplitudes increased from ~0.1 Hz to 0.5–1 Hz.

(6) As the AGW period increases from 20 min to 60 min, the amplitude of quasi-sinusoidal variations in the

electron density increases from 0.4 to 6 per cent.

(7) The most important mechanism of affecting the ionosphere has been confirmed to be associated with the

generation of the 20–120 min period AGW by the typhoon.

(8) The Doppler measurements have shown that dusk terminators and the super typhoon acted synergistically to

amplify the ionospheric response to these sources of energy.

**Code Availability**. Software for Passive 14-Channel Doppler Radar may be obtained from the website at

https://dataverse.harvard.edu/dataset.xhtml?persistentId=doi:10.7910/DVN/MTGAVH (Garmash, 2021).

**Data Availability**. The data sets discussed in this paper may be obtained from the website at https://dataverse.harvard.edu/dataset.xhtml?persistentId=doi:10.7910/DVN/VHY0L2 (Garmash, 2022).

**Author Contribution**

**Conceptualization:** Leonid Chernogor; **Data Curation**: Qiang Guo and Kostiantyn Garmash; **Formal Analysis**: All Authors: Yu Zheng, Leonid Chernogor, Kostiantyn Garmash, Qiang Guo, Victor Rozumenko; **Funding Acquisition**: Qiang Guo; **Investigation**: Qiang Guo and Kostiantyn Garmash; **Methodology**: Leonid Chernogor; **Project Administration**: Qiang Guo; **Resources**: Qiang Guo; **Software**: Kostiantyn Garmash; **Supervision:** Leonid Chernogor; **Validation**: All Authors: Leonid Chernogor, Kostiantyn Garmash, Qiang Guo, Victor Rozumenko, Yu Zheng; **Visualization**: Kostiantyn Garmash, Yu Zheng; **Writing – original draft**: All Authors: Leonid Chernogor, Kostiantyn Garmash, Qiang Guo, Victor Rozumenko, Yu Zheng; **Writing review & editing**: All Authors: Leonid Chernogor, Kostiantyn Garmash, Qiang Guo, Victor Rozumenko, Yu Zheng.

**Competing Interests**. The authors declare that they have no conflict of interests

**Acknowledgments**

This article makes use of data on typhoon 201825 Kong-Rey recorded by the Japan Meteorological Agency and published at http://agora.ex.nii.ac.jp/digital-typhoon/summary/wnp/s/201825.html.en. The solar wind parameters were retrieved from the Goddard Space Flight Center Space Physics Data Facility https://omniweb.gsfc.nasa.gov/form/dx1.html. This research also draws upon data provided by the WK546 URSI code ionosonde in the city of Wakkanai (45.16° N, 141.75° E), Japan, URL: https://wdc.nict.go.jp/IONO/HP2009/contents/Ionosonde_Map_E.html (ionosonde data are retrieved from http://wdc.nict.go.jp/IONO/HP2009/ISDJ/index-E.html). Work by Qiang Guo and Yu Zheng was supported by the National Key R&D Plan Strategic International Science and Technology Cooperation and Innovation (2018YFE0206500). Work by L. F. Chernogor was supported by the National Research Foundation of Ukraine for financial support (project 2020.02/0015, "Theoretical and experimental studies of global disturbances from natural and technogenic sources in the Earth-atmosphere-ionosphere system"). Work by L. F. Chernogor and V. T. Rozumenko was supported by the Ukraine state research project #0121U109881, and work by K. P. Garmash was supported by the Ukraine state research project #0121U109882.

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
