# Peer review of "Effects of super-powerful tropospheric Western Pacific phenomenon"

_Annales Geophysicae, 2022_

## Referee Comment (RC1)

Review report on the Manuscript: "Ionospheric Effects over the People's Republic of China from the Super-Powerful Tropospheric Western Pacific Phenomenon of September–October 2018: Results from Oblique Sounding"

**(A) General Comments** :

The authors deserve merit for putting up efforts in analyzing and presenting Doppler spectra of the signals recorded over eight propagation paths for identifying typhoon-induced effects at the ionospheric height, a subject of importance in understanding lower-upper atmosphere coupling dynamics. The prime data in the analyses are oblique incidence signal reception quality in the frequency range 6.015 MHz – 9.75 MHz at Hibon China from the transmitters located in Japan, South Korea, and China.

However, the work requires clarification on some vital issues and needs supporting parameters to justify the final conclusion of the work which goes as "the periodic components of 20 min to 120 min at the ionospheric heights as reflected in the received signals are the effects of the superpowerful typhoon of September – October 2018". The authors need to provide the required inputs and clarifications for assessing the fulfillment of the aims of the work and to judge the scientific merit of the paper.

The paper thus requires major revision to make it suitable for publication in the esteemed Journal. A few suggestions and recommendations are put forward for possible implementation in the revised MS :

(i) Going through the Doppler spectra  (Figure  Nos. 6 to 13) presented separately for the different paths covering the period from September 29 to October 6, it is, however, observed that as claimed in the MS the components within 20 mins to 120 mins,   are not visible ( except the latter component in some cases) and apparently indistinguishable. These components must be well displayed along with their respective power because these are the basic parameters leading to the conclusion of the work. The results of observations also need to be coherent and clear which are somewhat missing and thus difficult to keep track of the records imprinted by the Typhoon (if any) on different propagation paths,  to make a constructive comment.

(ii) To strengthen the conclusion, it is also recommended that Doppler spectra for the period not influenced by the Typhoon may be presented along with the observations from   September 29 to October 6, which cover the growth and landfall days of the typhoon.

(iii) Further, to establish the growth of such periodic structures at the ionospheric height by the typhoon-induced wave -dynamics at the lower atmosphere, supporting evidence is necessary. It is thus suggested that the authors present profiles of any lower atmospheric/near-earth parameter during the period and around the locations covered by the study, to identify the features present therein with the wave components provided by their Doppler analysis.

(iv)The authors no doubt have tried to associate density fluctuation (ionosphere) with the formation of waves but need justification for their approach as the considered paths are of varied propagation statuses and positions. It is important also to provide physics and system dynamics associated with density modulations leading to the formation of waves. The scientific explanation is missing.

(v)The abstract, the basic key to the contents of the paper is not well spelled out and needs to be rewritten with clear objectives and approaches.

(vi) Discussions and Conclusions are to be modified accordingly.

(vii) There are scopes for improvement in sentence construction and also in clarity.

**(B) Other comments :**

(i) Line number 20 -25:The authors' statement that "typhoon-induced effects are clear near the midpoint of communication path".

This needs clarification when several propagation paths covering SE,  S, and NW directions are taken for their study and the mid-point of one propagation path varies from the other. From Figure 1(a) and Figure 5  one can see that the Typhoon trajectory and mid-point of the communication link may come nearer only from October 4 that too in the Harbin –Yomotta, Harbin- Chiba, Harbin-Goyang and Harbin-Hwasenong paths. These points may be cleared and looked into while explaining their observational results.

(ii) Concerning (i) above,  it is observed that wave components of  20 mins to 120 mins as claimed as Typhoon–induced effects are not visible, except for the Chiba-Herban link Doppler spectra which provide relatively clear components of  2/3 hrs ( Figure 7). But those signatures appeared even on September 29 -30, and also on October 6 ( also identified by the authors). It is not understood why on September 29 relatively clean wave structure is seen  ( Figure 7) when it was in a  tropical storm category and only on September 30,   the storm attained Typhoon status. Further,  the Kong Rey then weakened to category 3 on October 3 and degraded to a tropical storm on October 4, and made landfall on early October 6. Therefore it seems that the superpowerful typhoon effect may not be seen by the authors from October 4 as  Kong-Rey penetrated the Chinese mainland, at  18:00hrs  on October 3. The authors are suggested to look into their statement and discussion in this background and to provide  Doppler spectra for days free from typhoons for clearing the issue.

 (iii)  As the observations are in the oblique mode, the tropospheric effect cannot be ignored, when the contribution varies with the looking angle of the signals from the transmitter.   It is suggested that the authors analyze tropospheric/near surface parameters along the path of propagation ( or around) and look for  ( if detected) supporting inputs to justify the authors' conclusion ( already suggested above).

(iv) The Wakkanai ionogram may also be examined for such waves. The relevance of figures 3 and 4 are to be clearly spelled out and may be brought into the discussion while explaining the observed Spectral components. The only points of reference of these figures in the MS perhaps are in terms of diurnal variations in layer reflection height and critical frequency status during the period of study.

**( C ) Additional Comments :**

(i) Caption of the MS :

The caption may be more appropriate as "Effects of Super-Powerful Tropospheric Western Pacific Phenomenon of September–October 2018 on ionosphere over china: Results from Oblique Sounding".

(ii) Abstract :

The abstract is not well spelled out and needs to be rewritten with clear objectives and approaches. It is necessary to mention what ionospheric parameters the authors are monitoring, and of signal sources.

A few examples :

"The ionospheric response to the super typhoon action was clearly observed to occur both on October 1–2, 2018 (when the typhoon was 2,800–3,300 km from the propagation path midpoints .." ( As already identified  in other comments (i)  above)

 Similarly:  " … on October 5–6, 2018 when the typhoon was 1,000–1,500 km from the midpoints and its energy decreased by a factor of about 4."

And also   " The ionospheric effects are more pronounced along the nearest propagation paths, whereas no effect was detected along the propagation path at the farthest distance from the typhoon ".

These are vague and unspecified sentences carrying no meaning if not mentioned the receive mode and propagation paths of signals.

(iii) Introduction and Discussion :

In the introduction, the authors introduce relevant references on the need of understanding typhoon-induced effects by coupling dynamics between ocean-lower and upper atmosphere through gravity waves, water vapor condensation, and severe thunderstorms ( to name a few).

The authors have not brought up these parameters and issues either in the analysis or in the discussion. The introduction and discussion aspects should bear relevance.

(iv) Figures :

Figure 2: Here Space weather knowledge is no doubt relevant. These plots however may be omitted and  Kp, Dst magnitude statement may be enough to support the status of the days.

Figures 3 and 4: The relevance of the content of the figures to be placed in the MS where appropriate ( already identified above).

Figure 4: Check for the y-axis
Virtual height  E is to be replaced with Es.

 (v) Clarity of sentence suggested ( examples):

 (a) Line 170: "…Main ray and few rays "
 (b) Line 225-230:: " L 1.5 Hz, broadening±he Doppler spectra exhibit significant, up to  and such a diffuseness that the main ray is practically not distinguishable"

(vi ) Reconstruction of sentence necessary  ( examples)

(a) Line No 320: "The frequency of this radio wave became greater than the maximum usable frequency and  the radio wave penetrated the ionosphere during the second half of the nights. Consequently, the observation of became impossible".

 (b) line No 325: "During the night of September 30, 2018, the reflection of radio waves took place from the sporadic E layer,  resulting in fD(t) ≈ 0 Hz. During October 1, 2018, nighttime, the Doppler shift fD(t) ≈ 0 Hz) as well. During the course of the October 2. 2018, night, the Doppler shift was observed to change from –0.3 Hz to 0.3 Hz, the signal amplitude was observed to exhibit considerable variability, up to 20 dBV, In the course of the October 3–6, 2018 nights, the measurements were ineffective, whereas fD(t) ≈ 0 Hz at daytime."

Note also the highlighted segments.

**Final Comment** :

The paper needs major revision in light of the above suggestions and comments, before being considered suitable for publication.

---

## Author Comment (AC1)

Comment on angeo-2022-24
Anonymous Referee #1

Referee comment on "Ionospheric Effects over the People's Republic of China from the Super-Powerful Tropospheric Western Pacific Phenomenon of September–October 2018: Results from Oblique Sounding" by Leonid Chernogor et al., Ann. Geophys. Discuss., https://doi.org/10.5194/angeo-2022-24-RC1 , 2022

Reply to Anonymous Referee #1

Dear Anonymous Referee #1,

Thank you very much for your valuable comments that have helped the Authors greatly improve the draft of their paper.

Your comments are placed together with the Authors' answers (marked in yellow), and the changes made in the text of the manuscript, are also marked in yellow.

Authors.

Review report on the Manuscript: "Ionospheric Effects over the People's Republic of China from the Super-Powerful Tropospheric Western Pacific Phenomenon of September–October 2018: Results from Oblique Sounding"

(A) General Comments :

The authors deserve merit for putting up efforts in analyzing and presenting Doppler spectra of the signals recorded over eight propagation paths for identifying typhoon-induced effects at the ionospheric height, a subject of importance in understanding lower-upper atmosphere coupling dynamics. The prime data in the analyses are oblique incidence signal reception quality in the frequency range 6.015 MHz – 9.75 MHz at Hibon China from the transmitters located in Japan, South Korea, and China.

However, the work requires clarification on some vital issues and needs supporting parameters to justify the final conclusion of the work which goes as "the periodic components of 20 min to 120 min at the ionospheric heights as reflected in the received signals are the effects of the superpowerful typhoon of September – October 2018". The authors need to provide the required inputs and clarifications for assessing the fulfillment of the aims of the work and to judge the scientific merit of the paper.

The paper thus requires major revision to make it suitable for publication in the esteemed Journal. A few suggestions and recommendations are put forward for possible implementation in the revised MS :

(i) Going through the Doppler spectra (Figure Nos. 6 to 13) presented separately for the different paths covering the period from September 29 to October 6, it is, however, observed that as claimed in the MS the components within 20 mins to 120 mins, are not visible ( except the latter component in some cases) and apparently indistinguishable. These components must be well displayed along with their respective power because these are the basic parameters leading to the conclusion of the work. The results of observations also need to be coherent and clear which are somewhat missing and thus difficult to keep track of the records imprinted by the Typhoon (if any) on different propagation paths, to make a constructive comment.

**Dear Anonymous Referee #1, Thank you very much for this comment**.

The ionosphere is rarely quiet. Aperiodic and quasi-periodic perturbations systematically arise in it. Figures 6–13 show enhancements in variations in the Doppler spectra, and in particular, an increase in the fluctuations in the main ray. These fluctuations were almost always quasi-periodic, with the number of periods ranging from one to 4–5. The amplitude changed from 0.05 Hz to ~ 0.5, with larger periods corresponding to larger Doppler shifts. Basic information on the quasi-periodic perturbations (wave disturbances) is presented in Table 4. Certainly, the processes with periods from ~20 min to 120 min occurred.

The authors have constructed Table 4 showing the main parameters of the wave disturbances in October 2018.

Table 4
Basic parameters of wave disturbances in October 2018

| Radio-wave propagation path | Date | | | |
|---|---|---|---|---|
| | October 01 | October 02 | October 05 | October 06 |
| Hwaseong to Harbin | $T = 120$; 24 min $f_{Da} = 0.4$; 0.1 Hz | $T = 120$; 24 min $f_{Da} = 0.25$; 0.1 Hz | $T = 100–110$; 15 min $f_{Da} = 0.1–0.2$; 0.05 Hz | $T = 120$; 20 min $f_{Da} = 0.3$; 0.05 Hz |
| Chiba/Nagara to Harbin | $T = 60–80$ min $f_{Da} = 0.4$ Hz | $T = 20–30$ min $f_{Da} = 0.2–0.3$ Hz | $T = 30–40$ min $f_{Da} = 0.3$ Hz | $T = 100$ min $f_{Da} = 0.3$ Hz |
| Hailar/Nanmen to Harbin | $T = 80$; 15 min $f_{Da} = 0.4$; 0.05 Hz | $T = 40–50$ min $f_{Da} = 0.2–0.3$ Hz | $T = 40$; 20 min $f_{Da} = 0.2–0.3$ Hz | $T = 40–60$ min $f_{Da} = 0.1–0.2$ Hz |
| Beijing to Harbin | $T = 30–40$; 20–24 min $f_{Da} = 0.2$; 0.1 Hz | $T = 60$; 20–25 min $f_{Da} = 0.2$; 0.1 Hz | $T = 40–60$; 20 min $f_{Da} = 0.2$; 0.05 Hz | $T = 80$; 20–30 min $f_{Da} = 0.2$ Hz |
| Goyang to Harbin | $T = 120$; 30–40 min $f_{Da} = 0.3$; 0.1 Hz | $T = 40–60$; 30 min $f_{Da} = 0.2$; 0.1 Hz | $T = 80–90$ min $f_{Da} = 0.2$ Hz | $T = 100–120$; 20 min $f_{Da} = 0.4$; 0.1 Hz |
| Shijiazhuang to Harbin | $T = 120$min $f_{Da} = 0.3$ Hz | $T = 60$ min $f_{Da} = 0.3$ Hz | $T = 120$ min $f_{Da} = 0.3$ Hz | $T = 120$ min $f_{Da} = 0.1$ Hz |
| Hohhot to Harbin | – | $T = 120$; 40–50 min $f_{Da} = 0.2–0.3$ Hz | – | – |
| Yamata to Harbin | $T = 65–80$ min $f_{Da} = 0.3–0.4$ Hz | $T = 80$ min $f_{Da} = 0.2$ Hz | $T = 40$; 25–30 min $f_{Da} = 0.2$; 0.1 Hz | $T = 70$ min $f_{Da} = 0.2$ Hz |

Regarding the comment that "The results of observations also need to be coherent and **clear** which are somewhat missing and thus difficult to keep track of the records imprinted by the Typhoon (**if any**) on different propagation paths, to make a constructive comment", we use the following general methodology for revealing perturbations arising from any powerful source of energy:

(1) perturbations **from any powerful source releasing energy** are in principle **not distinguishable with each other**; (2) **any powerful source** can be associated **with any changes in the character of the signal** (Doppler shift, Doppler spectrum, the number of rays, discrete spectrum broadening, changes in the signal amplitude, etc.); (3) **intercomparisons** of radio wave characteristics observed **prior to and after** the release of energy must be made; (4) **intercomparisons** of the behavior of the radio wave characteristics observed on the day when the event occurred and on the **reference days** must be made. **Any differences may be due to** the release of energy; (5) the magnitudes of the **speeds of propagation** of disturbances must have a physical significance and correspond to the known types of waves (seismic, atmospheric gravity waves, infrasound, magnetohydrodynamic); (6) the data acquired over a large (10–14) number of propagation paths **must be consistent with each other**.

In the manuscript, this methodology is placed at the end of the section "5 Instrumentation and techniques".

(ii) To strengthen the conclusion, it is also recommended that Doppler spectra for the period not influenced by the Typhoon may be presented along with the observations from September 29 to October 6, which cover the growth and landfall days of the typhoon.
**Dear Anonymous Referee #1, Thank you very much for this comment**.
Now, we have increased the number of reference days (September 27 and 28, 2018 and October 7 and 8, 2018) shown in Figures 6–13. They really strengthen the conclusion that the Doppler spectra observed on October 1, 2, 5, and 6, 2018, differ from those observed on the reference days, testifying to the effects from the typhoon.

(iii) Further, to establish the growth of such periodic structures at the ionospheric height by the typhoon-induced wave -dynamics at the lower atmosphere, supporting evidence is necessary. It is thus suggested that the authors present profiles of any lower atmospheric/near-earth parameter during the period and around the locations covered by the study, to identify the features present therein with the wave components provided by their Doppler analysis.
**Dear Anonymous Referee #1, Thank you very much for this comment**.
The authors do not have "profiles of any lower atmospheric/near-earth parameter". Moreover, the transport of the typhoon-induced wave dynamics at the lower atmosphere to ionospheric heights involves a chain of non-linear processes preventing a simple identification of the features present in the troposphere with the wave components provided by their Doppler analysis. First, a typhoon is accompanied by strong neutral air turbulence, which leads to the generation of acoustic–atmospheric gravity waves over a wide range of frequencies (Drobyazko and Krasil'nikov, 1975). Second, these waves propagate to the upper atmosphere, partially dissipating their energy for heating the neutral air and launching secondary gravity waves in wave breaking regions (see, e.g., Vadas et al., 2003; Vadas and Crowley, 2010). Third, the latter waves in the atmosphere modulate the electron density [Schunk & Nagy, 2009], tracing neutral air turbulence, and act to generate plasma turbulence. Fourth, as a result, the Doppler spectra and the

Doppler shifts are observed to exhibit regular (quasiperiodic) and chaotic variations [Matthew, 1998].

Drobyazko, I. N., & Krasil'nikov, V. N. (1985). Generation of acoustic-gravity waves by atmospheric turbulence. *Radiophysics and Quantum Electronics, 28*(11), 946-952. https://doi.org/10.1007/bf01040717

Vadas, S. L., Fritts, D. C., & Alexander, M. J. (2003), Mechanism for the Generation of Secondary Waves in Wave Breaking Regions. *Journal of the Atmospheric Sciences*, 60 (1), 194–214.

Vadas, S. L., & Crowley, G. (2010). Sources of the traveling ionospheric disturbances observed by the ionospheric TIDDBIT sounder near Wallops Island on 30 October 2007. J. Geophys. Res., 115, A07324, doi:10.1029/2009JA015053.

Schunk and A. F. Nagy, Ionospheres: Physics, plasma physics, and Chemistry, Cambridge University Press, 2009 (ISBN-13 978-0-521-87706-0:

Matthew J. Angling, Paul S. Cannon, Nigel C. Davies, Tricia J. Willink, Vivianne Jodalen, and Bengt Lundborg, Measurements of Doppler and multipath spread on oblique high-latitude HF paths and their use in characterizing data modem performance, Radio Science, Volume 33, Number 1, Pages 97-107, January-February 1998. Paper number 97RS02206. 0048-6604/98/97RS-02206

(iv)The authors no doubt have tried to associate density fluctuation (ionosphere) with the formation of waves but need justification for their approach as the considered paths are of varied propagation statuses and positions. It is important also to provide physics and system dynamics associated with density modulations leading to the formation of waves. The scientific explanation is missing.

**Dear Anonymous Referee #1, Thank you very much for this comment**. Use has been made of radio wave paths with various orientations and distances from the typhoon, which permitted the identification of the ionospheric effects associated with the typhoon. Indeed, certain effects were observed along the nearest radiowave propagation paths, while such effects were absent along the most distant ones (Hohhot to Harbin) from the typhoon. This is the best proof of the influence of the typhoon.

Regarding the physics and system dynamics associated with density modulations leading to the formation of waves, it is presented in textbooks on the ionosphere (see, e.g., Schunk and A. F. Nagy, Ionospheres: Physics, plasma physics, and Chemistry, Cambridge University Press, 2009 (ISBN-13 978-0-521-87706-0:), and therefore such an explanation is not being considered suitable for publication in the esteemed Journal.

(v)The abstract, the basic key to the contents of the paper is not well spelled out and needs to be rewritten with clear objectives and approaches.

**Dear Anonymous Referee #1, Thank you very much for this comment**.
The scientific objectives of the study are to reveal the possible perturbations caused by typhoon Kong–Rey action, and to estimate the possible wave parameters of the ionosphere and radio signals. The abstract has been updated, and it is as follows:

**Abstract.**
Doppler measurements at oblique propagation paths from the City of Harbin, People's Republique of China (PRC), to ten HF radio broadcast stations in the PRC, Japan, Mongolia, and the Republic of Korea captured the response in the ionosphere to the super typhoon Kong-Rey action from September 30, 2018, to October 6, 2018. The Harbin Engineering University coherent software defined radio system accumulates the database containing the complex amplitudes of the radio signals acquired along 14 propagation paths since 2018. The complex amplitudes are used for calculating the temporal dependences of the Doppler spectra and signal amplitudes, and the Doppler spectra are used to plot the Doppler shift as a function of time, $f_D(t)$, for all rays. The scientific objectives of this study are to reveal the possible perturbations caused by the action of typhoon Kong-Rey, and to estimate the magnitudes of wave parameters of the ionospheric plasma and radio signals. The amplitudes, $f_{Da}$, of the Doppler shift variations were observed to noticeably increase (factor of ~2–3) on October 1–2 and 5–6, 2018, while the 20–120 min periods, $T$, of the Doppler shift variations suggest that the wavelike disturbances in the ionosphere are caused by atmospheric gravity waves. The periods and amplitudes of quasi-sinusoidal variations in the Doppler shift, which have been determined for all propagation paths, may be used to estimate the amplitudes, $\delta_{Na}$, of quasi-sinusoidal variations in the electron density. Thus, $T \approx 20$ min and $f_{Da} \approx 0.1$ Hz yield $\delta_{Na} \approx 0.4\%$, whereas $T \approx 30$ min and $f_{Da} \approx 0.2$ Hz give $\delta_{Na} \approx 1.2\%$. If $T \approx 60$ min and $f_{Da} \approx 0.5$ Hz, then $\delta_{Na} \approx 6\%$. The periods $T$ are found to change within the 15–120 min limits, and the Doppler shift amplitudes, $f_{Da}$, show variability within the 0.05–0.4 Hz limits.

(vi) Discussions and Conclusions are to be modified accordingly.
**Dear Anonymous Referee #1, Thank you very much for this comment**.
The discussion and conclusions have been modified accordingly.

(vii) There are scopes for improvement in sentence construction and also in clarity.
(vii) We have made the improvements.

(B) Other comments :

(i) Line number 20 -25: The authors' statement that "typhoon-induced effects are clear near the midpoint of communication path". This needs clarification when several propagation paths covering SE, S, and NW directions are taken for their study and the mid-point of one propagation path varies from the other. From Figure 1(a) and Figure 5 one can see that the Typhoon trajectory and mid-point of the communication link may come nearer only from October 4 that too in the Harbin –Yomotta, Harbin- Chiba, Harbin-Goyang and Harbin-Hwasenong paths. These points may be cleared and looked into while explaining their observational results.

**Dear Anonymous Referee #1, Thank you very much for this comment**.
Regarding the midpoints, the movement of the midpoints makes the main contribution to the Doppler shift observed at oblique incidence (see, e.g., Davies,

K. Ionospheric radio. Peter Peregrinus Ltd., 2008. ISBN (13 digit) 978-0-86341-186-1, 1989).

The time when the effects are observed is more important than the place where the typhoon is located. If the typhoon approached on October 04, then the response to the typhoon might be naturally expected to occur on October 05 and 06.

(ii) Concerning (i) above, it is observed that wave components of 20 mins to 120 mins as claimed as Typhoon–induced effects are not visible, except for the Chiba-Herban link Doppler spectra which provide relatively clear components of 2/3 hrs ( Figure 7). But those signatures appeared even on September 29 -30, and also on October 6 ( also identified by the authors). It is not understood why on September 29 relatively clean wave structure is seen ( Figure 7) when it was in a tropical storm category and only on September 30, the storm attained Typhoon status. Further, the Kong Rey then weakened to category 3 on October 3 and degraded to a tropical storm on October 4, and made landfall on early October 6. Therefore it seems that the superpowerful typhoon effect may not be seen by the authors from October 4 as Kong-Rey penetrated the Chinese mainland, at 18:00hrs on October 3. The authors are suggested to look into their statement and discussion in this background and to provide Doppler spectra for days free from typhoons for clearing the issue.

**Dear Anonymous Referee #1, Thank you very much for this comment**.

This is not entirely true. Quasi-periodic perturbations are observed along all propagation paths except for the Hohhot to Harbin propagation path. The quasi-periodic perturbations in Figure 7 are just more clearly evident. Rather, such structures arise on October 1 and 2, 2018, as well as on October 5 and 6, 2018.

The authors have constructed Table 4 showing the main parameters of the wave disturbances in October 2018.

Table 4
Basic parameters of wave disturbances in October 2018

| Radio-wave propagation path | Date | | | |
|---|---|---|---|---|
| | October 01 | October 02 | October 05 | October 06 |
| Hwaseong to Harbin | $T = 120$; 24 min $f_{Da} = 0.4$; 0.1 Hz | $T = 120$; 24 min $f_{Da} = 0.25$; 0.1 Hz | $T = 100–110$; 15 min $f_{Da} = 0.1–0.2$; 0.05 Hz | $T = 120$; 20 min $f_{Da} = 0.3$; 0.05 Hz |
| Chiba/Nagara to Harbin | $T = 60–80$ min $f_{Da} = 0.4$ Hz | $T = 20–30$ min $f_{Da} = 0.2–0.3$ Hz | $T = 30–40$ min $f_{Da} = 0.3$ Hz | $T = 100$ min $f_{Da} = 0.3$ Hz |
| Hailar/Nanmen to Harbin | $T = 80$; 15 min $f_{Da} = 0.4$; 0.05 Hz | $T = 40–50$ min $f_{Da} = 0.2–0.3$ Hz | $T = 40$; 20 min $f_{Da} = 0.2–0.3$ Hz | $T = 40–60$ min $f_{Da} = 0.1–0.2$ Hz |
| Beijing to Harbin | $T = 30–40$; 20–24 min $f_{Da} = 0.2$; 0.1 Hz | $T = 60$; 20–25 min $f_{Da} = 0.2$; 0.1 Hz | $T = 40–60$; 20 min $f_{Da} = 0.2$; 0.05 Hz | $T = 80$; 20–30 min $f_{Da} = 0.2$ Hz |
| Goyang to Harbin | $T = 120$; 30–40 min $f_{Da} = 0.3$; 0.1 Hz | $T = 40–60$; 30 min $f_{Da} = 0.2$; 0.1 Hz | $T = 80–90$ min $f_{Da} = 0.2$ Hz | $T = 100–120$; 20 min $f_{Da} = 0.4$; 0.1 Hz |
| Shijiazhuang to | $T = 120$min | $T = 60$ min | $T = 120$ min | $T = 120$ min |

| | | | | |
|---|---|---|---|---|
| Harbin | $f_{Da}$ = 0.3 Hz | $f_{Da}$ = 0.3 Hz | $f_{Da}$ = 0.3 Hz | $f_{Da}$ = 0.1 Hz |
| Hohhot to Harbin | – | $T$ = 120; 40–50 min $f_{Da}$ = 0.2–0.3 Hz | – | – |
| Yamata to Harbin | $T$ = 65–80 min $f_{Da}$ = 0.3–0.4 Hz | $T$ = 80 min $f_{Da}$ = 0.2 Hz | $T$ = 40; 25–30 min $f_{Da}$ = 0.2; 0.1 Hz | $T$ = 70 min $f_{Da}$ = 0.2 Hz |

The authors also point out that although the typhoon somewhat weakened on October 3 and 4, 2018, it achieved the closest distance to the propagation paths. Therefore, the ionospheric effects from the typhoon were again observed on October 5 and 6, 2018.

(iii) As the observations are in the oblique mode, the tropospheric effect cannot be ignored, when the contribution varies with the looking angle of the signals from the transmitter. It is suggested that the authors analyze tropospheric/near surface parameters along the path of propagation ( or around) and look for ( if detected) supporting inputs to justify the authors' conclusion ( already suggested above).
**Dear Anonymous Referee #1, Thank you very much for this comment**.
The tropospheric effects cannot arise at 5–10 MHz frequencies, because the non-conducting atmosphere below the ionosphere is treated as free space, with refractive index unity, in the HF frequency range (see, e.g., Budden, K. G., The propagation of radio waves: The theory of radio waves of low power in the ionosphere and magnetosphere, Cambridge University Press, 1988).

(iv) The Wakkanai ionogram may also be examined for such waves. The relevance of figures 3 and 4 are to be clearly spelled out and may be brought into the discussion while explaining the observed Spectral components. The only points of reference of these figures in the MS perhaps are in terms of diurnal variations in layer reflection height and critical frequency status during the period of study.
**Dear Anonymous Referee #1, Thank you very much for this comment**.
We would also like the Wakkanai ionosonde to provide us with ionograms updated every minute. However, Figures 3 and 4 show ionogram measurements acquired with an update rate of one ionogram per 1 hour. Thus, these ionograms cannot give information on the ~20–120-min period wave processes. Nevertheless, they are used for analyzing the state of space weather in Section *4 Analysis of the State of the Ionosphere*.

( C ) Additional Comments :

(i) Caption of the MS :

The caption may be more appropriate as "Effects of Super-Powerful Tropospheric Western Pacific Phenomenon of September–October 2018 on ionosphere over china: Results from Oblique Sounding".
**Dear Anonymous Referee #1, Thank you very much for this comment**.
We have changed the title of this paper.

(ii) Abstract :

The abstract is not well spelled out and needs to be rewritten with clear objectives and approaches. It is necessary to mention what ionospheric parameters the authors are monitoring, and of signal sources.

**Dear Anonymous Referee #1, Thank you very much for this comment**.

We have rewritten the abstract as follows.

**Abstract.**

Doppler measurements at oblique propagation paths from the City of Harbin, People's Republique of China (PRC), to ten HF radio broadcast stations in the PRC, Japan, Mongolia, and the Republic of Korea captured the response in the ionosphere to the super typhoon Kong-Rey action from September 30, 2018, to October 6, 2018. The Harbin Engineering University coherent software defined radio system accumulates the database containing the complex amplitudes of the radio signals acquired along 14 propagation paths since 2018. The complex amplitudes are used for calculating the temporal dependences of the Doppler spectra and signal amplitudes, and the Doppler spectra are used to plot the Doppler shift as a function of time, $f_D(t)$, for all rays. The scientific objectives of this study are to reveal the possible perturbations caused by the action of typhoon Kong-Rey, and to estimate the magnitudes of wave parameters of the ionospheric plasma and radio signals. The amplitudes, $f_{Da}$, of the Doppler shift variations were observed to noticeably increase (factor of ~2–3) on October 1–2 and 5–6, 2018, while the 20–120 min periods, $T$, of the Doppler shift variations suggest that the wavelike disturbances in the ionosphere are caused by atmospheric gravity waves. The periods and amplitudes of quasi-sinusoidal variations in the Doppler shift, which have been determined for all propagation paths, may be used to estimate the amplitudes, $\delta_{Na}$, of quasi-sinusoidal variations in the electron density. Thus, $T \approx 20$ min and $f_{Da} \approx 0.1$ Hz yield $\delta_{Na} \approx 0.4\%$, whereas $T \approx 30$ min and $f_{Da} \approx 0.2$ Hz give $\delta_{Na} \approx 1.2\%$. If $T \approx 60$ min and $f_{Da} \approx 0.5$ Hz, then $\delta_{Na} \approx 6\%$. The periods $T$ are found to change within the 15–120 min limits, and the Doppler shift amplitudes, $f_{Da}$, show variability within the 0.05–0.4 Hz limits.

A few examples :

"The ionospheric response to the super typhoon action was clearly observed to occur both on October 1–2, 2018 (when the typhoon was 2,800–3,300 km from the propagation path midpoints .." ( As already identified in other comments (i) above)

Similarly: " … on October 5–6, 2018 when the typhoon was 1,000–1,500 km from the midpoints and its energy decreased by a factor of about 4."

And also " The ionospheric effects are more pronounced along the nearest propagation paths, whereas no effect was detected along the propagation path at the farthest distance from the typhoon ".

These are vague and unspecified sentences carrying no meaning if not mentioned the receive mode and propagation paths of signals.

**Dear Anonymous Referee #1, Thank you very much for these comments**. The seeming vagueness is rooted in the general methodology of revealing the effects that are due to any powerful source of energy above. For convenience, it is copied below:

To reveal perturbations arising from any powerful source of energy, the following general methodology is invoked: (1) perturbations from any powerful source releasing energy are in principle not distinguishable with each other; (2) any powerful source can be associated with any changes in the character of the signal (Doppler shift, Doppler spectrum, the number of rays, discrete spectrum broadening, changes in the signal amplitude, etc.); (3) intercomparisons of radio wave characteristics observed prior to and after the release of energy must be made; (4) intercomparisons of the behavior of the radio wave characteristics observed on the day when the event occurred and on the reference days must be made. Any differences may be due to the release of energy; (5) the magnitudes of the speeds of propagation of disturbances must have a physical significance and correspond to the known types of waves (seismic, atmospheric gravity waves, infrasound, magnetohydrodynamic); (6) the data acquired over a large (10–14) number of propagation paths must be consistent with each other.

(iii) Introduction and Discussion :

In the introduction, the authors introduce relevant references on the need of understanding typhoon-induced effects by coupling dynamics between ocean-lower and upper atmosphere through gravity waves, water vapor condensation, and severe thunderstorms ( to name a few).

The authors have not brought up these parameters and issues either in the analysis or in the discussion. The introduction and discussion aspects should bear relevance. **Dear Anonymous Referee #1, Thank you very much for this comment**.

Indeed, the introduction strives to depict an entire broad research effort among scientists from around the world aimed at acquiring a deep understanding of the physical processes that drive the ocean–atmosphere–ionosphere–magnetosphere system. Also, a broad spectrum of instruments to investigate these processes is mentioned.

The scope of the last paragraph in the introduction narrows to the scientific objectives achievable with the instrument the authors have created. The paragraph is as follows:

The scientific objectives of this study is to reveal the processes that the typhoon–dusk terminator coupling brings into play, to derive specifications of wave periods and amplitudes of perturbations in the in the electron density, and to estimate the space scales of the perturbations launched by the super typhoon Kong-Rey event of September–October 2018 into the ionosphere over the People's Republic of China (PRC).The observations were made using the Harbin Engineering University, PRC, multifrequency multiple path coherent software defined radio system for probing the ionosphere at oblique incidence.

(iv) Figures :

Figure 2: Here Space weather knowledge is no doubt relevant. These plots however may be omitted and Kp, Dst magnitude statement may be enough to support the status of the days.
**Dear Anonymous Referee #1, Thank you very much for this comment**.
The authors consider the presented in the MS analysis of the state of space weather to be important and its reduction to be inappropriate.

Figures 3 and 4: The relevance of the content of the figures to be placed in the MS where appropriate ( already identified above).
**Dear Anonymous Referee #1, Thank you very much for this comment**. The relevance of the data in Figures 3 and 4 have already been already identified above as follows.
Figures 3 and 4 show ionogram measurements acquired with an update rate of one ionogram per 1 hour. Therefore, these ionograms cannot provide information on the ~20–120-min period wave processes. Nevertheless, they are used for analyzing the state of space weather in Section *4 Analysis of the State of the Ionosphere*.

Figure 4: Check for the y-axis

Virtual height E is to be replaced with Es.
**Dear Anonymous Referee #1, Thank you very much for this comment**.
In Figure 4, both E and Es virtual heights are presented.

(v) Clarity of sentence suggested ( examples):

(a) Line 170: "…Main ray and few rays "

**Dear Anonymous Referee #1, Thank you very much for this comment**. Indeed, this phrase sounds stupid. The phrase is altered as follows: all rays under analysis

(b) Line 225-230:: " L 1.5 Hz, broadening±he Doppler spectra exhibit significant, up to and such a diffuseness that the main ray is practically not distinguishable"

**Dear Anonymous Referee #1, Thank you very much for this comment**. This misprint has been corrected as follows:
the Doppler spectra exhibit significant, up to ±1.5 Hz, broadening and such a diffuseness that the main ray is practically not distinguishable.

(vi ) Reconstruction of sentence necessary ( examples)

(a) Line No 320: "The frequency of this radio wave became greater than the maximum usable frequency and the radio wave penetrated the ionosphere during the second half of the nights. Consequently, the observation of became impossible".

**Dear Anonymous Referee #1, Thank you very much for this comment**.
The sentences have been reconstructed as follows:
The frequency of this radio wave became greater than the maximum usable frequency and the radio wave penetrated the ionosphere during the second half of all nights (see Figure 12). The received signal was absent, and the observation of the ionospheric dynamics became impossible.

(b) line No 325: "During the night of September 30, 2018, the reflection of radio waves took place from the sporadic E layer, resulting in fD(t) » 0 Hz. During October 1, 2018, nighttime, the Doppler shift fD(t) » 0 Hz) as well. During the course of the October 2. 2018, night, the Doppler shift was observed to change from –0.3 Hz to 0.3 Hz, the signal amplitude was observed to exhibit considerable variability, up to 20 dBV, In the course of the October 3–6, 2018 nights, the measurements were ineffective, whereas fD(t) » 0 Hz at daytime."

Note also the highlighted segments.

**Dear Anonymous Referee #1, Thank you very much for this comment**.
We have re-written line No 325 as follows:

On the UT night of September 30, 2018, the reflection of radio waves took place from the sporadic $E$ layer, resulting in $f_D(t) \approx 0$ Hz, and during the UT night of October 1, 2018, the Doppler shift $f_D(t) \approx 0$ Hz as well. On the UT night of October 2, 2018, the Doppler shift showed changes from –0.3 Hz to 0.3 Hz, while the signal amplitude exhibited considerable variability, up to 20 dBV. During the UT nights of October 3–6, 2018, the measurements were ineffective, whereas $f_D(t) \approx 0$ Hz at daytime.

Final Comment

The paper needs major revision in light of the above suggestions and comments, before being considered suitable for publication.

Please also note the supplement to this comment:
https://angeo.copernicus.org/preprints/angeo-2022-24/angeo-2022-24-RC1-supplement. pdf

**Dear Anonymous Referee #1, Thank you very much for this comment**. Your suggestions and comments have helped the Authors to significantly improve the manuscript.

Sincerely,
Authors.

---

## Author Response (AR1)

**This file contains Authors' responses and a list of changes made in the manuscript**

Ann. Geophys. Discuss., referee comment RC1 https://doi.org/10.5194/angeo-2022-24-RC1 , 2022 © Author(s) 2022. This work is distributed under the Creative Commons Attribution 4.0 License.

Comment on angeo-2022-24
Anonymous Referee #1

Referee comment on "Ionospheric Effects over the People's Republic of China from the Super-Powerful Tropospheric Western Pacific Phenomenon of September–October 2018: Results from Oblique Sounding" by Leonid Chernogor et al., Ann. Geophys. Discuss., https://doi.org/10.5194/angeo-2022-24-RC1 , 2022

Reply to Anonymous Referee #1

Dear Anonymous Referee #1,

Thank you very much for your valuable comments that have helped the Authors greatly improve the draft of their paper.

Referee #1 comments are placed together with the Authors' answers (marked in yellow), and the changes made in the text of the manuscript are also marked in yellow.

Authors.
* * *
Review report on the Manuscript: "Ionospheric Effects over the People's Republic of China from the Super-Powerful Tropospheric Western Pacific Phenomenon of September–October 2018: Results from Oblique Sounding"

(A) General Comments :

The authors deserve merit for putting up efforts in analyzing and presenting Doppler spectra of the signals recorded over eight propagation paths for identifying typhoon-induced effects at the ionospheric height, a subject of importance in understanding lower-upper atmosphere coupling dynamics. The prime data in the analyses are oblique incidence signal reception quality in the frequency range 6.015 MHz – 9.75 MHz at Hibon China from the transmitters located in Japan, South Korea, and China.

However, the work requires clarification on some vital issues and needs supporting parameters to justify the final conclusion of the work which goes as "the periodic components of 20 min to 120 min at the ionospheric heights as reflected in the received signals are the effects of the superpowerful typhoon of September – October 2018". The authors need to provide the required inputs and clarifications for assessing the fulfillment of the aims of the work and to judge the scientific merit of the paper.

The paper thus requires major revision to make it suitable for publication in the esteemed Journal. A few suggestions and recommendations are put forward for possible implementation in the revised MS :

(i) Going through the Doppler spectra (Figure Nos. 6 to 13) presented separately for the different paths covering the period from September 29 to October 6, it is, however, observed that as claimed in the MS the components within 20 mins to 120 mins, are not visible ( except the latter component in some cases) and apparently indistinguishable. These components must be well displayed along with their respective power because these are the basic parameters leading to the conclusion of the work. The results of observations also need to be coherent and clear which are

somewhat missing and thus difficult to keep track of the records imprinted by the Typhoon (if any) on different propagation paths, to make a constructive comment.

**Dear Anonymous Referee #1, Thank you very much for this comment**.

To make "the results of observations … clear"er, **(i) a general methodology** for revealing perturbations arising from any powerful source of energy **has been written and placed** at the end of the section "5 Instrumentation and techniques" (**Line 193–227, Page 11–12**)**, and (ii) Table 5** (Line 451–454, Page 37–38) **has been constructed**. The methodology is as follows:

(i) **Methodology** (**Line 193–227, Page 11–12**):

[revised manuscript text omitted]

(ii) **Table 5**

Regarding the comment that "the MS the components within 20 mins to 120 mins, are not visible", the ionosphere is rarely quiet. Aperiodic and quasi-periodic perturbations systematically arise in it. Figures 6–13 show enhancements in variations in the Doppler spectra, and in particular, an increase in the fluctuations in the main ray. These fluctuations were almost always quasi-periodic, with the number of periods ranging from one to 4–5. The amplitude changed from 0.05 Hz to ~ 0.5, with larger periods corresponding to larger Doppler shifts. Basic information on the quasi-periodic perturbations (wave disturbances) is presented in Table 5. Certainly, the processes with periods from ~20 min to 120 min occurred.

The authors **have constructed Table 5** (Line 451–454, Page 37–38) showing the main parameters of the wave disturbances in October 2018.

Table 5
Basic parameters of wave disturbances in October 2018

| Radio-wave propagation path | Date | | | |
|---|---|---|---|---|
| | October 01 | October 02 | October 05 | October 06 |
| Hwaseong to Harbin | $T = 120$; 24 min $f_{Da} = 0.4$; 0.1 Hz | $T = 120$; 24 min $f_{Da} = 0.25$; 0.1 Hz | $T = 100$–110; 15 min $f_{Da} = 0.1$–0.2; 0.05 Hz | $T = 120$; 20 min $f_{Da} = 0.3$; 0.05 Hz |
| Chiba/Nagara to Harbin | $T = 60$–80 min $f_{Da} = 0.4$ Hz | $T = 20$–30 min $f_{Da} = 0.2$–0.3 Hz | $T = 30$–40 min $f_{Da} = 0.3$ Hz | $T = 100$ min $f_{Da} = 0.3$ Hz |
| Hailar/Nanmen to Harbin | $T = 80$; 15 min $f_{Da} = 0.4$; 0.05 Hz | $T = 40$–50 min $f_{Da} = 0.2$–0.3 Hz | $T = 40$; 20 min $f_{Da} = 0.2$–0.3 Hz | $T = 40$–60 min $f_{Da} = 0.1$–0.2 Hz |
| Beijing to Harbin | $T = 30$–40; 20–24 min $f_{Da} = 0.2$; 0.1 Hz | $T = 60$; 20–25 min $f_{Da} = 0.2$; 0.1 Hz | $T = 40$–60; 20 min $f_{Da} = 0.2$; 0.05 Hz | $T = 80$; 20–30 min $f_{Da} = 0.2$ Hz |
| Goyang to Harbin | $T = 120$; 30–40 min $f_{Da} = 0.3$; 0.1 Hz | $T = 40$–60; 30 min $f_{Da} = 0.2$; 0.1 Hz | $T = 80$–90 min $f_{Da} = 0.2$ Hz | $T = 100$–120; 20 min $f_{Da} = 0.4$; 0.1 Hz |
| Shijiazhuang to Harbin | $T = 120$ min $f_{Da} = 0.3$ Hz | $T = 60$ min $f_{Da} = 0.3$ Hz | $T = 120$ min $f_{Da} = 0.3$ Hz | $T = 120$ min $f_{Da} = 0.1$ Hz |
| Hohhot to Harbin | – | $T = 120$; 40–50 min $f_{Da} = 0.2$–0.3 Hz | – | – |
| Yamata to Harbin | $T = 65$–80 min $f_{Da} = 0.3$–0.4 Hz | $T = 80$ min $f_{Da} = 0.2$ Hz | $T = 40$; 25–30 min $f_{Da} = 0.2$; 0.1 Hz | $T = 70$ min $f_{Da} = 0.2$ Hz |

Regarding the comment that "The results of observations also need to be coherent and clear which are somewhat missing and thus difficult to keep track of the records imprinted by the Typhoon (if any) on different propagation paths, to make a constructive comment", the Authors **have written** the general methodology presented above (**Line 193–227, Page 11–12**):

(ii) To strengthen the conclusion, it is also recommended that Doppler spectra for the period not influenced by the Typhoon may be presented along with the observations from September 29 to October 6, which cover the growth and landfall days of the typhoon.
**Dear Anonymous Referee #1, Thank you very much for this comment**.
The authors **have redone Figures 6–13 (pages 17,18, 20–25, 27–30, 32–35)** to show additional reference days (September 27 and 28, 2018 and October 7 and 8, 2018). They really strengthen the conclusion that the Doppler spectra observed on October 1, 2, 5, and 6, 2018, differ from those observed on the reference days, testifying to the effects from the typhoon.

(iii) Further, to establish the growth of such periodic structures at the ionospheric height by the typhoon-induced wave -dynamics at the lower atmosphere, supporting evidence is necessary. It is thus suggested that the authors present profiles of any lower atmospheric/near-earth parameter during the period and around the locations covered by the study, to identify the features present therein with the wave components provided by their Doppler analysis.

**Dear Anonymous Referee #1, Thank you very much for this comment**.

It is inconceivable "that the authors present profiles of any lower atmospheric/near-earth parameter … to identify the features present therein with the wave components provided by their Doppler analysis.". The transport of the typhoon-induced wave dynamics at the lower atmosphere to ionospheric heights involves a chain of non-linear processes preventing a simple identification of the features present in the troposphere with the wave components provided by their Doppler analysis. First, a typhoon is accompanied by strong neutral air turbulence, which leads to the generation of acoustic–atmospheric gravity waves over a wide range of frequencies (Drobyazko and Krasil'nikov, 1975). Second, these waves propagate to the upper atmosphere, partially dissipating their energy for heating the neutral air and launching secondary gravity waves in the wave breaking regions (see, e.g., Vadas et al., 2003; Vadas and Crowley, 2010). Third, the latter waves in the atmosphere modulate the electron density (e.g., Schunk & Nagy, 2009) and may act to generate irregularity structure (Perkins, 1973). Fourth, as a result, the Doppler spectra and the Doppler shifts are observed to exhibit regular (quasi-sinusoidal) or irregular behavior, and sometimes, diffuseness (or spread $F$ in ionograms, see, e.g., Wang et al., 2019).

Drobyazko, I. N., & Krasil'nikov, V. N. (1985). Generation of acoustic-gravity waves by atmospheric turbulence. *Radiophysics and Quantum Electronics, 28*(11), 946-952. https://doi.org/10.1007/bf01040717

Vadas, S. L., Fritts, D. C., & Alexander, M. J. (2003), Mechanism for the Generation of Secondary Waves in Wave Breaking Regions. *Journal of the Atmospheric Sciences*, 60 (1), 194–214.

Vadas, S. L., & Crowley, G. (2010). Sources of the traveling ionospheric disturbances observed by the ionospheric TIDDBIT sounder near Wallops Island on 30 October 2007. J. Geophys. Res., 115, A07324, doi:10.1029/2009JA015053.

Schunk, R. W., and A. F. Nagy, Ionospheres: Physics, plasma physics, and Chemistry, Cambridge University Press, 2009 (ISBN-13 978-0-521-87706-0:

Perkins, F. W.: Spread F and ionospheric currents. J. Geophys. Res. 78, 218–226. https://doi.org/10.1029/JA078i001p00218, 1973

Wang, N., Gui, L., Ding, Z., Zhao, Z., Xu Z., & Hu, Y., Longitudinal differences in the statistical characteristics of ionospheric spread-F occurrences at midlatitude in Eastern Asia, Earth, Planets and Space, (2019) 71, Paper #47, https://doi.org/10.1186/s40623-019-1026-6

**Therefore, no changes were made to the revised version of the manuscript.**

(iv)The authors no doubt have tried to associate density fluctuation (ionosphere) with the formation of waves but need justification for their approach as the considered paths are of varied propagation statuses and positions. It is important also to provide physics and system dynamics associated with density modulations leading to the formation of waves. The scientific explanation is missing.

**Dear Anonymous Referee #1, Thank you very much for this comment**.

Use has been made of radio wave paths with various orientations and distances from the typhoon, which permitted the identification of the ionospheric effects associated with the typhoon.

Certain effects were observed along the nearest radiowave propagation paths, while such effects were absent along the most distant ones (Hohhot to Harbin) from the typhoon. This is the best proof of the influence of the typhoon.

Regarding the "physics and system dynamics associated with density modulations leading to the formation of waves", the basics of the physics are well-known and presented in textbooks on the ionosphere (see, e.g., Schunk and A. F. Nagy, Ionospheres: Physics, plasma

physics, and Chemistry, Cambridge University Press, 2009 (ISBN-13 978-0-521-87706-0: www.cambridge.org/9780521877060). Therefore, the basics cannot be considered suitable for publication in an "esteemed Journal".

At the same time, the real problem of electron density behavior is beyond the current instrumental capabilities of the Authors.

Generally, atmospheric gravity waves (AGWs) are responsible for traveling ionospheric disturbances (TIDs). There are a few outcomes depending on AGW numerical parameters. First, the height of the level of reflection can show variability within up to an order of 100 km limits. Second, there may arise a few mirror points, and consequently the number of rays increases. Third, plasma irregularities can occur in the ionosphere, and as consequence diffuseness occur in the Doppler measurements, and spread $F$ in the ionograms (see, e.g., Perkins, F. W.: Spread F and ionospheric currents. J. Geophys. Res. 78, 218–226. https://doi.org/10.1029/JA078i001p00218, 1973). The subtle issues of the generation of the latter irregularities remain unsolved till now (see, e.g., Wang, N., Gui, L., Ding, Z., Zhao, Z., Xu Z., & Hu, Y., Longitudinal differences in the statistical characteristics of ionospheric spread-F occurrences at midlatitude in Eastern Asia, Earth, Planets and Space, (2019) 71, Paper #47, https://doi.org/10.1186/s40623-019-1026-6).

**Therefore, the physics of the formations of waves cannot be and is not the goal of the present paper.**

Regarding HF radio wave propagation (according to the textbooks (see, e.g., Davies, K. (2008). Ionospheric Radio. Peter Peregrinus, London)), there should be four rays even in a steady state and smooth ionosphere (the low-angle ray and the high-angle or Pedersen ray for both ordinary and extraordinary radio waves), with the real number of rays observed depending on the signal-to-noise ratio. The Authors do determine the rays via Doppler shifts. Taking measurements of the angles of arrival of all rays requires an antenna array, whereas the radio system created by the Authors employs only a single loop antenna.

**Therefore, no changes were made to the revised version of the manuscript.**

(v) The abstract, the basic key to the contents of the paper is not well spelled out and needs to be rewritten with clear objectives and approaches.

**Dear Anonymous Referee #1, Thank you very much for this comment**.
The scientific objectives of the study are to reveal the possible perturbations caused by typhoon Kong–Rey action, and to estimate the possible wave parameters of the ionosphere and radio signals. **The abstract has been updated** (Line 14–28, Page1)**,** and it is as follows:

**Abstract.**
Doppler measurements at oblique propagation paths from the City of Harbin, People's Republic of China (PRC), to ten HF radio broadcast stations in the PRC, Japan, Mongolia, and the Republic of Korea captured the response in the ionosphere to the super typhoon Kong-Rey action from 30 September 2018 to 6 October 2018. The Harbin Engineering University coherent software defined radio system accumulates the database containing the complex amplitudes of the radio signals acquired along 14 propagation paths since 2018. The complex amplitudes are used for calculating the temporal dependences of the Doppler spectra and signal amplitudes, and the Doppler spectra are used to plot the Doppler shift as a function of time, $f_D(t)$, for all rays. The scientific objectives of this study are to reveal the possible perturbations caused by the action of typhoon Kong-Rey, and to estimate the magnitudes of wave parameters of the ionospheric plasma and radio signals. The amplitudes, $f_{Da}$, of the Doppler shift variations were observed to noticeably increase (factor of ~2–3) on 1–2 and 5–6 October 2018, while the 20–120 min periods, $T$, of the Doppler shift variations suggest that the wavelike disturbances in the ionosphere are caused by atmospheric gravity waves. The periods and amplitudes of quasi-sinusoidal variations in the Doppler shift, which have been determined for all propagation paths, may be used to estimate the amplitudes, $\delta_{Na}$, of quasi-sinusoidal variations in the electron density. Thus, $T \approx 20$ min and $f_{Da} \approx 0.1$ Hz yield $\delta_{Na} \approx 0.4\%$, whereas $T \approx 30$ min and $f_{Da} \approx 0.2$ Hz give $\delta_{Na} \approx 1.2\%$. If $T \approx 60$ min and $f_{Da} \approx 0.5$ Hz, then $\delta_{Na} \approx 6\%$. The periods $T$ are

found to change within the 15–120 min limits, and the Doppler shift amplitudes, $f_{Da}$, show variability within the 0.05–0.4 Hz limits.

(vi) Discussions and Conclusions are to be modified accordingly.
**Dear Anonymous Referee #1, Thank you very much for this comment**.
The discussion and conclusions **have been modified** accordingly.

(vii) There are scopes for improvement in sentence construction and also in clarity.
**Dear Anonymous Referee #1, Thank you very much for this comment**.
Indeed, you are right. The authors **have made improvements** throughout the text (marked in green).

(B) Other comments :

(i) Line number 20 -25: The authors' statement that "typhoon-induced effects are clear near the midpoint of communication path". This needs clarification when several propagation paths covering SE, S, and NW directions are taken for their study and the mid-point of one propagation path varies from the other. From Figure 1(a) and Figure 5 one can see that the Typhoon trajectory and mid-point of the communication link may come nearer only from October 4 that too in the Harbin –Yomotta, Harbin- Chiba, Harbin-Goyang and Harbin-Hwasenong paths. These points may be cleared and looked into while explaining their observational results.

**Dear Anonymous Referee #1, Thank you very much for this comment**.
    Regarding the midpoints, the movement of the midpoints makes the main contribution to the Doppler shift observed at oblique incidence (see, e.g., Davies, K. Ionospheric radio. Peter Peregrinus Ltd., 2008. ISBN (13 digit) 978-0-86341-186-1, 1989).
    The time when the effects are observed is more important than the place where the typhoon is located. If a surge in the typhoon's power (marked in red in Figure 1a) occurred during the time interval between the noon of 5 and 6 October 2018, when the typhoon and the propagation path midpoints were at ~1,000–1,500 km distances apart, then the response to the typhoon might be naturally expected to occur on October 05 or even October 6.
    **No changes were made to the revised version of the manuscript.**

(ii) Concerning (i) above, it is observed that wave components of 20 mins to 120 mins as claimed as Typhoon–induced effects are not visible, except for the Chiba-Herban link Doppler spectra which provide relatively clear components of 2/3 hrs ( Figure 7). But those signatures appeared even on September 29 -30, and also on October 6 ( also identified by the authors). It is not understood why on September 29 relatively clean wave structure is seen ( Figure 7) when it was in a tropical storm category and only on September 30, the storm attained Typhoon status. Further, the Kong Rey then weakened to category 3 on October 3 and degraded to a tropical storm on October 4, and made landfall on early October 6. Therefore it seems that the superpowerful typhoon effect may not be seen by the authors from October 4 as Kong-Rey penetrated the Chinese mainland, at 18:00hrs on October 3. The authors are suggested to look into their statement and discussion in this background and to provide Doppler spectra for days free from typhoons for clearing the issue.
**Dear Anonymous Referee #1, Thank you very much for this comment**.
This is not entirely true. Quasi-periodic perturbations are observed along all propagation paths except for the Hohhot to Harbin propagation path. The quasi-periodic perturbations in Figure 7 are just more clearly evident. Rather, such structures arise on October 1 and 2, 2018, as well as on October 5 and 6, 2018.

The authors **have constructed** Table 4 showing the main parameters of the wave disturbances in October 2018.

Table 4

Basic parameters of wave disturbances in October 2018

| Radio-wave propagation path | Date | | | |
|---|---|---|---|---|
| | October 01 | October 02 | October 05 | October 06 |
| Hwaseong to Harbin | $T = 120$; 24 min $f_{Da} = 0.4$; 0.1 Hz | $T = 120$; 24 min $f_{Da} = 0.25$; 0.1 Hz | $T = 100–110$; 15 min $f_{Da} = 0.1–0.2$; 0.05 Hz | $T = 120$; 20 min $f_{Da} = 0.3$; 0.05 Hz |
| Chiba/Nagara to Harbin | $T = 60–80$ min $f_{Da} = 0.4$ Hz | $T = 20–30$ min $f_{Da} = 0.2–0.3$ Hz | $T = 30–40$ min $f_{Da} = 0.3$ Hz | $T = 100$ min $f_{Da} = 0.3$ Hz |
| Hailar/Nanmen to Harbin | $T = 80$; 15 min $f_{Da} = 0.4$; 0.05 Hz | $T = 40–50$ min $f_{Da} = 0.2–0.3$ Hz | $T = 40$; 20 min $f_{Da} = 0.2–0.3$ Hz | $T = 40–60$ min $f_{Da} = 0.1–0.2$ Hz |
| Beijing to Harbin | $T = 30–40$; 20–24 min $f_{Da} = 0.2$; 0.1 Hz | $T = 60$; 20–25 min $f_{Da} = 0.2$; 0.1 Hz | $T = 40–60$; 20 min $f_{Da} = 0.2$; 0.05 Hz | $T = 80$; 20–30 min $f_{Da} = 0.2$ Hz |
| Goyang to Harbin | $T = 120$; 30–40 min $f_{Da} = 0.3$; 0.1 Hz | $T = 40–60$; 30 min $f_{Da} = 0.2$; 0.1 Hz | $T = 80–90$ min $f_{Da} = 0.2$ Hz | $T = 100–120$; 20 min $f_{Da} = 0.4$; 0.1 Hz |
| Shijiazhuang to Harbin | $T = 120$min $f_{Da} = 0.3$ Hz | $T = 60$ min $f_{Da} = 0.3$ Hz | $T = 120$ min $f_{Da} = 0.3$ Hz | $T = 120$ min $f_{Da} = 0.1$ Hz |
| Hohhot to Harbin | – | $T = 120$; 40–50 min $f_{Da} = 0.2–0.3$ Hz | – | – |
| Yamata to Harbin | $T = 65–80$ min $f_{Da} = 0.3–0.4$ Hz | $T = 80$ min $f_{Da} = 0.2$ Hz | $T = 40$; 25–30 min $f_{Da} = 0.2$; 0.1 Hz | $T = 70$ min $f_{Da} = 0.2$ Hz |

It should be pointed out that although the typhoon somewhat weakened on October 3 and 4, 2018, it achieved the closest distance to the propagation paths and a surge in the typhoon's power (marked in red in Figure 1a) occurred during the time interval between the noon of 5 and 6 October 2018. Therefore, the ionospheric effects from the typhoon were again observed on October 5 and 6, 2018.

(iii) As the observations are in the oblique mode, the tropospheric effect cannot be ignored, when the contribution varies with the looking angle of the signals from the transmitter. It is suggested that the authors analyze tropospheric/near surface parameters along the path of propagation ( or around) and look for ( if detected) supporting inputs to justify the authors' conclusion ( already suggested above).

**Dear Anonymous Referee #1, Thank you very much for this comment**.

The tropospheric effects cannot arise at 5–10 MHz frequencies, because the non-conducting atmosphere below the ionosphere is treated as free space, with refractive index unity, in the HF frequency range (see, e.g., Budden, K. G., The propagation of radio waves: The theory of radio waves of low power in the ionosphere and magnetosphere, Cambridge University Press, 1988; Davies, K. Ionospheric radio. Peter Peregrinus Ltd., 2008. ISBN (13 digit) 978-0-86341-186-1, 1989).

**Therefore, no changes were made to the revised version of the manuscript.**

(iv) The Wakkanai ionogram may also be examined for such waves. The relevance of figures 3 and 4 are to be clearly spelled out and may be brought into the discussion while explaining the observed Spectral components. The only points of reference of these figures in the MS perhaps are in terms of diurnal variations in layer reflection height and critical frequency status during the period of study.

**Dear Anonymous Referee #1, Thank you very much for this comment**.

We would also like the Wakkanai ionosonde to provide us with ionograms updated every minute. However, Figures 3 and 4 show ionogram measurements acquired with an update rate of one ionogram per 1 hour. Thus, these ionograms cannot give information on the ~20–120-min period wave processes. Nevertheless, they **are used** for analyzing the state of space weather in Section *4 Analysis of the state of the ionosphere*.

( C ) Additional Comments :

(i) Caption of the MS :

The caption may be more appropriate as "Effects of Super-Powerful Tropospheric Western Pacific Phenomenon of September–October 2018 on ionosphere over China: Results from Oblique Sounding".
**Dear Anonymous Referee #1, Thank you very much for this comment**.
The Authors **have changed the title** of this paper (Page 1, Line 1–3).

(ii) Abstract :

The abstract is not well spelled out and needs to be rewritten with clear objectives and approaches. It is necessary to mention what ionospheric parameters the authors are monitoring, and of signal sources.
**Dear Anonymous Referee #1, Thank you very much for this comment**.
The Authors **have rewritten the abstract** as follows (Page 1, Line 14–28).

**Abstract.**
Doppler measurements at oblique propagation paths from the City of Harbin, People's Republic of China (PRC), to ten HF radio broadcast stations in the PRC, Japan, Mongolia, and the Republic of Korea captured the response in the ionosphere to the super typhoon Kong-Rey action from 30 September 2018 to 6 October 2018. The Harbin Engineering University coherent software defined radio system accumulates the database containing the complex amplitudes of the radio signals acquired along 14 propagation paths since 2018. The complex amplitudes are used for calculating the temporal dependences of the Doppler spectra and signal amplitudes, and the Doppler spectra are used to plot the Doppler shift as a function of time, $f_D(t)$, for all rays. The scientific objectives of this study are to reveal the possible perturbations caused by the action of typhoon Kong-Rey, and to estimate the magnitudes of wave parameters of the ionospheric plasma and radio signals. The amplitudes, $f_{Da}$, of the Doppler shift variations were observed to noticeably increase (factor of ~2–3) on 1–2 and 5–6 October 2018, while the 20–120 min periods, $T$, of the Doppler shift variations suggest that the wavelike disturbances in the ionosphere are caused by atmospheric gravity waves. The periods and amplitudes of quasi-sinusoidal variations in the Doppler shift, which have been determined for all propagation paths, may be used to estimate the amplitudes, $\delta_{Na}$, of quasi-sinusoidal variations in the electron density. Thus, $T \approx 20$ min and $f_{Da} \approx 0.1$ Hz yield $\delta_{Na} \approx 0.4\%$, whereas $T \approx 30$ min and $f_{Da} \approx 0.2$ Hz give $\delta_{Na} \approx 1.2\%$. If $T \approx 60$ min and $f_{Da} \approx 0.5$ Hz, then $\delta_{Na} \approx 6\%$. The periods $T$ are found to change within the 15–120 min limits, and the Doppler shift amplitudes, $f_{Da}$, show variability within the 0.05–0.4 Hz limits.

A few examples :

"The ionospheric response to the super typhoon action was clearly observed to occur both on October 1–2, 2018 (when the typhoon was 2,800–3,300 km from the propagation path midpoints .." ( As already identified in other comments (i) above)
The Authors **have rewritten the abstract** as follows (Page 1, Line 14–28).

Similarly: " … on October 5–6, 2018 when the typhoon was 1,000–1,500 km from the midpoints and its energy decreased by a factor of about 4."
The Authors **have rewritten the abstract** as follows (Page 1, Line 14–28).

And also " The ionospheric effects are more pronounced along the nearest propagation paths, whereas no effect was detected along the propagation path at the farthest distance from the typhoon ".
The Authors **have rewritten the abstract** as follows (Page 1, Line 14–28).

These are vague and unspecified sentences carrying no meaning if not mentioned the receive mode and propagation paths of signals. (Line 14–28)
**Dear Anonymous Referee #1, Thank you very much for these comments**. The seeming vagueness is rooted in the general methodology of revealing the effects that are due to any powerful source of energy, presented above. For convenience, it is copied below (Line 199–228, Page 11–12):

Used in this study, the radio system probes the ionosphere at 14 radio propagation path midpoints of the order of 1,000 km distance apart, which are randomly distributed in the ~100–300 km altitude range. Generally, the perturbations under study may be produced either by an impulsive release of energy at a fixed location, as in the case of an earthquake, or by significant releases of energy, which change their location and power as well as persist for a few days, as in the case of a typhoon. On the way from their origin to the radio propagation path midpoints in the upper atmosphere, the perturbations may undergo various nonlinear transformations. In the case of a typhoon event, atmospheric gravity waves, generated via a nonlinear prosses (Drobyazko and Krasil'nikov, 1975), travel up to the ionosphere (partially dissipating their energy for heating the neutral air) and launch secondary gravity waves in the wave breaking regions (see, e.g., Vadas et al., 2003; Vadas and Crowley, 2010). The latter waves in the atmosphere modulate the electron density, which can result in the level of reflection variability, the appearance of a few rays, or in some cases, in diffuseness in the Doppler measurements or spread $F$ in ionograms, which is an indicator of the occurrence of plasma irregularities in the ionosphere (see, e.g., Perkins (1973)). As a consequence, the measurements taken at each midpoint produce a single realization of a random process, which means that the Doppler or amplitude signatures of the sources of perturbations are unrepeatable neither in time nor in space. The observational methodology that enables identification and investigation of such perturbations arising from any deposition of large amounts of energy include the following basic principles invoked consecutively. (i) During the initial stage of employing this methodology, the perturbations originating from a particular powerful source are in principle not distinguishable qualitatively from the perturbations caused by energy released from any other powerful source. (ii) A particular powerful source releasing energy can be associated with any changes in the character of the signal (Doppler shift, Doppler spectrum, the number of rays, discrete spectrum broadening, changes in the signal amplitude, etc.), in accordance with (i) above). This condition is necessary but insufficient. (iii) Intercomparisons between the behavior of radio wave characteristics observed prior to and after an impulsive release of energy must be made. (iv) An intercomparison of the behavior of the radio wave characteristics observed on the day when a particular massive release of energy occurred and during quiet time reference days must be made. Any differences may be due to this particular source. Points (iii) and (iv) serve as control stages. During these stages, the effects that are not associated with the massive release of energy are discarded. (v) The magnitudes of the speeds of propagation of the disturbances must have a physical significance and correspond to known types of waves (seismic, atmospheric gravity waves, infrasound, magnetohydrodynamic). This stage proves sufficiency. (vi) The data acquired over a large (10–14, in the case of the Harbin Engineering University system) number of propagation paths must be consistent with each other to prove sufficiency additionally. (vii) The main signs of a particular powerful source should be observed during other analogous events. First of all, this principle refers to the observed velocities and types of waves. The speeds of perturbations traveling to the radio propagation path midpoints should be contained within the speed limits characteristic of each particular wave type.
(iii) Introduction and Discussion (Page 2–4, 36–40):

In the introduction, the authors introduce relevant references on the need of understanding typhoon-induced effects by coupling dynamics between ocean-lower and upper atmosphere through gravity waves, water vapor condensation, and severe thunderstorms ( to name a few).

The authors have not brought up these parameters and issues either in the analysis or in the discussion. The introduction and discussion aspects should bear relevance.
**Dear Anonymous Referee #1, Thank you very much for this comment**.

    Indeed, the introduction strives to depict an entire broad research effort among scientists from around the world aimed at acquiring a deep understanding of the physical processes that drive the ocean–atmosphere–ionosphere–magnetosphere system. Also, a broad spectrum of instruments to investigate these processes is mentioned.

    The scope of the last paragraph in the introduction **narrows to the scientific objectives achievable with the instrument the authors have created**. The paragraph is as follows (Page 3–4, Line 93–101):

The scientific objectives of this study is to determine the response of the ionosphere to approaching super typhoon Kong-Rey by making use of variations in Doppler spectra, Doppler shift, and HF signal amplitudes recorded at oblique propagation paths, as well as to estimate the parameters of the ionospheric perturbations. An estimate of the joint influence of the typhoon and the dusk terminator is also a phenomenon of interest. The observations were made using the Harbin Engineering University, the People's Republic of China (PRC), multifrequency multiple path coherent software defined radio system for probing the ionosphere at oblique incidence. The data sets discussed in this paper may be obtained from the website at https://dataverse.harvard.edu/dataset.xhtml?persistentId=doi:10.7910/DVN/VHY0L2 (Garmash, 2022), and the software for Passive 14-Channel Doppler Radar may be obtained from https://dataverse.harvard.edu/dataset.xhtml?persistentId=doi:10.7910/DVN/MTGAVH (Garmash, 2021).

(iv) Figures :

Figure 2: (Page 7–8) Here Space weather knowledge is no doubt relevant. These plots however may be omitted and Kp, Dst magnitude statement may be enough to support the status of the days.
**Dear Anonymous Referee #1, Thank you very much for this comment**.
The authors consider the presented in the MS analysis of the state of space weather to be important and its reduction to be inappropriate.
**Therefore, no changes were made to the revised version of the manuscript.**

Figures 3 and 4 (Page 9–10, 13–14):
**Dear Anonymous Referee #1, Thank you very much for this comment**. The relevance of the data in Figures 3 and 4 have already been already identified above as follows.

    Figures 3 and 4 show ionogram measurements acquired with an update rate of one ionogram per 1 hour. Therefore, these ionograms cannot provide information on the ~20–120-min period wave processes. Nevertheless, they **are used for analyzing** the state of space weather in Section *4 Analysis of the State of the Ionosphere*.
**Therefore, no changes were made to the revised version of the manuscript.**

Figure 4: Check for the y-axis (Page 13–14)

Virtual height E is to be replaced with Es.
**Dear Anonymous Referee #1, Thank you very much for this comment**.
In Figure 4, both E and Es virtual heights **are presented**. There is nothing to be replaced.

(v) Clarity of sentence suggested ( examples):

(a) Line 170: "…Main ray and few rays "(Line 187, Page 11)

**Dear Anonymous Referee #1, Thank you very much for this comment**. (Now Line 187, Page 11) Indeed, this phrase sounds stupid. The phrase **has been altered** as follows: all rays under analysis

(b) Line 225-230:: " L 1.5 Hz, broadening±he Doppler spectra exhibit significant, up to and such a diffuseness that the main ray is practically not distinguishable"

**Dear Anonymous Referee #1, Thank you very much for this comment** (Now Line 281-282, Page 19). This misprint **has been corrected** as follows:

the Doppler spectra exhibit significant, up to ±1.5 Hz, broadening and such a diffuseness that the main ray is practically not distinguishable.

(vi ) Reconstruction of sentence necessary ( examples)

(a) Line No 320: "The frequency of this radio wave became greater than the maximum usable frequency and the radio wave penetrated the ionosphere during the second half of the nights. Consequently, the observation of became impossible".

**Dear Anonymous Referee #1, Thank you very much for this comment**.
The sentences **have been reconstructed as follows** (now Line 366-368, Page 31):

The frequency of this radio wave became greater than the maximum usable frequency and the radio wave penetrated the ionosphere during the second half of all nights (see Figure 12). The received signal was absent, and the observation of the ionospheric dynamics became impossible.

(b) line No 325: "During the night of September 30, 2018, the reflection of radio waves took place from the sporadic E layer, resulting in fD(t) » 0 Hz. During October 1, 2018, nighttime, the Doppler shift fD(t) » 0 Hz) as well. During the course of the October 2. 2018, night, the Doppler shift was observed to change from –0.3 Hz to 0.3 Hz, the signal amplitude was observed to exhibit considerable variability, up to 20 dBV, In the course of the October 3–6, 2018 nights, the measurements were ineffective, whereas fD(t) » 0 Hz at daytime."

Note also the highlighted (in yellow) segments.

**Dear Anonymous Referee #1, Thank you very much for this comment**.
The Authors **have re-written** line No 325 (now Line 369-373, Page 31) as follows:

On the UT night of September 30, 2018, the reflection of radio waves took place from the sporadic $E$ layer, resulting in $f_D(t) \approx 0$ Hz, and during the UT night of October 1, 2018, the Doppler shift $f_D(t) \approx 0$ Hz as well. On the UT night of October 2, 2018, the Doppler shift showed changes from –0.3 Hz to 0.3 Hz, while the signal amplitude exhibited considerable variability, up to 20 dBV. During the UT nights of October 3–6, 2018, the measurements were ineffective, whereas $f_D(t) \approx 0$ Hz at daytime.

Final Comment

The paper needs major revision in light of the above suggestions and comments, before being considered suitable for publication.

Please also note the supplement to this comment: https://angeo.copernicus.org/preprints/angeo-2022-24/angeo-2022-24-RC1-supplement. pdf

**Dear Anonymous Referee #1, Thank you very much for this comment**. Your suggestions and comments have helped the Authors to significantly improve the manuscript.

Sincerely,
Authors.

Ann. Geophys. Discuss., referee comment RC1 https://doi.org/10.5194/angeo-2022-24-RC2 , 2022 © Author(s) 2022. This work is distributed under the Creative Commons Attribution 4.0 License.

Comment on angeo-2022-24
Anonymous Referee #2

Referee comment on "Ionospheric Effects over the People's Republic of China from the Super-Powerful Tropospheric Western Pacific Phenomenon of September–October 2018: Results from Oblique Sounding" by Leonid Chornogor et al., Ann. Geophys. Discuss., https://doi.org/10.5194/angeo-2022-24-RC2 , 2022

Reply to Anonymous Referee #2

Dear Anonymous Referee #2,

Thank you very much for your valuable comments that have helped the Authors greatly improve the draft of their paper.

Referee #2 comments are placed together with the Authors' answers (marked in green), and the changes made in the text of the manuscript, are also marked in green.

Authors.
* * *
Dear Dr. Ana Elias!

Thank you for the nomination to evaluate the manuscript "Ionospheric Effects over the People's Republic of China from the Super-Powerful Tropospheric Western Pacific Phenomenon of September–October 2018: Results from Oblique Sounding" by Dr. Chernogor et al. The topic sounds interesting and within the scope of the Annales Geophysicae. The authors performed an interesting experiment to investigate the ionosphere using oblique soundings during the passage of the Super Typhoon Kong-Rey in 2018. I have few comments and suggestions to improve the manuscript to be appreciated by you and the authors and I am willing to revise the manuscript again, if you consider appropriate.

Please, see below, my comments:

Main points

1. From my point of view, the citations of the scientific works is not good form. When there are more than three works cited in the beginning of the statement, I suggest removing those citations to the end of the phrase as the suggestion below. Please, note that it repeats throughout the manuscript.

Lines 63-4: -> Observations of AGWs from meteorological origin have been reported elsewhere (Boška and Šauli, 2001; Šindelarova et al., 2009; Chernigovskaya et al., 2015).

Lines 65-6: -> Recently, theoretical studies on the coupling between the lower and upper atmosphere  by the propagation of AGWs have been published as well (Hickey et al., 2001, 2011; Kuester et al., 2008, Gavrilov and Kshevetskii. 2015, Karpov and Kshevetskii, 2017).

**Dear Anonymous Referee #2, Thank you very much for this comment**. We **have removed** the multiple citation to the end of the phrases throughout the manuscript.

2. I missed connections between the paragraphs of the Introduction. It is not clear how the state of art of the investigated topic and how are, in fact, the contributions of the authors to

understanding the coupling between the typhoon and the ionosphere. I would suggest revising the Introduction to improve the text itself.

**Dear Anonymous Referee #2, Thank you very much for this comment**. We have re-organized the paragraphs in the Introduction (**the changes are marked in green**) as follows (**Line 30–101, Page 2–4**):

[revised manuscript text omitted]

3. In the present manuscripts, the authors are assuming that the periodic oscillations in the Doppler shift signal might be gravity waves from typhoons. They can be, but gravity waves can be produced by several other atmospheric processes, even small scale structures compared to typhoons. So, in this case, from my point of view, it will be very welcome, further analysis on the periodic structure in order to resolve the phases and find out the propagation direction of the wave structures. Certainly, they are propagating from the region of the typhoon. If the authors could address this point, the scientific discussion on gravity waves will be stronger and more convincing.

**Dear Anonymous Referee #2, Thank you very much for this comment.** Regarding "to resolve the phases and find out the propagation direction", the propagation path midpoints are scattered within an ~100–300 km altitude range, which makes phase measurements impossible. Nevertheless, the Authors **have taken Referee #2 suggestion into account**. The Authors have **constructed Table 4, which is described in the following paragraph**, which we have inserted into the Discussion section (after the first two paragraphs) (**Line 410–427, Page 36–37**). Especially the Authors are grateful to Referee #2 for the last sentence (in **bold** type, here) in the following paragraph:

In order to find out that the observed Doppler shift variations are associated with the typhoon, the Doppler variations were low-pass filtered, and the Doppler variations, with periods of greater than 40 min, were found to occur during the period 10:00–14:00 UT, October 02, 2018, along all propagation paths. A characteristic feature, a fading, which could be traced in all temporal dependences of the identified Doppler variations, was selected for analyzing. The UT moments, $t^*$, when this feature arrived at each propagation path midpoint are presented in Table 4. At 12:00 UT on October 02, 2018, the typhoon center was located at (18.9° N, 131.2° E) at the distances $D$ from the propagation path midpoints, with the midpoint of the 9.750 MHz propagation path being closest (2,492 km) to the typhoon center, while other midpoints were found to be at (2,492 + $\Delta D$) km ranges, where the characteristic feature arrived with time delays of $\Delta t$ with respect to the arrival time at the 9.750 MHz midpoint. As can be seen in Table 4, the $\Delta D$ and $\Delta t$ yield the values of the apparent speeds, $v$, quite close to each other. These estimates testify to the adequacy of the assumption that the propagation of the disturbances from the typhoon is the cause of the observed Doppler shift variations. The mean value of the speed of the strongest 60 – 70-min period component, estimated to be 205±6 m/s, corresponds to a TID with wavelength equal to approximately 800 km. **Taking a look at the Kong-Rey trajectory in Figure 1, one can notice that the TIDs traveled northwestward in this case, contrary to the southwestward direction observed in this area of the world in the climatological study by Shiokawa et al. (2003).**

Table 4. Distances $D$ over which TIDs traveled at apparent speeds $v$ and arrived at the propagation pass midpoints from the center of typhoon Kong-Rey with relative time delays $\Delta t$ at the UT moments $t^*$.

| $f$ (MHz) | 6.015 | 6.055 | 6.080 | 6.175 | 6.600 | 9.500 | 9.520 | 9.750 |
|---|---|---|---|---|---|---|---|---|
| $D$ (km) | 2,574 | 2,454 | 3,296 | 2,826 | 2,595 | 2,803 | 2,963 | 2,492 |
| $\Delta D$ (km) | 120 | 38 | 842 | 372 | 141 | 349 | 509 | 0 |
| $t^*$ (UT) | 11:00 | 10:53 | 12:00 | 11:20 | 11:03 | 11:15 | 11:25 | 10:50 |
| $\Delta t$ (min) | 10 | 3 | 70 | 30 | 12 | 28 | 38 | - |
| $v$ (m/s) | 200 | 210 | 200 | 205 | 195 | 205 | 220 | - |

Specific points:

1. Line 93: The minimum value of the pressure is different from the value presented in the first paragraph of the introduction.
**Dear Anonymous Referee #2, Thank you very much for this comment.** Indeed, they should differ because the Introduction is concerned with features of typhoons in general, while Line 93 refers to typhoon Kong-Rey.
**No changes were made to the revised version of the manuscript.**

2. I also missed some citations on periodic gravity waves/MSTID, which could sustain the argumentation of the authors. Please, see some suggestions:

**Dear Anonymous Referee #2, Thank you very much for this collection of interesting studies.** We **have included** the suggested citations into the Introduction section (marked in green) after Line 72, as follows (Line 72–85, Page 3):

[revised manuscript text omitted]

**Dear Anonymous Referee #2, Thank you very much for your comments**. Your suggestions and comments have helped the Authors to significantly improve the manuscript.

Sincerely,
Authors.

List of All the Changes

Authors have corrected the first coauthor surname as follows: Chernogor
Authors have inserted references to 13 papers in the References section (marked in yellow or green)

List of all the changes made in the revised version of the manuscript in accordance with each suggestion and recommendation given by Anonymous Referee #1

**Anonymous Referee #1 has numbered his suggestions and recommendations in Roman Numerals as follows (marked in pink here):**

**(A) General Comments :**
(i) – (vii)

**(B) Other comments :**
(i) – (iv)

**( C ) Additional Comments :**
(i) – (iv)

(v)
(a)
(b)

(vi)
(a)
(b)

**Final Comment :**

**List of all the answers and changes the Authors have made are numbered consecutively in Arabic numerals (marked in yellow).**

**(A) General Comments :**

(i)
**1.** General methodology for revealing perturbations arising from any powerful source of energy **has been written and placed** at the end of the section "5 Instrumentation and techniques" (Line 194–228) and **the corresponding references to three papers** have also been **inserted into the Reference list**.

**2.** The authors **have constructed Table 5** showing the main parameters of the wave disturbances in October 2018 (Line 452–455, Page 37–38).
(ii)
**3.** The authors **have redone Figures 6–13** (**pages 17,18, 20–25, 27–30, 32–35**) to show additional reference days (September 27 and 28, 2018 and October 7 and 8, 2018), as was recommended by Anonymous Referee #1.

(iii)
**4.** The Authors have explained in detail why they cannot present profiles of any lower atmospheric/near-earth parameter during the period and around the locations covered by the study, to identify the features present therein with the wave components provided by their Doppler analysis.
**No changes were made to the revised version of the manuscript.**

(iv) The Wakkanai ionogram

**5.**
No changes were made to the revised version of the manuscript.

(v)The abstract, the basic key to the contents of the paper is not well spelled out and needs to be rewritten with clear objectives and approaches.
The abstract has been updated, (Line 14–28, Page1).

**( C ) Additional Comments :**

(i)
**6.** The Authors **have changed the caption of the paper** (Page 1, Line 1–3).

(ii)
7. The **abstract has been re-written** (Page 1, Line 14–28)..

In addition, the Authors have explained that the seeming vagueness of some statements in the old abstract is rooted in the **general methodology** (Line 199–228, Page 11–12) of revealing the effects that are due to any powerful source of energy.

(iii) Introduction and Discussion (Page 2–4, 36–40)
**8.** The Authors explain that the introduction strives to depict an entire broad research effort among scientists from around the world making use of a broad spectrum of instruments, whereas the Authors' achievable scientific objectives are narrowed by the capabilities of the instrument the Authors have created, and therefore, the Authors cannot have brought up those broad research effort among scientists from around the world neither in the analysis nor in the discussion. **Nevertheless, Authors have made numerous corrections to the Introduction section** (Page 2–4, 36–40) (marked in green).

Also, the Authors **have re-written** the last paragraph (Page 3–4, Line 93–101) in the Introduction section as follows:
The scientific objectives of this study is to determine the response of the ionosphere to approaching super typhoon Kong-Rey making use of variations in Doppler spectra, Doppler shift, and HF signal amplitudes recorded at oblique propagation paths, as well as to estimate the parameters of the ionospheric perturbations. An estimate of the joint influence of the typhoon and the dusk terminator is also a phenomenon of interest. The observations were made using the Harbin Engineering University, the People's Republic of China (PRC), multifrequency multiple path coherent software defined radio system for probing the ionosphere at oblique incidence. The data sets discussed in this paper may be obtained from the website at https://dataverse.harvard.edu/dataset.xhtml?persistentId=doi:10.7910/DVN/VHY0L2 (Garmash, 2022), and the software for Passive 14-Channel Doppler Radar may be obtained from the website at https://dataverse.harvard.edu/dataset.xhtml?persistentId=doi:10.7910/DVN/MTGAVH (Garmash, 2021).

(iv) Figures :

Figure 2: (Page 7–8)
**9.** The authors consider the presented in the MS analysis of the state of space weather to be important and its reduction to be inappropriate.
No changes were made to the revised version of the manuscript.

Figures 3 and 4 (Page 9–10, 13–14)
**10.** The Authors have explained that ionogram measurements acquired with an update rate of one ionogram per 1 hour cannot give information on the ~20–120-min period wave processes.
No changes were made to the revised version of the manuscript.

Figure 4
**11.** In Figure 4, both E and Es virtual heights are presented.
**No changes were made to the revised version of the manuscript.**

(v)

(a) Line 170:
**12.** The phrase has been altered (Now Line 187, Page 11).

(b) Line 225-230::
**13.** This misprint has been corrected (Now Line 281-282, Page 19).

(vi )
(a) Line No 320
**14.** The sentences have been reconstructed (now Line 366-368, Page 31).

(b) line No 325:
**15.** The Authors have re-written Line 325 (now Line 369-373, Page 31).

**16.** The highlighted segments have been corrected (now Line 369-373, Page 31).

**Final Comment**

The paper needs major revision in light of the above suggestions and comments, before being considered suitable for publication.

**17.** The Authors have considered all Anonymous Referee #1 suggestions and comments.

**Dear Anonymous Referee #1, Thank you very much for this comment**. Your suggestions and recommendations have helped the Authors to significantly improve the manuscript.

Sincerely,
Authors.

**List of all the answers and changes made in the revised version of the manuscript in accordance with each suggestion and recommendation given by Anonymous Referee #2**

**Anonymous Referee #2 has numbered his comments and suggestions in Arabic Numerals as follows (marked in pink here):**

**Main points**

**1.**

**Lines 63-4:**

**Lines 65-6:**

**2.**

**3.**

**Specific points:**

**1. Line 93:**

**2.**

**List of all the changes the Authors have made continues in Arabic numerals (marked in green).**

**Main points**

**1.**

**Lines 63-4:**

**Lines 65-6:**

**18.** The Authors **have removed** the multiple citation to the end of the phrases throughout the manuscript.

**2.**

**19.** The Authors **have revised the entire** Introduction section for cohesion.

**3.**

**20.** The Authors **have added** a paragraph and Table 4 to the Discussion section (Line 407–421).

**Specific points:**

**1. Line 93:**

The Authors have explained the difference between the minimum value of the pressure in the Introduction and in the section "2 General information on the super typhoon Kong-Rey" **No changes were made to the revised version of the manuscript.**

**2**.

**21.** The Authors **have included** the suggested by Anonymous Referee #2 **citations** into the Introduction section after Line 72 ((Line 73–84): **215 words altogether**) and have also inserted the references to eight papers into the Reference list.

**Dear Anonymous Referee #2, Thank you very much for your comments**. Your suggestions and comments have helped the Authors to significantly improve the manuscript.

Sincerely,
Authors.

---

## Referee Report (RR1)

Review report on the Manuscript:  "Effects of super-powerful tropospheric Western Pacific phenomenon 2 of September–October 2018 on ionosphere over China: Results from the oblique sounding "

Author(s): Leonid Chornogor, Kostiantyn Garmash, Qiang Guo, Victor Rozumenko, and Yu Zheng
MS No. angeo-2022-24

**Comments** :

 The authors have made substantial improvements in the quality of the paper,  the suggested modifications are more or less incorporated to make the MS a coherent reading, and the abstract is now well-framed to provide a clear goal of objectives and approaches of the work.

Regarding "the periodic components of 20 min to 120 min at the ionospheric heights as reflected in the received signals are the effects of the superpowerful typhoon of September – October 2018" the explanation if made explicit would be better.

Figures 6 to 13 have a scope for improvement. The Doppler shifts of October 7 and 8 as displayed in these figures are seen to be somewhat significant and as these are visible in almost all the paths, the authors may at least mention their presence.

The paper may be accepted for publication with the final decision from the Editor.

---

## Author Response (AR2)

Report of Reviewer #2:

Review report on the Manuscript: "Effects of super-powerful tropospheric Western Pacific phenomenon 2 of September–October 2018 on ionosphere over China: Results from the oblique sounding" Authors: Leonid Chernogor, Kostiantyn Garmash, Qiang Guo, Victor Rozumenko, and Yu Zheng MS No. angeo-2022-24

Comments: The authors have made substantial improvements in the quality of the paper, the suggested modifications are more or less incorporated to make the MS a coherent reading, and the abstract is now well-framed to provide a clear goal of objectives and approaches of the work. Regarding "the periodic components of 20 min to 120 min at the ionospheric heights as reflected in the received signals are the effects of the superpowerful typhoon of September – October 2018" the explanation if made explicit would be better. Figures 6 to 13 have a scope for improvement. The Doppler shifts of October 7 and 8 as displayed in these figures are seen to be somewhat significant and as these are visible in almost all the paths, the authors may at least mention their presence. The paper may be accepted for publication with the final decision from the Editor.

Reply to Anonymous Referee #2

Comment #1.
Regarding "the periodic components of 20 min to 120 min at the ionospheric heights as reflected in the received signals are the effects of the superpowerful typhoon of September – October 2018" the explanation if made explicit would be better.

**Dear Anonymous Referee #2, Thank you very much for this comment**. Indeed, this phrase sounds vague, and it has been **deleted**. Instead, we retained explicit statements concerning the 20 min to 120 min period components. First, Table 4 has been constructed to prove the adequacy of the assumption that the Doppler shift variations are caused by the action of the typhoon. Second, the periods of wave disturbances observed are summarized in Table 5. Third, explicit statements about the 20 min to 120 min period components are made in both Abstract and Conclusions sections.

Comment #2.
Figures 6 to 13 have a scope for improvement. The Doppler shifts of October 7 and 8 as displayed in these figures are seen to be somewhat significant and as these are visible in almost all the paths, the authors may at least mention their presence.

**Dear Anonymous Referee #2, Thank you very much for this comment**. Indeed, the Doppler shifts registered on October 7 and 8 are greater than those observed during the rest of the time interval displayed in these figures. The typhoon originated on September 29, 2018, and ceased to exist on October 6, 2018, whereas "on October 7, 2018, a moderate magnetic storm started, with $K_{p\max} = 5.3$, and $D_{st\min} \approx -53$ nT" (Section 3 "Analysis of the state of space weather").
  Since the magnetic storm occurred after the typhoon ceased to exist, it has nothing to do with the purpose of the paper, although this observational fact justifies a meticulous examination of space weather made by the Authors.
  Nevertheless, on Referee's advice, in order to not divert the reader's attention from the purpose of the paper, the Authors have inserted the following sentence at the end of Section 3 "Analysis of the state of space weather":

The magnetic storm occurred after the typhoon ceased to exist, from October 7 through 9, 2018, when the Doppler shifts exhibited variations greater than those observed under the action of the typhoon, which justifies the need for a thorough analysis of space weather.

**Dear Anonymous Referee #2, Thank you very much for your comments**. Your suggestions and comments have helped the Authors to significantly improve the manuscript.

Sincerely,
Authors.

**List**
**of the changes made in response to Reviewer # 2 additional minor comments**

The changes (marked in green) have been made in the last paragraph of Section 3 "Analysis of the state of space weather" (Line 140–142). The last paragraph of Section 3 now looks as follows:

Thus, solar activity and the state of space weather were conducive to observing the ionospheric effects from typhoon Kong-Rey. Only on 7 October 2018, a moderate magnetic storm started, with $K_{p\max} = 5.3$, and $D_{st\min} \approx -53$ nT. Thus, the days of 26 and 27 September 2018, the first half of 28 and entire 29 September 2018, and partially the days of 1 and 2 October 2018 were weakly disturbed. The magnetic storm occurred after the typhoon ceased to exist, from October 7 through 9, 2018, when the Doppler shifts exhibited variations greater than those observed under the action of the typhoon, which justifies the need for a thorough analysis of space weather. Consequently, 28 September 2018, and 4 October 2018, have been chosen to be quiet time references.